# Dual electrical stimulation at spinal-muscular interface reconstructs spinal sensorimotor circuits after spinal cord injury

Kai Zhou [1,2,9], Wei Wei[1,9], Dan Yang[1,3,9], Hui Zhang[1], Wei Yang[1], Yunpeng Zhang[4], Yingnan Nie[5], Mingming Hao[6,7], Pengcheng Wang[8], Hang Ruan[8], Ting Zhang [6], Shouyan Wang[5] & Yaobo Liu [1,2] ✉

The neural signals produced by varying electrical stimulation parameters lead to characteristic neural circuit responses. However, the characteristics of neural circuits reconstructed by electrical signals remain poorly understood, which greatly limits the application of such electrical neuromodulation techniques for the treatment of spinal cord injury. Here, we develop a dual electrical stimulation system that combines epidural electrical and muscle stimulation to mimic feedforward and feedback electrical signals in spinal sensorimotor circuits. We demonstrate that a stimulus frequency of 10−20 Hz under dual stimulation conditions is required for structural and functional reconstruction of spinal sensorimotor circuits, which not only activates genes associated with axonal regeneration of motoneurons, but also improves the excitability of spinal neurons. Overall, the results provide insights into neural signal decoding during spinal sensorimotor circuit reconstruction, suggesting that the combination of epidural electrical and muscle stimulation is a promising method for the treatment of spinal cord injury.

Electrical stimulation can augment or modify neuronal function and can have therapeutic benefits for certain neurological disorders[1–4]. Indeed, there is evidence that enhancing spinal excitability with either epidural or transcutaneous stimulation can restore some volitional motor output after spinal cord injury (SCI)[5,6]. For example, lumbosacral epidural stimulation temporarily improves locomotor and autonomic function in both rodents and humans with SCI[7]. When combined with overground locomotor training enabled by a weight-supporting device, epidural electrical stimulation (EES) promotes extensive reorganization of residual neural pathways that improves

locomotion after the stimulation has stopped or stimulation therapy has ended[8–10]. Although a technological platform has been established to optimize neuromodulation in real time to achieve high-fidelity control of leg kinematics during locomotion in animals and patients[11,12], the parameters of electrical stimulation to promote neural circuit remodeling and neural axon regeneration are poorly understood. It remains unclear whether electrical stimulation can be used to precisely reassemble neural-circuit structures and restore function after SCI. In addition, the exact mechanism underlying the reconstruction of spinal cord neural circuits with electrical stimulation

[1]Jiangsu Key Laboratory of Neuropsychiatric Diseases and Institute of Neuroscience, Soochow University; Clinical Research Center of Neurological Disease, The Second Affiliated Hospital of Soochow University, Suzhou 215123, China. [2]Co-innovation Center of Neuroregeneration, Nantong University, Nantong 226001, China. [3]Department of Anatomy, School of Basic Medical Science, Guizhou Medical University, Guiyang 550025, China. [4]Suzhou Institute of Biomedical Engineering and Technology, Chinese Academy of Sciences, Suzhou, Jiangsu 215163, China. [5]Institute of Science and Technology for Brain-Inspired Intelligence, Fudan University, Shanghai 200433, China. [6]i-Lab, Key Laboratory of Multifunctional Nanomaterials and Smart Systems, Suzhou Institute of Nano-tech and Nano-bionics, Chinese Academy of Sciences, Suzhou, Jiangsu 215123, China. [7]Ningbo Medical Centre Lihuili Hospital, Ningbo, Zhejiang 315048, China. [8]Institutes of Biology and Medical Sciences, Soochow University, Suzhou, Jiangsu 215123, China. [9]These authors contributed equally: Kai Zhou, Wei Wei, Dan Yang. ✉e-mail: liuyaobo@suda.edu.cn

remains unknown. These challenges have greatly limited the application of electrical neuromodulation techniques to treat SCI.

To address these issues, we utilized spinal sensorimotor circuits that govern the contraction of the hindlimb flexor, i.e., the tibialis anterior (TA) muscle, as the focus of our study. It has been reported that paired stimulation of the motor cortex and cervical spinal cord can enhance motor responses through their convergence and alter the connections between specific neural circuits[13–15]. Recent studies have shown that the interaction of stimulation frequency and pulse interval at different sites in the spinal cord can produce various motor effects in rats with SCI[16,17]. Therefore, a dual electrical stimulation system combining epidural electrical and muscle stimulation (EEMS) was established in our research using feedforward EES[7,18,19] combined with biofeedback after TA muscle electrical stimulation[20,21]. EEMS was used to mimic the operation of sensorimotor neural circuits via electrical stimulation of the spinal cord to simulate motor commands. The muscle was then stimulated to simulate sensory feedback. To prevent overlap of the dual stimulation effects, muscle stimulation (MS) was administered after the end of the spine-evoked response. Therefore, the duration from the start of the spinal pulse to the end of the muscular response was used as the stimulation interval, simulating the closed-loop conduction of neural signals within a sensorimotor neural circuit. As a type of field potential stimulation, epidural electrical stimulation has a wide range of activation. Electrical stimulation of the TA muscle activates sensory feedback pathways associated with the TA muscles. With EEMS at the spine and muscle, we could more specifically target the sensorimotor circuit for modulation. Complete T9 spinal cord transection leads to the complete interruption of supraspinal input to the lumbosacral neural circuit, providing an ideal research model to investigate the neuroplasticity of these spinal sensorimotor circuits after electrical stimulation. Briefly, this study leads to a deeper understanding of the role of neuromodulation in neural circuit reassembly and suggest a potential framework for clinical application after neural trauma.

## Results

### EEMS system

In mice with complete SCI, interruption of supraspinal innervation leads to disassembly of interneuronal connections in spinal sensorimotor circuits below the level of injury; these circuits transmit neural signals between the spine and muscles and control the coordinated contraction of flexor and extensor muscles[22–24]. Previous studies have demonstrated that the feedforward and feedback of neural signals in spinal sensorimotor circuits are necessary to maintain sensorimotor functional output in uninjured mice[25–27]. To investigate whether EEMS leads to reassembly of interneuronal connections in spinal sensorimotor circuits, we selected TA sensorimotor reflex circuits that govern the contraction of the right hindlimb flexor TA muscle as the focus of our study. We established a EEMS system to mimic sensory feedback and feedforward muscle contraction loops with the integration of EES and MS (Fig. 1a).

To determine the proper location for implantation of epidural electrodes in the spinal cord, a retrograde, adeno-associated virus (AAV) vector carrying enhanced green fluorescent protein (EGFP; AAV2/2Retro-hSyn-EGFP-WPRE-pA) was injected into the TA muscle of uninjured C57BL/6J mice to trace the retrograde projection of the corresponding motoneurons in the spinal cord. The motoneurons innervating the TA muscle were mainly distributed in the L2–L4 segment of the spinal cord based on analysis of continuous coronal and sagittal sections of the lumbar spinal cord (Fig. 1b, c). A fine needle electrode was designed and manufactured to minimize damage caused by electrode implantation (Fig. S1a). In mice with T9 spinal transection, this epidural (i.e., EES) electrode was placed in the dura mater of the L2–L4 segment of the spinal cord during T9 transection to avoid secondary damage due to electrode implantation (Fig. 1a, d). At

4 weeks post-implantation, magnetic resonance imaging was used to determine the accuracy of positioning of the implanted electrode (Fig. 1d). A peripheral electrode was synchronously implanted only in the TA muscle of the right hindlimb as the MS electrode (Fig. 1a and Fig. S1b).

To analyze the conduction of electrical signals in the TA sensorimotor reflex circuit under normal physiological conditions, the spinal cord (L2–L4) was stimulated along the midline in anesthetized, uninjured mice. The EES was administered to evoke a response in the TA muscle. We define this evoked response recorded in the muscle as spinal cord-evoked potential (SCEP). According to literature reports[28,29] and analysis of our data, the SCEP recorded in muscle had early latency responses (ERs, 4–7 ms), medium latency responses (MRs, 7–10 ms), and late latency responses (LRs, 10–15 ms) (Fig. 1e and Fig. S2a). Under normal physiological conditions, we calculated that the latency from the end of the spinal pulse was $2.83 \pm 0.22$ ms (Fig. S2b), whereas that from the start of the spinal pulse was $5.83 \pm 0.22$ ms (Fig. S2c).

Pharmacological tests were subsequently conducted to isolate the ERs, MRs, and LRs that govern feedforward transmission in spinal sensorimotor circuits. The anesthetized mice underwent EES, while recording the evoked potentials in the muscles and analyzing the changes in SCEPs before and after drug administration. To evaluate the signal pathway corresponding to ERs, we blocked sodium currents through local intrathecal delivery of tetrodotoxin (TTX) into spinal cord segments L5/L6 (Fig. 1f). The intrathecal administration of TTX primarily affected the spinal cord and the dorsal roots in the portion that enter the spinal cord, with a lesser impact on efferent nerves and the peripheral nerves of DRG, which inhibited chemical neurotransmission evoked by action potentials[28–32]. In this study, TTX was found to abolish both MRs and LRs, but ERs were not affected (Fig. 1g, h). These results suggested that ER was not dependent on chemical neurotransmission of spinal cord neurons but rather relied on the direct recruitment of motor axons. Meanwhile, the direct recruitment of afferent nerves may also contribute to ER. Furthermore, intrathecal administration of the alpha2-adrenergic receptor agonist tizanidine not only interfered with pain transmission but also suppressed group II reflex pathways[33] (Fig. 1f). The amplitude of the LRs decreased substantially after high-dose intrathecal injection of tizanidine, but neither ERs nor MRs were affected (Fig. 1g, h). These results indicated that only the LRs of SCEPs were relevant to the polysynaptic response in feedforward transmission mediated through Group II/Ib interneurons of spinal sensorimotor circuits. Consequently, the SCEPs were subjected to various intensities of EES (Fig. 1i). LR amplitudes peaked when the EES intensity was 120 μA, indicating full activation of feedforward transmission mediated by spinal sensorimotor circuits. Considering individual differences among mice, EES intensity was adjusted between 100 μA and 140 μA in the EEMS system to transmit feedforward signals to the TA muscle. In addition, we analyzed the latency of ERs under EES of different amplitude. The ER latency did not change significantly under stimulation conditions with different amplitudes, although the absolute value of ER latency decreased gradually with the increase in EES intensity (Fig. S2d, e). Spinal epidural recording confirmed the effective response (unimodal peak) with stimulation of 400 μA administered to the TA muscle (Fig. 1j).

To determine whether MS could activate sensory fibers, we first examined the expression of c-Fos, a neuronal activity marker, in the lumbar spinal cord 1.5 h after MS (Fig. S3a). The proportion of c-Fos[+] neurons ipsilateral to the stimulation as well as contralateral to the stimulation was calculated (Fig. S3b). The results showed that the proportion of c-Fos[+] neurons on the dorsal horn ipsilateral to the stimulation was significantly greater than that of the contralateral side (Fig. S3c). Then, we injected the rAAV-hSyn-GCaMP6f virus into the dorsal horn of the spinal cord of mice to facilitate GCaMP6f expression in dorsal-horn neurons. At the same time, an optical fiber was

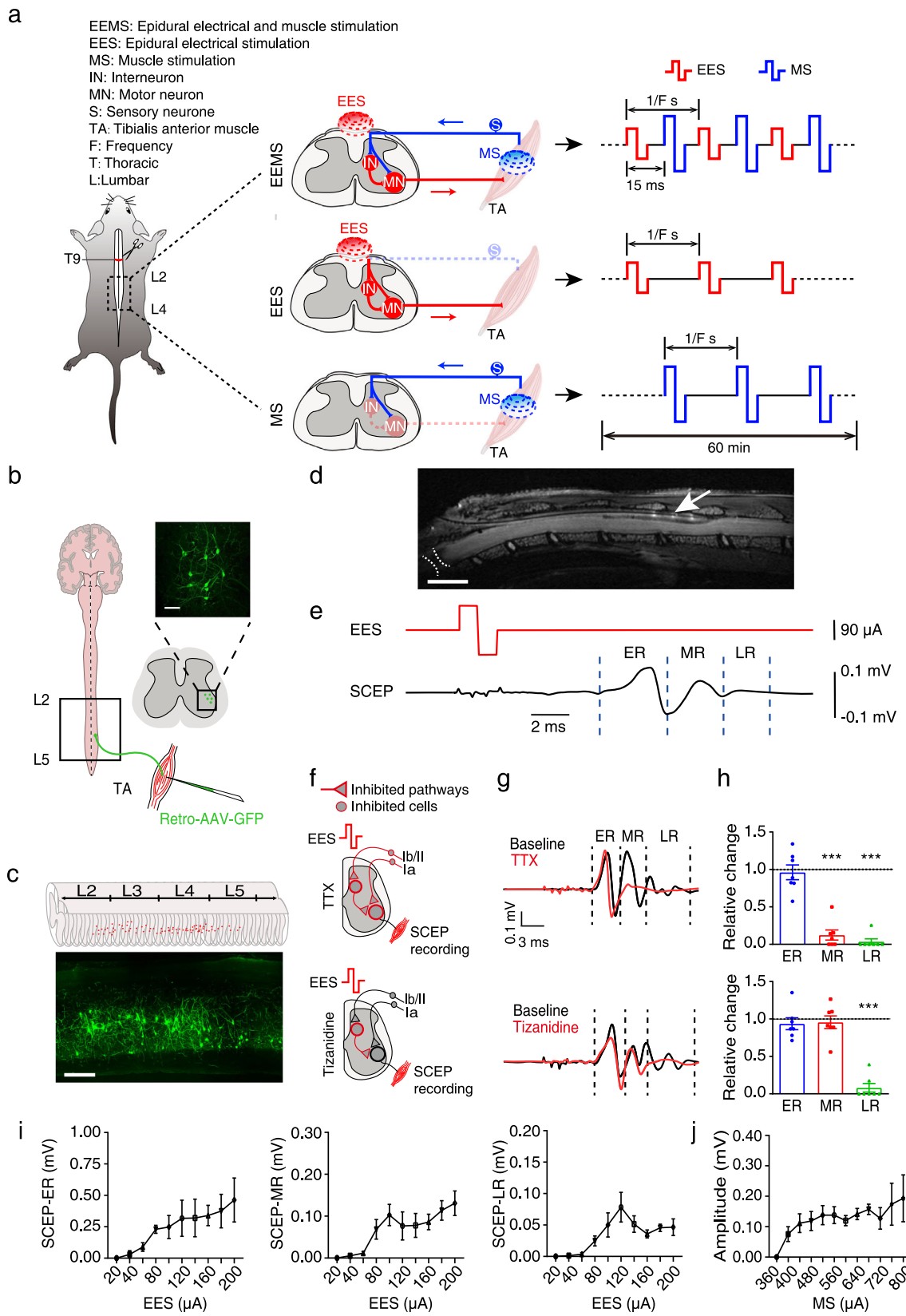

implanted into the dorsal horn of the spinal cord and fixed with dental cement. At 10 days post-surgery, the electrode was implanted in the right hind limb of the anesthetized mice (Fig. S3d). At the same time, changes in $Ca^{2+}$ signaling (measured as fluorescence) of spinal dorsal-horn neurons were recorded during MS. The results showed that 40 consecutive MS pulses could steadily increase $Ca^{2+}$ signaling in the

dorsal-horn neurons (Fig. S3e). In addition, we recorded evoked signals in the spinal cord and muscle simultaneously after MS (Fig. S3f), and the latency of evoked signals was also measured (Fig. S3g). Consistent with previous studies[34–36], the latency of evoked signals in the spinal cord following MS was measured as $5.279 \pm 0.1466$ ms. And, the latency of evoked signals in muscle after MS was measured

**Fig. 1 | Characteristics of the EEMS system at the interface of the spinal cord and muscle. a** Schematic showing the application of the EEMS system, which comprises EES and MS, to the sensorimotor circuit in SCI mice. The interval between spinal and muscular stimulation was set to 15 ms. The interval between the two EES pulses or two MS pulses depended on the stimulation frequency. **b** AAV2/2Retro-hSyn-EGFP-WPRE-pA was injected into the TA muscles of intact mice to label motoneurons. Scale bar, 100 μm. The experiment was repeated 3 times independently with similar results. **c** Spatial distribution of motoneurons innervating the TA muscles. Scale bar, 300 μm. The experiment was repeated 3 times independently with similar results. **d** Correct positioning of the electrode was confirmed by magnetic resonance imaging analysis at 4 weeks after SCI; the white arrow points to sites of contact of the electrode, and the dashed white lines indicate the SCI site. Scale bar, 2.5 mm. The experiment was repeated one time. **e** EES (red line) was administered, and SCEP (black line) was recorded with the TA muscles of intact mice. ER: early latency response, MR: medium latency response, LR: late latency response. **f** The schematic illustrates which neurons, fibers, and/or circuits may be inhibited as stimulation signals delivered to the muscles from the spinal cord during each experimental operation in anesthetized intact mice. **g** Representative SCEP recorded in the muscle with different mice during repeated EES (1 Hz) and after the administration of tetrodotoxin (TTX) or tizanidine. Each waveform represents the average of the responses to 50 EES. **h** Histograms for TTX (upper) and tizanidine (lower) conditions showing the relative change in the SCEPs as compared with baseline ($n = 7$ mice per group). **i** ERs, MRs, and LRs induced by EES ($n = 8$ mice per group). **j** Curve showing the electrical signal received in the spinal cord after MS ($n = 5$ mice per group). Schematics in **a–c**, and **f** were created with BioRender.com. Data represent the mean ± SEM; Statistical analysis was performed using two-tailed unpaired *t*-test (**h**), ****p* < 0.001.

5.713 ± 0.1747 ms (Fig. S3h). These results indicate that MS activates the sensory circuit by directly activating sensory fibers, thereby eliciting epidural potentials. Thus, in subsequent experiments, TA muscles received 400 μA of stimulation via the EEMS system to transmit sensory feedback to the spinal cord.

Our EEMS system was designed such that EES and MS have a timing-sequence stimulation pattern. For example, EES was administered first, followed by MS, and then EES was administered again, and so on. The stimulus frequency range we selected was 1–40 Hz, to prevent overlap of the dual-stimulation effects; the sum of the time from the beginning of the spinal pulse to the end of the muscular response and the time from the beginning of the muscular pulse to the end of the spinal response was <25 ms (1/40 s). Then, under normal physiological conditions, we calculated that the time from the start of the spinal pulse to the end of muscular response was 13.34 ± 0.66 ms (Fig. S4a), and the time from the start of the muscular pulse to the end of the spinal response was 8.35 ± 0.49 ms (Fig. S4b). Therefore, the interval between spinal stimulation and muscular stimulation was set to 15 ms to ensure that the spinal-evoked muscle response was complete before subsequent EES.

### Restoration of polysynaptic electrical conduction in spinal sensorimotor circuits by EEMS in SCI mice

The current research suggests that electrical stimulation at a certain frequency range can contribute to the recovery of sensory and motor functions following spinal cord injury[5,17,37,38]. To investigate the specific frequencies in the EEMS system that play a crucial role in electrical stimulation, we chose to identify the frequency characteristics of effective electrical stimulation in the frequency range 1–40 Hz. SCI mice and intact mice were implanted with spinal cord electrodes on the same day. Seven days after electrode implantation, SCEPs were detected, and electrical stimulation training began for 3 weeks; SCEPs were then recorded on the day after training ended (Fig. 2a).

Mice in the sham and untrained groups did not undergo electrical stimulation training, and SCEPs were recorded 7 days and 29 days after electrode implantation. (Fig. 2b). Electrical stimulation training started 1 week after SCI (ES onset) and continued for 3 weeks (Fig. 2b). SCEPs were recorded the day after the end of electrical stimulation training (Fig. 2b). MRs and LRs were clearly observed in the sham group (115.6 ± 17.14 μV, 37.02 ± 7.102 μV) yet were almost nonexistent in the untrained group (13.78 ± 2.883 μV, 0 μV), although ERs were unchanged (Fig. 2c–e). This suggested that polysynaptic electrical conduction in the sensorimotor circuits was interrupted by SCI. MRs and LRs were almost undetectable in the 1- to 40-Hz EES as well as 1- to 40-Hz MS groups 4 weeks after SCI (Fig. 2c–e). For the 10- to 20-Hz EEMS groups, however, MR and LR amplitudes increased significantly after 3 weeks of EEMS training as compared with mice in the untrained group (Fig. 2c–e).

To clarify the effect of spinal cord high-current stimulation training, 300 μA high-current electrical stimulation was applied to mice in the EES group (EEShc). The MR and LR amplitudes did not improved significantly in the 1- to 40-Hz EEShc groups compared with the untrained group at 4 weeks after SCI (Fig. 2d, e).

There were no statistically significant differences in the latency and duration of ERs among the sham, 1- to 40-Hz EEMS, 1- to 40-Hz EES, 1- to 40-Hz MS and 1- to 40-Hz EEShc mice as compared with the untrained mice, and the latency of all ERs was almost always 4–7 ms (Fig. S5a, b). Compared with the untrained group, the absolute value of ER duration in the sham group and the 10- to 20-Hz EEMS groups was lower, whereas the duration of ERs did not different significantly among the three groups (untrained group versus sham group, $p = 0.1483$; untrained group versus 10- to 20-Hz group, $p = 0.1434$) (Fig. S5b). Thus, indicates that the latency and duration of ERs were not affected by electrical stimulation training.

In summary, the results demonstrated that polysynaptic electrical conduction in sensorimotor circuits was restored by only 10- to 20-Hz EEMS. Thus, the range of 10- to 20-Hz EEMS was identified as effective for electrical stimulation, whereas 1-Hz, 5-Hz, 30-Hz, and 40-Hz EEMS were ineffective.

### Restoration of motor function of hind limbs in SCI mice by 10- to 20-Hz EEMS

To further evaluate the effect of EEMS, EES, MS and EEShc on motor function in SCI mice, a basso mouse scale (BMS) score was calculated to evaluate overall locomotor performance under different experimental conditions. Mice were observed in an open field and subjected to BMS scoring at 0, 1, 3, 7, 14, 21, and 28 days after SCI (Fig. S6a). Compared with the untrained group, the motor function of the hind limbs was significantly restored by 10- to 20-Hz EEMS (Fig. S6b). In contrast, mice in the 1- to 40-Hz EES, 1- to 40-Hz MS, and 1- to 40-Hz EEShc groups showed no improvement in hind limb locomotor function as compared with mice in the untrained group (Fig. S6c–e). We further assessed hindlimb motor function in the 10- to 20-Hz EES, 10- to 20-Hz MS, 10- to 20-Hz EEShc, and 10- to 20-Hz EEMS groups using BMS scoring. Compared with the 10- to 20-Hz EES, 10- to 20-Hz MS, and 10- to 20-Hz EEShc groups, motor function recovery of the hind limbs was increased by 10- to 20-Hz EEMS (Fig. S6f). Under other frequencies of electrical stimulation, however, as compared with the EES, MS, and EEShc groups, the motor function of the hind limbs of mice was not improved in the EEMS group (Fig. S6b–e). Thus, these results indicated that the dual stimulation was necessary for the improvement of locomotor function after SCI, and functional motor recovery was effectively promoted by 10- to 20-Hz EEMS instead of EES or MS after SCI.

To detect EMG bursts during rhythmic movement of mice, we designed a body weight-supporting device combined with a treadmill so that the hind limbs of the mice just touched the track of the treadmill. The treadmill operated at a speed of 2 m·min⁻¹, and EMG signals from the TA muscles were observed when the mice were moving (Fig. S6g). For these experiments, we marked the hip, knee,

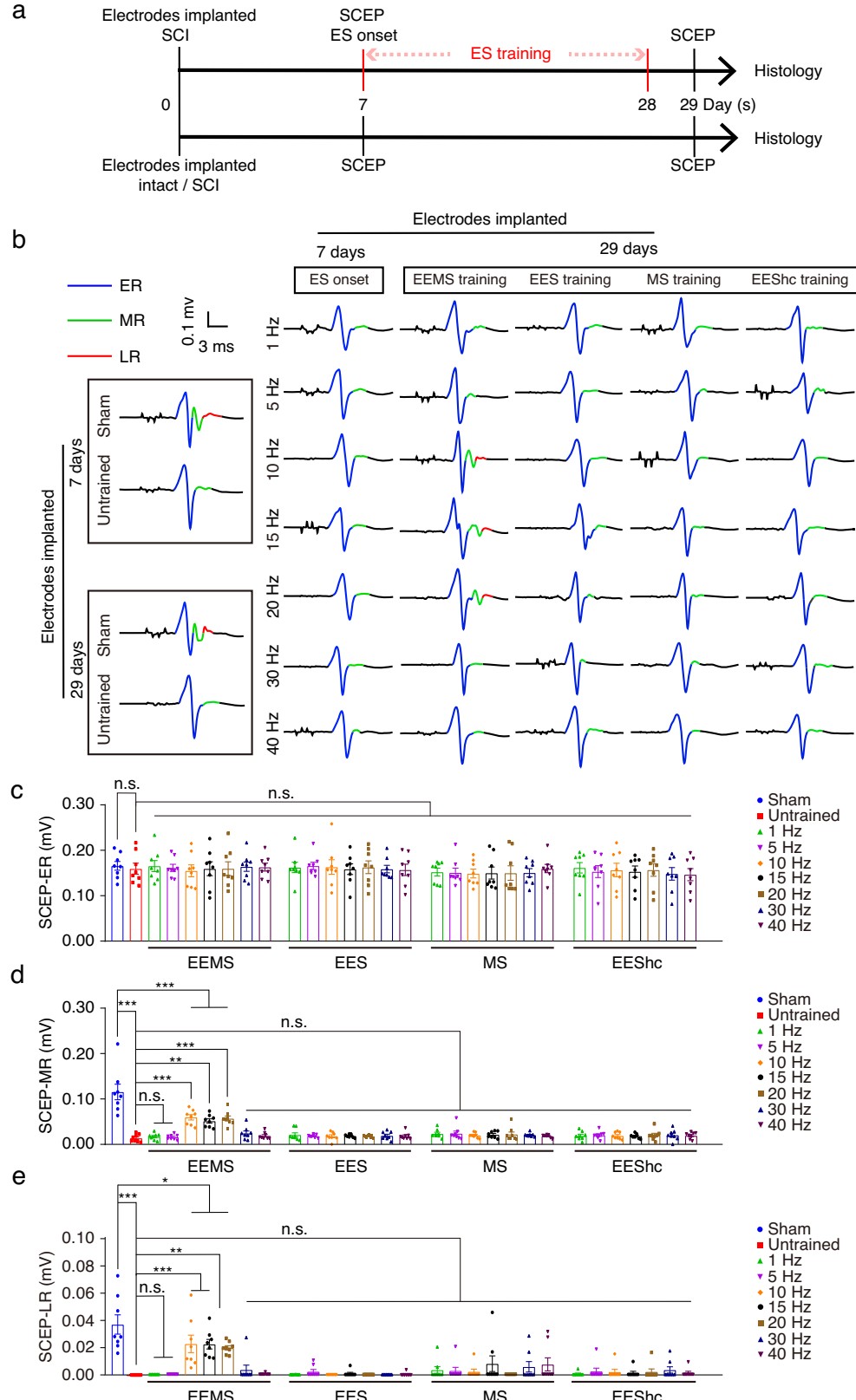

**Fig. 2 | SCEPs recorded in the TA muscle of SCI mice after EEMS. a** Experimental scheme. **b** SCEPs were recorded for mice in the sham, untrained, and ES groups 7 days and 29 days after electrode implantation. The waveform in the figure is the average of 50 SCEPs. **c**–**e** Histogram reporting the amplitudes of ERs (**c**), MRs (**d**), and LRs (**e**) of SCEPs in the sham, untrained, 1- to 40-Hz EEMS, 1- to 40-Hz EES, 1- to 40-Hz MS, and 1- to 40-Hz EEShc groups 29 days after electrode implantation ($n = 8$ mice per group). Data represent the mean ± SEM; ns: no statistically significant difference, *$p < 0.05$, **$p < 0.01$, ***$p < 0.001$, one-way ANOVA followed by the Bonferroni post hoc test.

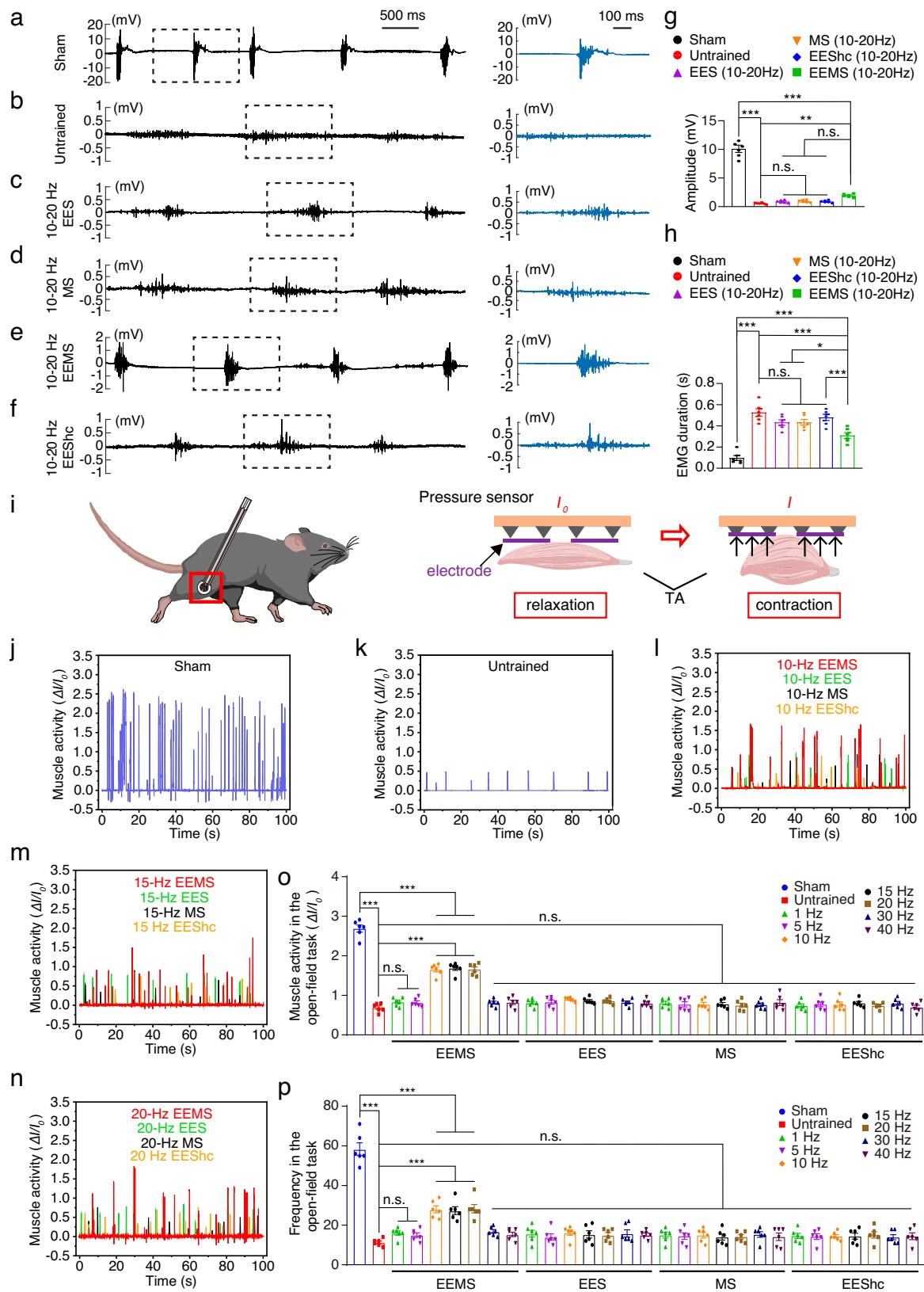

ankle joint, and sole of the foot of each mouse hind limb to determine movement trajectories (Fig. S6h). For mice of the untrained group, the ankle joint of the hind limb was barely flexed, failed to regain normal gait (Fig. S6i). However, the extent of dorsiflexion of the hind limb ankle joint was greater in the 10- to 20-Hz EEMS groups relative to that in the 10- to 20-Hz EES and MS groups and was more similar to the sham group, while the hind limb gait was more similar to that of the sham group (Fig. S6i, Supplementary movie 1). Next, we recorded the EMG bursts in the TA of the mice during treadmill exercise (Fig. 3a–f). With the sham group (10.09 ± 0.636 mV, 5 steps) mice serving as a control, we detected the amplitude of EMG bursts in the TA muscle of mice in the untrained (0.564 ± 0.054 mV, 3 steps), 10- to 20-Hz EES

**Fig. 3 | Evaluation of function of TA in SCI mice after EEMS, EES, MS, or EEShc.** **a–f** Left (black), the graphs show surface EMGs for the TA muscles in the sham (**a**), untrained (**b**), 10–20 Hz EES (**c**), 10–20 Hz MS (**d**), 10–20 Hz EEMS (**e**), and 10–20 Hz EEShc (**f**) groups over a period 5 s, selected from a 30-second recording. Right (blue), enlarged view of the EMGs burst in the dashed box on the left. Mice from the sham group exhibited five steps within 5 s, while both untrained and trained groups displayed three to four steps. **g** The maximum amplitude of surface EMG bursts continuously for 5 s in the sham, untrained, 10–20 Hz EES, 10–20 Hz MS, 10–20 Hz EEShc, and 10–20 Hz EEMS groups (*n* = 6 mice per group). **h** Average duration of a single TA burst in 5 s in the sham, untrained, 10–20 Hz EES, 10–20 Hz MS, 10–20 Hz EEShc, and 10–20 Hz EEMS groups (*n* = 6 mice per group). **i** Schematic diagram

showing the experimental approach for measuring muscle strength. When the muscle was relaxed, the current through the sensor was denoted as $I_0$; when the muscle contracted, the current through the sensor was $I$. Schematic was created with BioRender.com. **j, k** TA muscle contraction and its corresponding relative current in mice of the sham (**j**) and untrained (**k**) groups. **l–n** TA muscle contraction curve for mice in the 10- to 20- HzEEMS (red), EES (green), MS (black), and EEShc (orange) groups. **o, p** The intensity (**o**) and frequency (**p**) of TA muscle contraction in mice in the sham, untrained, and 1- to 40-Hz EEMS, 1- to 40-Hz EES, 1- to 40-Hz MS, and 1- to 40-Hz EEShc groups (*n* = 6 mice per group). Data represent the mean ± SEM; ns: no statistically significant difference, *$p < 0.05$, **$p < 0.01$, ***$p < 0.001$, one-way ANOVA followed by the Bonferroni post hoc test.

(0.883 ± 0.104 mV, 3 steps), 10- to 20-Hz MS (0.896 ± 0.094 mV, 3 steps), and 10- to 20-Hz EEShc (0.880 ± 0.094 mV, 3 steps) groups (Fig. 3g). The amplitude of TA EMG bursts in the 10- to 20-Hz EEMS (1.928 ± 0.120 mV, 4 steps) groups was significantly higher than that of the untrained (Fig. 3g). The duration of a single TA burst was also measured and was significantly longer in the untrained (0.525 ± 0.036 s) group than that in the sham (0.097 ± 0.023 s) group (Fig. 3h). However, the duration of a single TA burst recorded in the 10- to 20-Hz EEMS (0.308 ± 0.028 s) groups was significantly shorter than that in the untrained, 1- to 20-Hz EES (0.432 ± 0.026 s), 1- to 20-Hz MS (0.434 ± 0.026 s), and 1- to 20-Hz EEShc (0.480 ± 0.029 s) groups (Fig. 3h). Together, 10- to 20-Hz EEMS enhanced the electrical signal emission of spinal cord neurons to peripheral muscles after SCI. In addition, we increased the stimulation interval between EES and MS in EEMS. The amplitude and duration of a single TA burst after increase in the stimulation interval were consistent with those before the stimulation interval increased (Fig. S7).

To accurately and objectively evaluate the effect of EEMS on TA muscle contractility after SCI, we developed a pressure sensor that does not affect locomotion in intact mice[39]. Here, the intensity and frequency of TA muscle contractions were measured using pressure sensors while the mice freely moved in an open field (Fig. 3i). In the sham, untrained, 1- to 40-Hz EEMS, 1- to 40-Hz EES, 1- to 40-Hz MS, and 1- to 40-Hz EEShc groups, TA muscle contraction was detected by the pressure sensor (Fig. 3j–n, Fig. S6j–m). SCI dramatically decreased the intensity and frequency of TA muscle contraction relative to that of sham group mice (Fig. 3o, p). As compared with the untrained group, both the frequency and the intensity of activity increased significantly in the 10- to 20-Hz EEMS groups (Fig. 3o, p). In contrast, TA muscle contractility was not improved in the 1- to 40-Hz EES, 1- to 40-Hz MS, and 1- to 40-Hz EEShc groups as compared with mice in the untrained group (Fig. 3o, p). These results indicated that TA muscle contractility was recovered by 10- to 20-Hz EEMS, but not by EES or MS after SCI.

In summary, in vivo functional analyses provided evidences that motor function of the hind limbs was restored by 10- to 20-Hz EEMS but not by EES or MS after SCI.

### Reconstruction of hind limb neuromuscular junctions (NMJs) in SCI mice by 10- to 20-Hz EEMS

To verify that the spinal sensorimotor circuit had been reconstructed as a result of EEMS, we examined each component of the circuit. We first observed the TA muscle fibers and NMJs that formed between the motor nerve and TA muscle in C57BL/6J mice (Fig. 4a). NMJs were labeled by immunofluorescence staining with α-bungarotoxin (α-BTX) to label the acetylcholine receptor (AChR) and with antibodies to neurofilament/synapsin (NF/Syn) to label axon terminals (Fig. 4b). Hematoxylin and eosin staining of muscle tissue was performed to quantify muscle atrophy after SCI (Fig. 4b). The cross-sectional area of muscle fibers was significantly smaller in the untrained and ES onset groups than in the sham group (Fig. 4c). In addition, compared with the ES onset group, the cross-sectional area of the muscle fiber was significantly smaller in the untrained group, indicating muscle atrophy

at 4 weeks after SCI (Fig. 4c). The cross-sectional area of the muscle fibers was significantly larger in the 10- to 20-Hz EEMS groups compared with the untrained group, indicating that 10- to 20-Hz EEMS could effectively rescue muscle atrophy after SCI (Fig. 4c). Next, we compared the NMJs of different EEMS groups. We randomly selected six TA muscle endplates in the field of view and calculated the percentage of fully innervated, partially innervated, and denervated endplates (fully innervated, syn and α-BTX overlap; partial innervation, syn and α-BTX partially overlap; denervated, syn and α-BTX do not overlap). There were fewer endplates in the TA muscles that were fully innervated in the untrained (10.2%) and ES onset (20%) groups than in the sham group (92.7%) (Fig. 4d). Notably, fully innervated endplates were more numerous in the 10- to 20-Hz EEMS groups (46.8%, 56.6%, 43.2%) (Fig. 4d). Together, these results suggested that NMJ deficiency in SCI mice could be partially restored by 10- to 20-Hz EEMS, with consequent reduction of muscle atrophy.

### Reestablishment of hind limb spinal sensory-motor connectivity in SCI mice by 10- to 20-Hz EEMS

We next measured the projection of sensory neuron axons to the spinal cord. To trace proprioceptive axon terminals[40] (PTs) in the spinal cord, Cholera toxin B (CTB) was injected into the TA muscle of *Lbx1^Cre* mice that had been trained with electrical stimulation for 1 week. The mice were euthanized after 2 weeks of electrical stimulation training (Fig. 5a). Three-dimensional (3D) reconstruction was performed to quantify the labeled PTs in the spinal cord (Fig. 5b, Fig. S8). Compared with the sham group, the absolute value of the number of PTs in the ES onset and untrained groups was higher, whereas the number of PTs did not differ significantly among the three groups (sham group versus ES onset group, *p* = 0.4597; sham group versus untrained group, *p* = 0.0598) (Fig. 5c). In contrast, PTs were significantly more numerous in the 10- to 20-Hz EEMS groups compared with the untrained group (Fig. 5c). These results suggested that 10- to 20- Hz EEMS promoted the axons of the proprioceptive neurons innervate the spinal neurons.

We next explored the specificity of monosynaptic sensory inputs into motoneurons supplying the TA muscles. Before spinal cord electrodes were implanted, AAV-DIO-mCherry was injected into the intermediate area of gray matter at the L2−L4 segment of *Lbx1^cre* mice to label the interneurons (Fig. 5a). 1 week after electrical stimulation, we then tracked the central projection of sensory neuron afferents by injecting AAV-GFP into the DRG at the L2−L4 segment in *Lbx1^cre* mice (Fig. 5a). The appearance of DRG fibers extending into the ventral spinal cord, where motoneurons reside, was observed in the 10- to 20- Hz EEMS groups (Fig. 5d). In order to further observe the connection between DRG fibers and interneurons, we assessed the colocalization of DRG fibers and interneurons, performed 3D modeling, and measured both the area and number of DRG fibers connected with interneurons (Fig. 5e). Both the area and number of connections between sensory afferent fibers and interneurons were significantly reduced in the untrained and ES onset groups compared with the sham group, indicating the sensory-interneuron connections were destroyed one

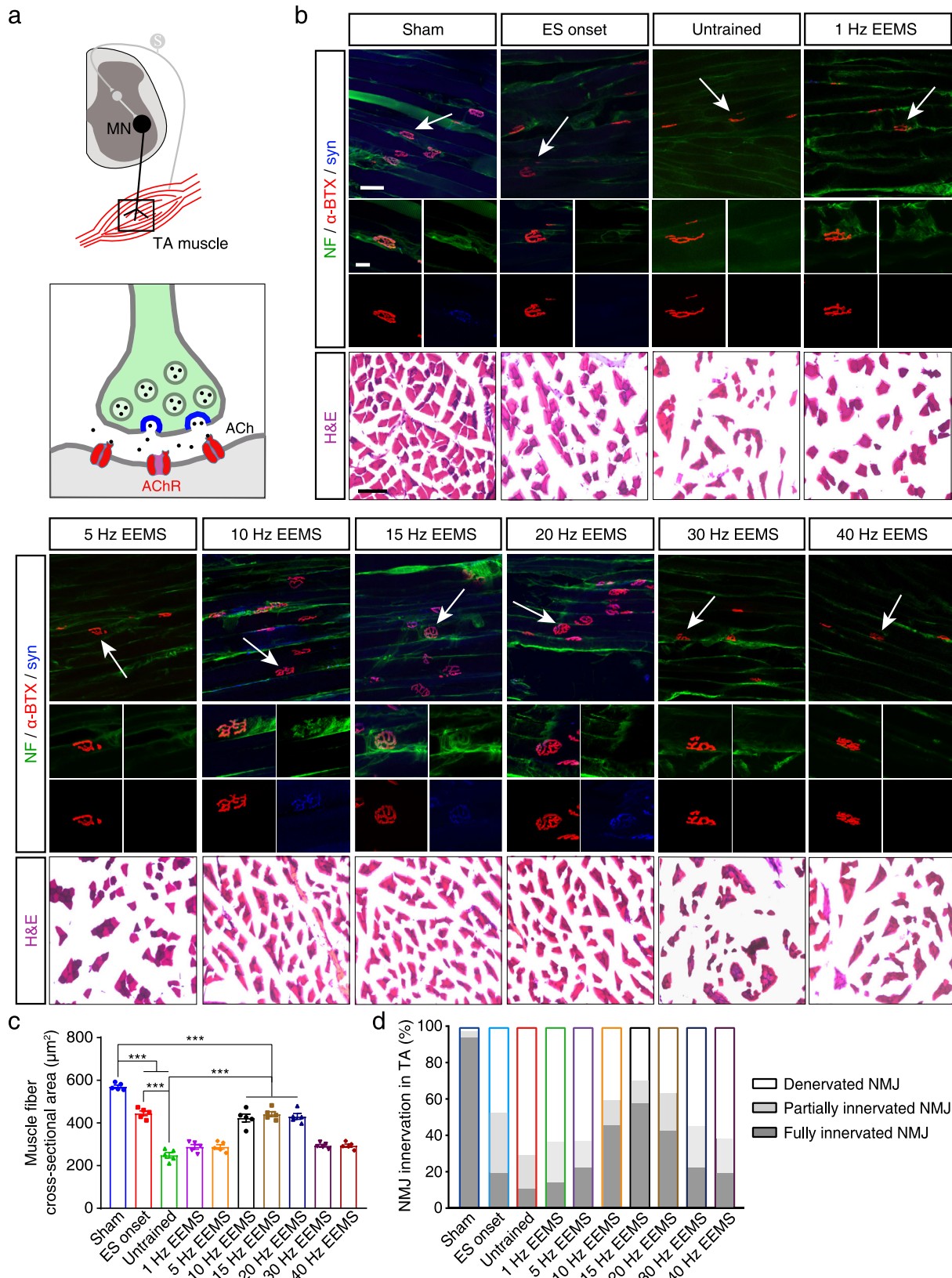

week after SCI (Fig. 5f, g). Compared with the untrained group, the area and number of DRG fibers and interneuron connections were significantly increased in the 10- to 20-Hz EEMS groups (Fig. 5f, g). These results indicated that the sensory-interneuron connections were rebuilt in response to 10- to 20-Hz EEMS.

After 1 week of electrical stimulation, Texas Red-dextran amine (TRDA) was injected into the TA muscle for retrograde tracing of motoneurons (Fig. 5a). We assessed the colocalization of DRG fibers and motoneurons, performed 3D modeling, and measured the area and number of DRG fibers connected with motoneurons (Fig. 5h).

**Fig. 4 | Morphology and neural innervation of the TA muscles in SCI mice after EEMS. a** Schematic diagram of the NMJ. Bottom: green indicates motoneuron axons, blue indicates presynaptic vesicle proteins, and red indicates the acetylcholine receptor (AChR). Schematic was created with BioRender.com. **b** Top, antibodies against NF (green), α-BTX (red), and syn (blue) were used to label NMJs in the TA muscle from mice in the sham, ES onset, untrained, and 1- to 40-Hz EEMS groups. In each image, the white arrow indicates a single NMJ. Scale bar, 50 μm. Middle, higher-magnification images of the NMJ indicated by each white arrow. Scale bar, 20 μm. Bottom, hematoxylin and eosin (H&E) staining of the TA muscle from mice in the sham, ES onset, untrained, and 1- to 40-Hz EEMS groups. Scale bar, 50 μm. **c** Cross-sectional area of muscle fibers in the TA muscle from mice in the sham, ES onset, untrained, and 1- to 40-Hz EEMS groups ($n = 5$ mice per group). **d** Percentage of denervation, partial innervation, and complete innervation of the TA muscles from mice in the sham, ES onset, untrained, and 1- to 40-Hz EEMS groups ($n = 5$ mice per group). Data represent the mean ± SEM; ***$p < 0.001$, one-way ANOVA followed by the Bonferroni post hoc test (**c**).

Compared with the sham group, the area and the number of connections between sensory afferent fibers and motoneurons were significantly reduced in the untrained and ES onset groups, indicating the sensory-motor connections were destroyed one week after SCI (Fig. 5i, j). As compared with the untrained group, the area and number of DRG fibers and motoneuron connections were significantly increased in the 10- to 20-Hz EEMS groups (Fig. 5i, j). These results indicated that the sensory-motor connections were reconstructed in response to 10- to 20-Hz EEMS.

### Morphological characterization of motoneurons that were restored by 10- to 20-Hz EEMS

After 1 week of electrical stimulation, AAV2/2Retro-hSyn-EGFP-WPRE-pA was injected into the TA muscle of C57BL/6J mice to trace the retrograde projection of the corresponding motoneurons in the spinal cord (Fig. 6a, b). The number of motoneurons in the untrained, ES onset, and 1-40 Hz EEMS groups did not differ from that of the sham group (Fig. 6c). 3D reconstruction was applied to analyze the dendrites of labeled motoneurons (Fig. 6b and Fig. S9a). The number of dendritic terminal points of motoneurons was counted, and the results showed that SCI led to a decrease in dendrites branches of motoneurons compared with the sham group (Fig. 6d). On the contrary, the number of dendritic terminal points of motoneurons was significantly increased in the 10- to 20- Hz EEMS groups compared with the untrained group (Fig. 6d). We subsequently found that the volume of distal dendrites of motoneurons was significantly reduced in the untrained and ES onset groups compared with the sham group, indicating that SCI led to dendrite atrophy (Fig. 6e). In contrast, dendrite atrophy was effectively reversed in the 10- to 20-Hz EEMS groups, and the volume of motoneuron dendrites in the 10- to 20- Hz EEMS groups was almost recovered to the preinjury level (Fig. 6e). These results demonstrated that the application of 10- to 20-Hz EEMS could effectively regenerate the dendrite branching structures of motoneurons after SCI.

To confirm the reassembly of the spinal sensorimotor circuit in response to 10- to 20-Hz EEMS after SCI, AAV2/2Retro-hSyn-EGFP-WPRE-pA was injected into the TA muscle of C57BL/6J mice to trace the retrograde projection of the corresponding spinal motoneurons in the L2−L4 spinal cord (Fig. 6a). At 2 weeks post-injection, the L1−L6 spinal cord was collected and made transparent by using a tissue optical clearing technique[41], and whole-mount immunostaining was carried out to visualize the labeled motoneurons and branches (Fig. 6f, g and Fig. S9b). 3D reconstruction of the whole-mount staining images was conducted to comprehensively observe motoneuron dendrites and quantify their complexity (Fig. 6g and Fig. S9b). Compared with the sham group, the average length of dendrites per motoneuron was significantly reduced in the untrained and ES onset groups, indicating that the complexity of motoneuron branches and the possibility of making connections with interneurons was reduced one week after SCI (Fig. 6h). The average length of dendrites per motoneuron was significantly increased relative to the untrained mice in the 10- to 20-Hz EEMS groups (Fig. 6h). These results indicated that the application of 10- to 20-Hz EEMS significantly promoted the branching of motoneuron dendrites and increased the chance of synaptic reformation between interneurons and motoneurons after SCI.

### Restoration of premotor interneuron input into motoneurons in SCI mice by 10- to 20-Hz EEMS

Premotor interneurons responsible for relaying signals are essential for the movement of mouse hindlimbs[42]. Therefore, we were curious whether EEMS affects the communication between premotor interneurons and motoneurons in SCI mice. Before spinal cord electrodes were implanted, an anterograde AAV vector carrying rabies virus glycoprotein (RVG) and the DIO promoter (AAV-DIO-RVG) was injected into the ventral spinal cord of $ChAT^{cre}$ mice, and RVG was expressed in motoneurons; further, EGFP-labeled and G-deleted rabies virus (RV-N2C (G)-ΔG-EGFP) was injected into the TA muscle to trace the retrograde movement of the corresponding premotor interneurons (Fig. 7a). AAV2/2Retro-nEf1α-mCherry-WPRE-pA was injected into the TA muscle to retrogradely trace motoneurons (Fig. 7a). We found that the connections between premotor interneurons and motoneurons were distributed mainly in lamina VII of the Rexed's laminae (Fig. 7b). Axons of premotor interneurons and the dendrites of motoneurons colocalized in lamina VII, and 3D modeling was performed (Fig. 7c, Fig. S10a). The area and number of synaptic connections between premotor interneurons and motoneurons were significantly reduced in the untrained and ES onset groups as compared with the sham group (Fig. 7d, e). Compared with the untrained group, the number and area of synaptic connections between premotor interneurons and motoneurons were significantly increased in the 10- to 20-Hz EEMS groups (Fig. 7d, e). These results indicated that 10- to 20-Hz EEMS effectively promoted the reinnervation of premotor interneurons to motoneurons after SCI.

### Reconstruction of glutamatergic synaptic connections between neurons in SCI mice by 10- to 20-Hz EEMS

We detected changes in the SCEP waveforms 1 week after SCI and found that there was no significant change in ER amplitude compared with the sham group, whereas the MR amplitude was significantly decreased and the LR amplitude was absent (Fig. S10b). These results showed that the sensorimotor circuit of the spinal cord was dysfunctional after SCI, which led to a blockade of polysynaptic electrical conduction.

To identify the receptors involved in the sensorimotor circuits blocked by SCI, we performed a series of pharmacological and electrophysiological tests. First, SCEPs were detected in intact mice; then, in the same mice, we blocked glutamate binding to N-methyl-D-aspartic acid (NMDA) receptors through local intrathecal delivery of D (−)-2-amino-5-phosphonopentanoic acid (AP5), an NMDA receptor antagonist at spinal segments L5/L6 (Fig. S10c). Similarly, we administered intrathecal pentylenetetrazole (PTZ) to other mice to prevent γ-aminobutyric acid (GABA) from binding to GABA-A receptors (Fig. S10d). There was no change in ER, MR, or LR amplitudes after the use of AP5 or PTZ (Fig. S10c, d). In contrast, we prevented glutamate from binding to AMPA/kainate receptors through local intrathecal delivery of the AMPA/kainate receptor antagonist 6-cyano-7-nitroquinoxaline-2,3-dione (CNQX) at spinal segments L5/L6, and the results showed that the amplitude for each of MR and LR of SCEPs was significantly decreased or absent (Fig. S10e). Taken together, these results suggested that the glutamatergic synaptic connections between neurons

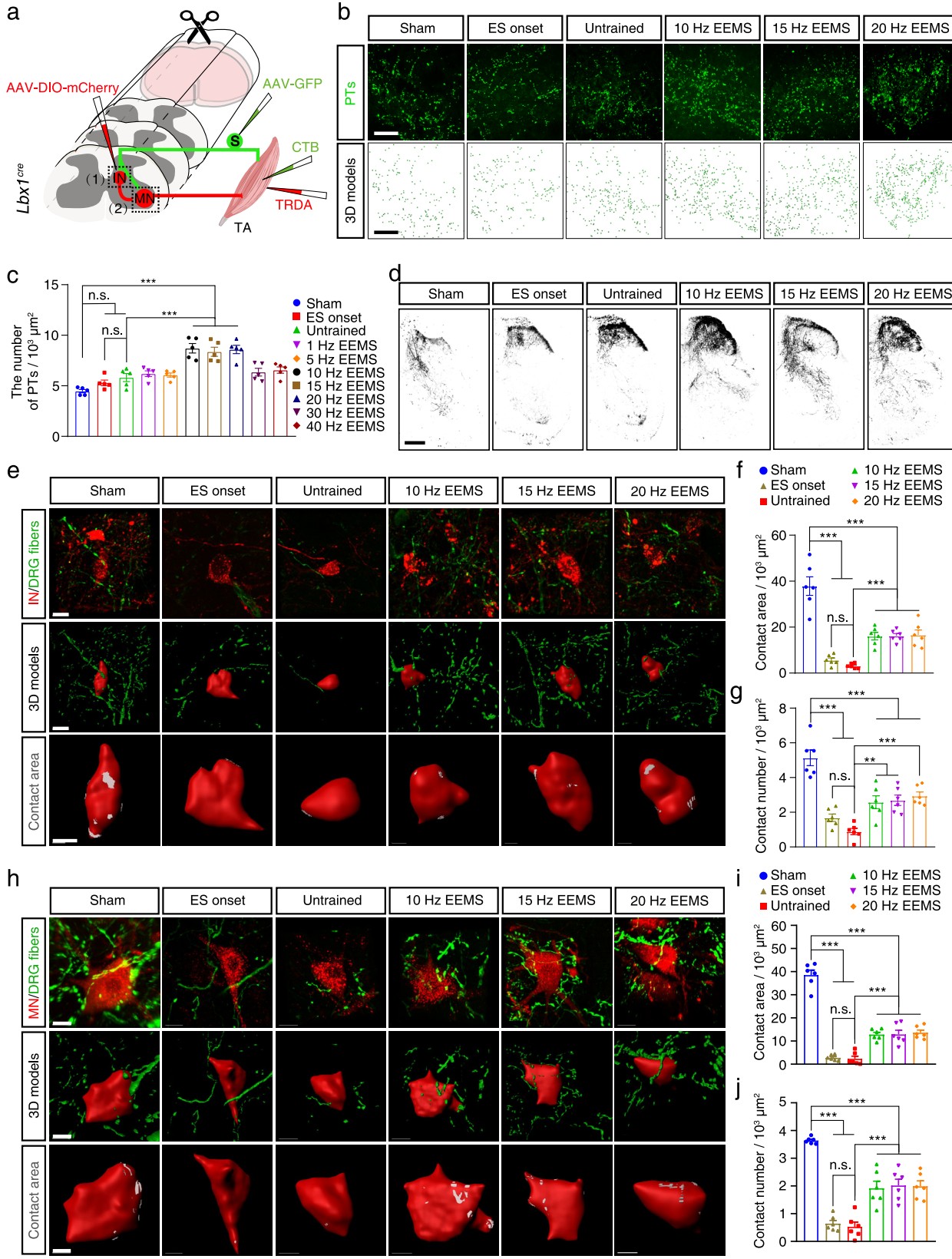

in sensorimotor neural circuits—specifically those connections involving AMPA/kainate receptors—were disrupted after SCI.

We then injected Cholera toxin B subunit into the TA muscle of C57BL/6J mice after SCI to retrogradely trace the corresponding motoneurons in the spinal cord (Fig. 7f). Motoneurons were stained for vesicular glutamate transporter 1 (vGluT1) in the same mice to label the

glutamatergic axon terminals of premotor neurons (Fig. 7g, Fig. S10f). 3D reconstruction of staining images was performed to quantify the vGluT1+ boutons on motoneurons in each group (Fig. 7g, Fig. S10f). Compared with the sham group, the average number of vGluT1+ boutons on each motoneuron was significantly reduced in the untrained and ES onset groups (Fig. 7h). This indicated that glutamatergic

**Fig. 5 | Sensory-motor connectivity in SCI mice after EEMS. a** Diagram illustrating the tracing of PTs (1), interneurons (1), motoneurons (2), and DRG fibers (1 and 2) in *Lbx1^cre* mice. Schematic was created with BioRender.com. **b** The PTs (green) in the spinal cord (upper images) were labeled in the sham, ES onset, untrained, and 10- to 20-Hz EEMS groups, and modelled in 3D (lower images). Scale bar, 50 μm. **c** Number of PTs in the sham, ES onset, untrained, and 1- to 40-Hz EEMS groups (*n* = 5 mice per group). **d** DRG fibers in the spinal cord were labeled in the sham, ES onset, untrained, and 10- to 20-Hz EEMS groups. Scale bar, 200 μm. The experiment was repeated 3 times independently with similar results. **e** Upper, colocalization of DRG fibers and interneurons in the sham, ES onset, untrained, and 10- to 20-Hz EEMS groups. Scale bar, 15 μm. Middle, 3D reconstruction of DRG fibers and interneurons. Scale bar, 15 μm. Lower, highly magnified images of interneurons. Gray region represents the region of connection between DRG fibers and interneurons. Scale bar, 10 μm. **f, g** Area (**f**) and number (**g**) of contacts of DRG fibers per 1000 μm² of the interneuron surface area in the sham, ES onset, untrained, and 10- to 20-Hz EEMS groups (*n* = 6 mice per group). **h** Upper, colocalization of DRG fibers and motoneurons in the sham, ES onset, untrained, and 10- to 20-Hz EEMS groups. Scale bar, 15 μm. Middle, 3D reconstruction of DRG fibers and motoneurons. Scale bar, 15 μm. Lower, highly magnified image of motoneurons. Gray region represents the region of connection between DRG fibers and motoneurons. Scale bar, 10 μm. **i, j** Area (**i**) and number (**j**) of contacts of DRG fibers per 1000 μm² of the motoneuron surface in the sham, ES onset, untrained, and 10- to 20-Hz EEMS groups (*n* = 6 mice per group). Data represent the mean ± SEM; ns: no statistically significant difference, **$p < 0.01$, ***$p < 0.001$, one-way ANOVA followed by the Bonferroni post hoc test.

synaptic connections between premotor neurons and motoneurons were disrupted one week after SCI. The average number of vGluT1⁺ boutons on each motoneuron was restored significantly in the 10- to 20-Hz EEMS groups compared with the untrained group (Fig. 7h). Moreover, compared with the sham group, the average number of vGluT1⁺ boutons on each motoneuron in the 10- to 20-Hz EEMS groups was almost recovered to the preinjury level (Fig. 7h). These results indicated that glutamatergic synapses could reform between premotor neurons and motoneurons after SCI.

The MRs and LRs of SCEP in SCI mice were restored by 10- to 20-Hz EEMS (Fig. 2b). To verify that the recovery of LRs in response to 10- to 20-Hz EEMS after SCI was due to the reconstruction of glutamatergic synapses between premotor neurons and motoneurons, SCEPs were recorded in mice in the 10- to 20-Hz EEMS groups after intrathecal injection of CNQX (Fig. S10g–j). The ER amplitude did not change after the application of CNQX, but the amplitudes of each of the MR and LR were significantly decreased or absent (Fig. S10h–j). Together, these results demonstrated that 10- to 20-Hz EEMS consistently promoted the reformation of synapses between glutamatergic neurons and motoneurons. This evidence also indicated that the spinal sensorimotor circuit was rebuilt morphologically in response to 10- to 20-Hz EEMS after SCI.

**Activation of motoneurons by premotor neuron input in SCI mice after 10- to 20-Hz EEMS**

To determine whether physiologically functional synaptic connections between premotor neurons and motoneurons in the sensorimotor circuit could be reestablished by 10- to 20-Hz EEMS, Ca²⁺ signals from TA motoneurons of awake mice were recorded using fiber photometry (Fig. 8a). AAV2/2Retro-EF1a-GCaMp6m-WPRE-hGH-pA was injected into the TA muscle, and the Ca²⁺ indicator GCaMP6 was expressed in the corresponding motoneurons of the TA muscle of C57BL/6J mice. We implanted a small optical fiber (250 μm in diameter) with its tip in the motoneuron pool to record changes in GCaMP fluorescence (Fig. 8a). We first examined GCaMP signals when the sham group mice moved. Hindlimb ankle flexion reliably increased GCaMP fluorescence during movement of these mice, indicating that the motoneurons innervating the TA muscle were activated (Fig. 8b). The enhancement of GCaMP fluorescence in motoneurons indicated that premotor neurons transmitted neural signals to motoneurons and activated them. Ca²⁺ signals were recorded in the motoneurons of mice under different experimental conditions (Fig. 8c–l). SCI resulted in a significant reduction in Ca²⁺ signal intensity as compared with the sham group (Fig. 8m). In contrast, the GCaMP signals in motoneurons were significantly more intense in the 10- to 20-Hz EEMS mice than in untrained mice (Fig. 8m). The area under curve (AUC) of the motoneuron Ca²⁺ signaling curve was also measured. The average AUC decreased significantly after SCI (Fig. 8n). The average AUC was significantly increased by 10- to 20-Hz EEMS as compared with the untrained group (Fig. 8n), and the average AUC in the 10- to 20-Hz EEMS groups almost recovered to the preinjury state (Fig. 8n). In

summary, these findings indicated that the premotor inputs of motoneurons in the sensorimotor neural circuits were disrupted by SCI, and the functional synaptic connections between premotor neurons and motoneurons were reconstructed by 10- to 20-Hz EEMS.

**Changes in the molecular network in spinal motoneurons activated by 10- to 20-Hz EEMS**

To gain mechanistic insights into the reassembly of the spinal sensorimotor circuits in response to 10- to 20-Hz EEMS, we performed gene expression profiling analysis in motoneurons. At preinjury, TRDA was injected into the TA muscle of C57BL/6J mice to retrogradely trace the corresponding motoneurons in the spinal cord, and SCI mice were sacrificed after 3 weeks of untrained, effective EEMS (10- to 20- Hz), EES (10- to 20- Hz), MS (10- to 20- Hz), and ineffective EEMS (1-Hz, 5-Hz, 30-Hz, and 40-Hz). TRDA-labeled spinal motoneurons were isolated using the glass microtube aspiration method[43] under a fluorescence microscope (Fig. 9a). mRNA from the purified spinal motoneurons was then used for library preparation and sequencing (Fig. 9a). The effective EEMS group identified 777, 101, 569 and 124 genes transcriptome changes compared to the untrained, EES, MS, or ineffective EEMS groups, respectively (fold change ≥ 1.5, false discovery (FDR) ≤ 0.1) (Fig. S11). Recent studies have revealed that genes related to Neuron axon development, Synaptic function, Inflammation and apoptosis are involved in the regulation of neuronal plasticity[44]. Furthermore, our study shows that specific frequency of EEMS enhances the plasticity of motoneurons. These results suggest that genes related to Neuron axon development, Synaptic function, Inflammation and apoptosis may be involved in the regulation of motoneuron plasticity. Therefore, we defined three functional terms: Neuron axon development, Synaptic function, Inflammation and apoptosis. The differential genes regulated by effective EEMS were found to be associated with events pertaining to the development of neuronal axons, synaptic function, inflammation, and apoptosis when compared to both the untrained group and the MS group (Fig. 9b, c). The differential genes regulated by effective EEMS were only specifically associated with events pertaining to the development of neuronal axons, in contrast to both the EES and ineffective EEMS groups (Fig. 9b, c). These results suggest that, compared with EES, MS and ineffective EEMS, genes that play unique roles in effective EEMS may be involved in events related to the development of neuronal axons (Fig. 9b, c). Then, we used quantitative real-time PCR to quantify mRNAs transcribed from certain genes in the lumbar spinal cord, and the results showed that the detected gene expression trends were consistent with the results of single-cell transcriptome sequencing (Fig. S12a).

To provide further insight into the mechanisms involved, we also used bioinformatics to assess the potential involvement of genes related to signaling pathways (Fig. S12b). We analyzed functional pathways to determine how the differentially expressed genes could be coordinated to achieve protein-network transduction to mediate sensorimotor circuit remodeling. After parameter correction, there was no significant difference in the signal pathway of effective EEMS

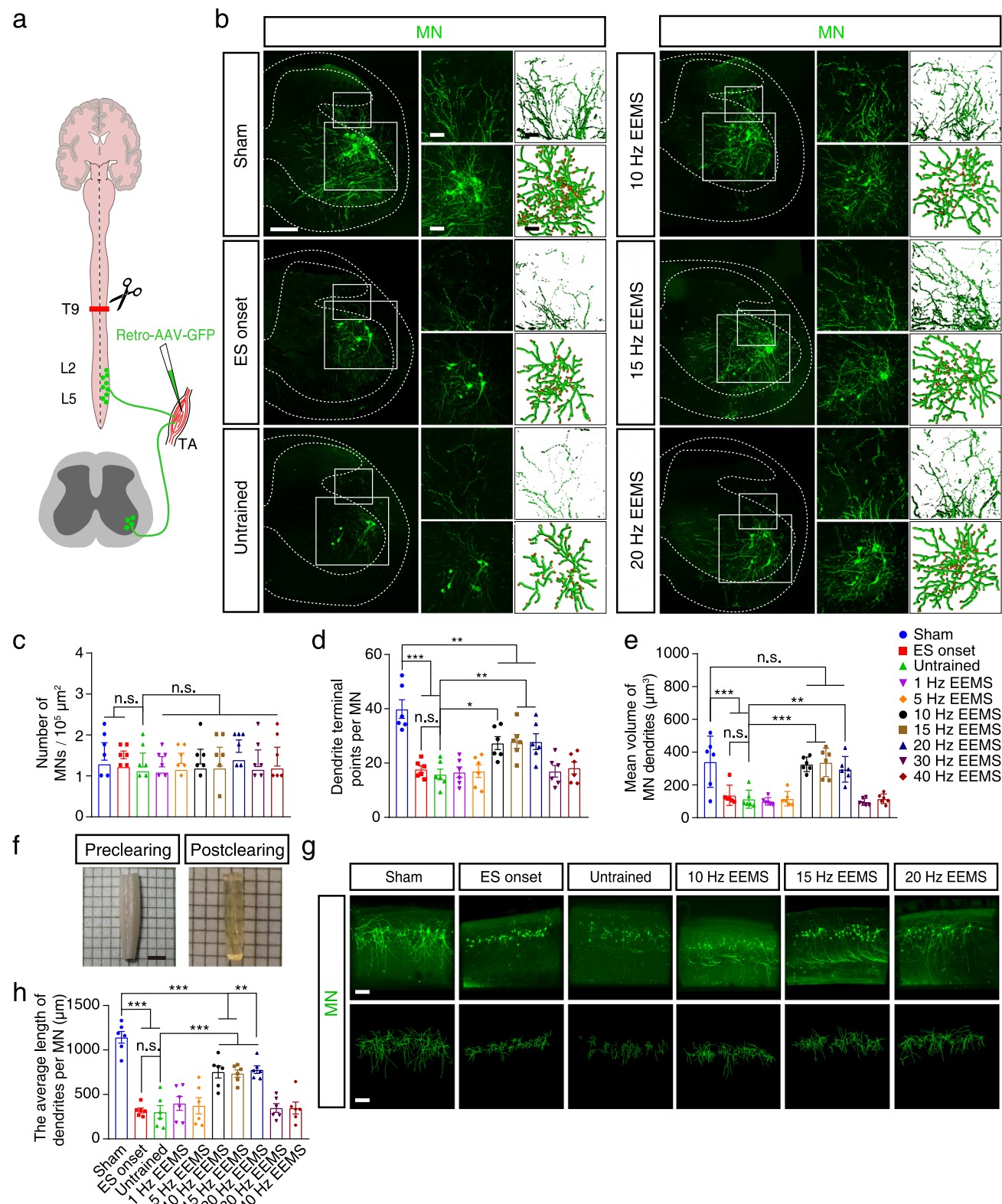

enrichment compared with EES, MS and ineffective EEMS groups. However, compared with the untrained group, effective EEMS significantly activated the PI3K-Akt signaling pathway ($q$-value < 0.05), apoptosis signaling pathway ($q$-value < 0.1), and Gap junction signal path ($q$-value < 0.1). Among them, the PI3K-Akt signaling pathway has been reported to regulate axon regeneration[45,46]. Immunohistochemistry revealed that the expression of Akt did not change in AAV2/2Retro-hSyn-EGFP-WPRE-pA labeled motoneurons of mice in the sham

group, the untrained group, and 10- to 20-Hz EEMS groups (Fig. S12c, d). As compared with the untrained group, however, the expression of phosphorylated Akt (p-Akt) was upregulated in the 10−20 Hz EEMS groups, suggesting that the PI3K-Akt signaling pathways was activated (Fig. S12c, e).

In conclusion, effective EEMS may enhance the plasticity of motor neurons by activating a multitude of genes associated with neuronal axon development, thereby reshaping the sensorimotor neural

**Fig. 6 | Morphological characteristics of spinal motoneurons in SCI mice after EEMS. a** Experimental scheme for labeling motoneurons. Schematic was created with BioRender.com. **b** AAV2/2Retro-hSyn-EGFP-WPRE-pA was injected into the TA muscle to trace the retrograde projection spinal motoneurons. Left, spinal motoneurons in the sham, ES onset, untrained, and 10- to 20-Hz ES groups. Scale bar, 200 μm. Upper right, higher-magnification images and 3D model of the dendrites in the small boxed area. Scale bar, 50 μm. Lower right, partial images and 3D model of the motoneurons in the large boxed area. Scale bar, 100 μm. **c** Number of MNs per 100,000 μm² of the coronal section of the spinal cord in the sham, ES onset, untrained, and 1- to 40-Hz ES groups ($n = 6$ mice per group). **d, e** Mean

number of dendrite terminal points (**d**) and volume of dendrites (**e**) of moto-neurons in the sham, ES onset, untrained, and 1- to 40-Hz EEMS groups ($n = 6$ mice per group). **f** Tissue clearing technique was performed on the lumbar spinal cord. Scale bar, 2 mm. **g** Upper, spatial distribution of motoneurons in the L2−L4 spinal cord in the sham, ES onset, untrained, and 10- to 20-Hz EEMS groups. Scale bar, 200 μm. Lower, 3D models of the motoneurons. Scale bar, 200 μm. **h** Average length of motoneuron dendrites in the L2−L4 spinal cord in the sham, ES onset, untrained, and 10- to 20-Hz EEMS groups ($n = 6$ mice per group). Data represent the mean ± SEM; ns: no statistically significant difference, $*p < 0.05$, $**p < 0.01$, $***p < 0.001$, one-way ANOVA followed by the Bonferroni post hoc test.

circuitry within the spinal cord and subsequently restoring hind limb motor function in mice with spinal cord injuries.

### 10- to 20-Hz EEMS enhances and maintains the flow of neuro-transmitters in the spinal cord of SCI mice

The balance of excitatory and inhibitory neurotransmitters in the spinal cord is crucial for the recovery of motor function after SCI[47]. After incomplete SCI, spinal cord inhibitory interneurons limit functional recovery[48,49]. By increasing the excitability of the spinal cord, individuals with incomplete spinal cord injury can regain autonomic motor function[6]. Therefore, increasing the excitability of the spinal cord and reducing inhibition of inhibitory neurons in the spinal cord may contribute to motor function recovery following a spinal cord injury. To explore the physiological mechanism by which 10- to 20-Hz EEMS at the spinal-muscle interface promotes neural circuit remodeling and functional improvement, we recorded the real-time in vivo flow of Glu and GABA in the spinal cord of mice before, during and after stimulation through neurotransmitter probes. Twenty-nine days after electrode implantation, viruses carrying Glu- and GABA-targeting fluorescent probes were injected into the intermediate-dorsal region of the L4 spinal cord of C57BL/6J mice in the 10−20 Hz EES, 10−20 Hz MS, and 10−20 Hz EEMS groups (Fig. 10a). One week later, the fluorescence changes of Glu and GABA neurotransmitters in the spinal cord of the 10−20 Hz EES, the 10−20 Hz MS, and 10−20 Hz EEMS groups were detected in the awake state. The mice in an awakened state were subjected to a 30 min signal collection period prior to the commencement of stimulation. The stimulation phase lasted for a duration of 90 min, with each stimulation lasting for 15 min followed by a rest period of 10 min. Subsequent to the conclusion of the stimulation phase, signal recording was conducted for an additional 30 min. After recording the fluorescence probe, immunohistochemistry was performed on the L4 segment of the spinal cord to assess viral infection in neurons (Fig. 10a, c, and f).

The results from the glutamate probe demonstrated a significant increase in burst frequency of glutamate flow during and post EEMS (10−20 Hz) compared to pre-EEMS (10−20 Hz), with higher burst frequency observed during stimulation than after (Fig. 10b, d). On the contrary, there was no significant increase in burst frequency of glutamate flow during and post EES (10−20 Hz) and MS (10−20 Hz) compared to before stimulation (Fig. 10d and Fig. S13a, b). Furthermore, it was found that both EEMS (10−20 Hz) and EES (10−20 Hz) induced a higher burst frequency of glutamate flow compared to MS (10−20 Hz) (Fig. 10d and Fig. S13a, b).

The results of the GABA probe demonstrated a significant decrease in burst frequency of GABA flow during and post EEMS (10−20 Hz) compared to pre-EEMS (10−20 Hz) (Fig. 10e, g). Similarly, the burst frequency of GABA flow during EES (10−20 Hz) was significantly lower than pre-EES (10−20 Hz), while no significant difference was observed in the burst frequency of GABA flow after cessation of stimulation compared to pre-EES (10−20 Hz) levels (Fig. 10g and Fig. S13c, d). In addition, there was no statistically significant difference in the frequency of GABA stream bursts during and after MS (10−20 Hz) compared to pre-MS (10−20 Hz) (Fig. 10g and Fig. S13c, d).

In summary, both EEMS (10−20 Hz) and EES (10−20 Hz) exhibited an excitatory effect on spinal neurons surpassing that of MS (10−20 Hz). Notably, following discontinuation of EEMS (10−20 Hz), its excitatory effects on the spinal cord persisted for a certain duration. The EEMS (10−20 Hz) and EES (10−20 Hz) interventions both effectively mitigate the inhibition of spinal neurons, surpassing the efficacy of MS (10−20 Hz). Furthermore, this reduction in inhibition persists for a considerable duration even after discontinuation of EEMS (10−20 Hz). Therefore, only 10−20 Hz EEMS can effectively maintain the excitability of spinal neurons post-stimulation cessation, while concurrently mitigating the inhibition of spinal neurons. Consequently, this facilitates easier activation of the spinal neural circuit and restoration of hind limb motor function, aligning with previously reported findings[6,48,49].

## Discussion

We developed a EEMS system targeting the mice lumbosacral spinal cord and TA muscle, which mimics sensory feedback and feedforward muscle contraction loops. We discovered that dual stimulation at a frequency of 10−20 Hz was optimal for structural and functional reconstruction of spinal sensorimotor circuits. This discovery provides a mechanistic framework for the design of neuromodulation systems based on EES to improve the recovery of motor functions following a neurological disorder.

For our present study, we focused on the sensorimotor reflex circuit, which is composed of the TA muscle, sensory neurons, motor neurons and interneurons. Motor neurons dominate TA muscles and thus directly impact motor function[50,51]. Interneurons are essential for producing reflex and rhythmic motor activity as well as many other activities[42,52,53]. Sensory feedback systems based on sensory neurons constantly monitor the consequences of motor action[42,54]. Studies have shown that EES enhances the excitability of the spinal cord and sensory feedback information is essential for motor recovery below the injury level after SCI[25,27,55]. Based on this, we established a dual electrical neuromodulation system of the spinal cord and muscles to enhance motor output and sensory feedback by combining central and peripheral dual electrical stimulation.

Awake mice trained with electrical stimulation were fixed to a simple fixator in the study. The trunk of the mouse was fixed to the fixator, while its limbs were in a suspended state (no load). As is well known, there are neuronal circuits in the lumbosacral segment of the spinal cord that control motor output, called central pattern generator (CPG)[56,57]. Body weight-supported treadmill training can provide combined sensory cues of tactile, proprioceptive, and kinesthetic for CPG in SCI mice[26,58]. This multimodal sensory input is crucial for activating the basic neural circuits in the spinal cord, adjusting motor patterns, and improving motor performance[58,59]. However, in order to analyze the characteristics of electrical signals that promote the remodeling of spinal cord local neural circuits and the recovery of motor function, spinal T9 complete transection model was used in this study. Complete T9 spinal cord transection leads to the complete interruption of supraspinal input to the lumbosacral neural circuit. Combined with the condition of complete spinal cord transection and

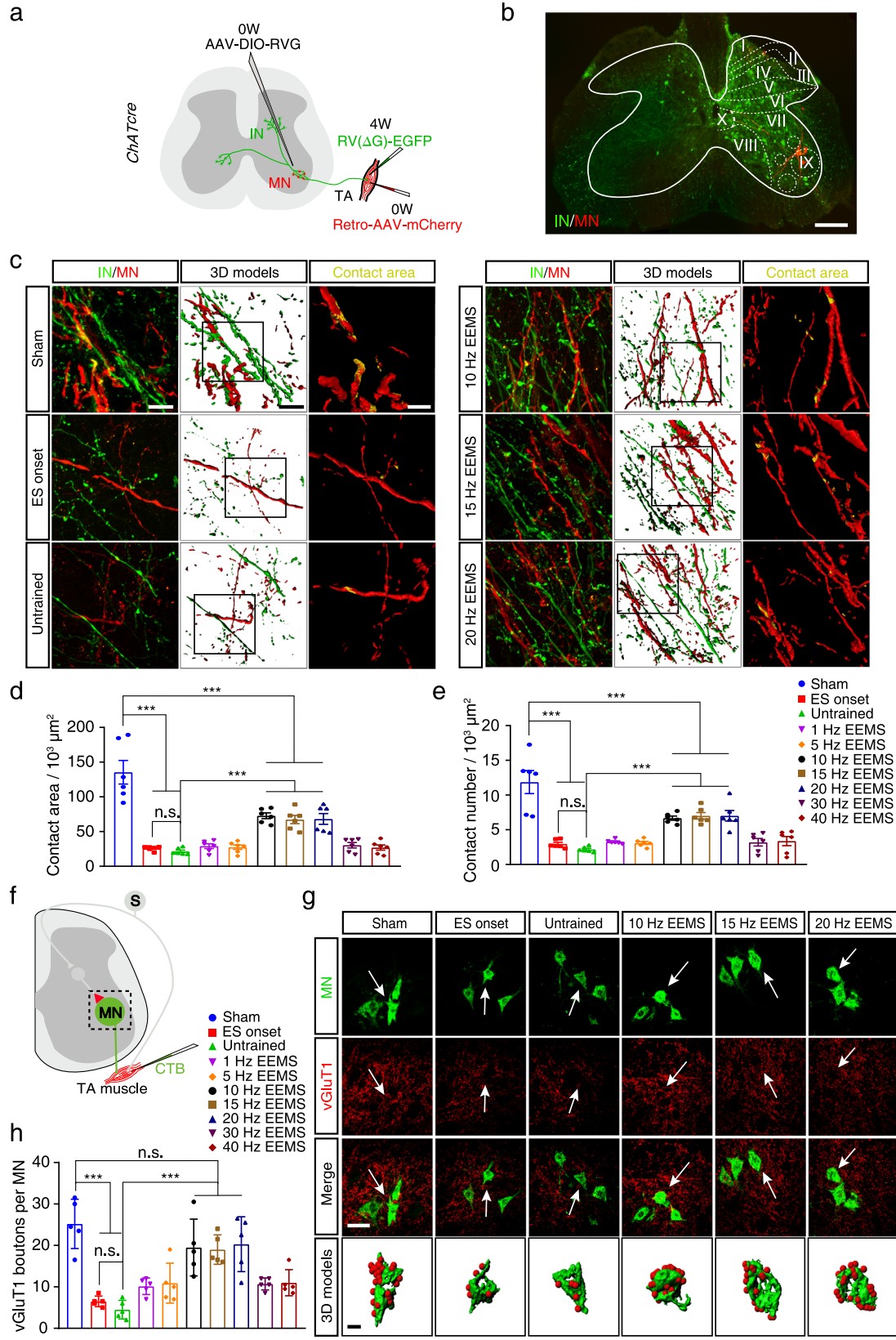

hind limb suspension, the supraspinal excitation source in the lumbar segment was interrupted, and the touch, proprioception and kinesthetic of the hind limb were excluded. EEMS is used to simulate the motor output of the spinal cord and peripheral sensory feedback under normal physiological conditions. The results showed that under the condition of no load on the hindlimb, only 10–20 Hz EEMS could

effectively promote the rearrangement of local neural circuits and the recovery of hindlimb motor function after spinal cord injury compared with other electrical stimulation groups of mice.

We found that the amplitudes of MR and LR were increased in response to stimulation with 10- to 20-Hz EEMS compared with results obtained with the untrained group (Fig. 2d, e). Stimulation of the hind

**Fig. 7 | Connection between spinal premotor interneurons and motoneurons in SCI mice after EEMS. a** Injection scheme to visualize the network of spinal interneurons innervating the TA muscle in *ChAT^cre* mice. **b** Interneurons (green) and motoneurons (red) are shown from an intact mouse. The spatial organization of Rexed's laminae I−X is shown. Scale bar, 200 μm. The experiment was repeated 3 times independently with similar results. **c** Left, synaptic connections between interneurons and motoneurons in the sham, ES onset, untrained, and 10- to 20-Hz EEMS groups. Scale bar, 20 μm. Middle, a 3D model of the connections between interneurons and motoneurons. Scale bar, 20 μm. Right, higher-magnification images of the boxed area. Yellow regions (overlap of GFP and mCherry staining) indicate regions of connection between interneurons and motoneurons. Scale bar,

10 μm. **d, e** Contact area (**d**) and number (**e**) of interneurons per 1000 μm² of the motoneuron surface in the sham, ES onset, untrained, and 1- to 40-Hz EEMS groups (*n* = 6 mice per group). **f** Diagram illustrating how motoneurons were traced. **g** CTB was injected into the TA muscle to retrogradely trace motoneurons (green). vGluT1 (red) labeled axonal terminals of glutamatergic neurons in the sham, ES onset, untrained, and 10- to 20-Hz EEMS groups. Scale bar, 50 μm. The motoneuron and vGluT1 terminal indicated by the white arrow in each image was subjected to 3D modeling. Scale bar, 10 μm. **h** Number of vGluT1 boutons in contact with each motoneuron (*n* = 5 mice per group). Schematics in **a** and **f** were created with BioRender.com. Data represent the mean ± SEM; ns: no statistically significant difference, \*\*\**p* < 0.001, one-way ANOVA followed by the Bonferroni post hoc test.

limb in this manner improved motor function in SCI mice compared with the untrained group (Fig. 3o, p and Fig. S6b, f). Quantitative analysis revealed that the EMG amplitudes of bursts from the TA muscle were increased in the 10- to 20-Hz EEMS groups, as compared with the untrained group (Fig. 3g). Taken together, the results demonstrate that treatment of SCI mice with 10- to 20-Hz EEMS could restore, at least partially (i.e., compared with the sham group), both polysynaptic electrical conduction in sensorimotor circuits and hindlimb motor function, although the restoration did not achieve the level of function observed with the sham group (Figs. 2, 3, and S6). For example, the amplitudes of MR and LR in the 10- to 20-Hz EEMS groups were only 48.4% and 59.1%, respectively, of the values measured for sham group; moreover, the amplitude of the EMG and contractility of the TA muscle were only 19.1% and 61.9%, respectively, of that of the sham group. We speculate that the ability of EEMS to potentiate the recovery from SCI was limited owing to the short training period and application of only a single MS.

Spinal interneurons play a vital role in sensorimotor reflex circuits because the interneurons integrate information feedback from sensory neurons and then transmit motor instructions to activate or inhibit motoneurons to control muscle contraction[60,61]. We found that innervation of motor neurons by sensory neurons and interneurons was disrupted by SCI, whereas disrupted connections could be partially rebuilt by 10- to 20-Hz EEMS (Figs. 5f, g, i, j and 7d, e). However, the connectivity of the reconstructed synapses could not be restored to the level measured for the sham group (Figs. 5f, g, i, j and 7d, e). For example, the number and area of sensory-motor connections reached only 54.3% and 34.1%, respectively, of values measured for the sham group, and the corresponding values measured for interneuron-motoneuron connections were only 58.1% and 51.2% of that of the sham group. In addition, compared with the untrained group, the TA motor neurons of the 10- to 20-Hz EEMS mice were fully activated (Fig. 8m), whereas neurons were activated only 49.8% of that of the sham group. These results suggest that 10- to 20-Hz EEMS could effectively reestablish synaptic connections between neurons in spinal neural circuits, but the number, area, and function of synaptic connections between neurons did not return to preinjury levels. With a single exception, we found that the average volume of motoneuron dendrites and the number of glutamatergic synaptic connections between premotor neurons and motoneurons reached 93.6% and 77.8%, respectively, of that of the sham group (Figs. 6e and 7h).

The results of the functional and structural analyses revealed that neural circuits could be remodeled by application of EEMS to a single muscle and spinal cord, but most of the indicators we measured did not return to preinjury levels. Therefore, longer EEMS treatments could further enhance the restoration of neural-circuit function, leading to more thorough sensorimotor reflex circuit reconstruction and hence greater recovery of function. In the present study, however, owing to technical limitations such as electrode material and size, electrode transplantation in mice could be safely carried out for only 4 weeks, thus prohibiting longer periods of 10- to 20-Hz EEMS application. In the future, we will consider further improving electrode

material and production processes so that the electrode could be transplanted into mice for longer periods.

Motor function in paralyzed hind limbs of SCI rats can be restored after continuous EES with L2 and S1 electrodes[8,9,62]. In our present study, however, EEMS underperformed relative to results obtained in previous studies. The reason for this may be that, in our study, electrical stimulation was given only to the L2–L4 segments of the spinal cord, where stimulation activates only those motor neuron pools that control the flexor muscles, but not the pools that control the extensors, thus preventing complete flexion and extension, however, coordinated movement which require coordinated activation of both muscles. The addition of EES sites on the basis of spinal cord–muscle EEMS may further enhance the motor function of the hind limbs of mice with SCI.

EES of the spinal cord restores locomotion in animal models of SCI but is less effective in humans[63–65]. This interspecies discrepancy is attributable to interference between EES and proprioceptive information in humans[55]. This transient deafferentation prevents modulation of reciprocal inhibitory networks involved in locomotion and reduces or abolishes the conscious perception of leg position. In our study, electrical stimulation of the mouse spinal cord may also have produced a similar antidromic conduction, counteracting incoming proprioceptive information. As both the antidromic response of the EES signal and the forward conduction of the MS signal (Fig. S4b) occur via sensory fibers, the conduction time of both signals should be ≤10 ms. As the MS begins at 15 ms after EES, the MS signal does not interact with the inverse response of the EES. In our EEMS system at the spinal cord and muscle interface, EES was administered first, followed by MS after the end of the spinal-evoked response. Therefore, the paired EES and MS have a timing-sequence stimulation pattern that avoids the mutual interference of antidromic reactions and enhances sensory feedback. We can mitigate the antidromic response of spinal stimulation by EEMS at the spinal-muscular interface to a certain extent.

Theoretically, each neuron has its own neural signal firing frequency[66]. Real-time and multimodal variable electrical stimulation can effectively improve the plasticity of spinal neural circuits after SCI[12,67], but it is not clear how these effective stimuli can be decoded. In the present study, we used a EEMS system to obtain neural signal rules (with respect to certain parameters) that promote sensorimotor reflex circuit reconstruction. Only 10- to 20-Hz EEMS could able to promote the correct reconstruction of sensorimotor reflex circuits in the spinal cord. Hence, the question arises as to whether this electrical stimulation frequency will also be applicable to the reconstruction of other types of spinal circuits. It is likely that neuromodulation therapies of SCI target distinct neural structures but share common principles. Thus, future research may provide a theoretical framework for the broad advancement of neuromodulation therapies to improve function after SCI.

## Methods

All the procedures were in accordance with the Institute of Neuroscience (Soochow University) guidelines for the use of experimental

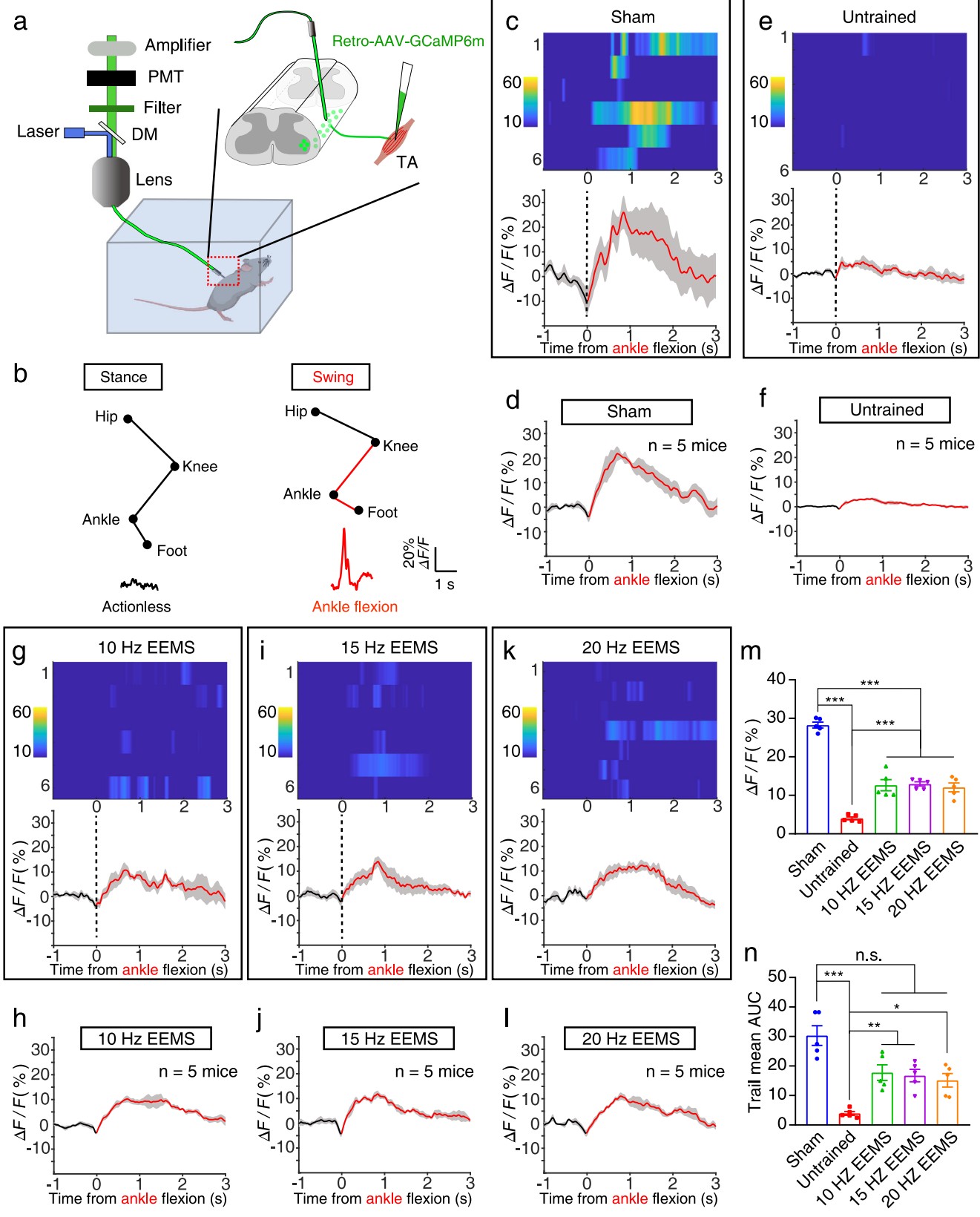

animals and were approved by the Institutional Animal Care and Use Committee at Soochow University.

## Animals

*ChAT^cre* mouse strains (Jackson Laboratory stock #006410) were maintained on a mixed genetic background (129/C57Bl6), kindly provided by Dr. Zilong Qiu (Institute of Neuroscience, Chinese Academy of Sciences, Shanghai, China). *Lbx1^cre* transgenic mice were constructed by GemPharmatech (China) and maintained on a mixed genetic background (129/C57Bl6). Wild-type C57BL/6J mice were purchased from the Shanghai SLAC Laboratory Animal Company (Shanghai, China). All mice including wild-type C57BL/6J, *ChAT^cre*, and

**Fig. 8 | Ca²⁺ signaling in lumbar motoneurons in SCI mice after EEMS.**
**a** Schematic of the fiber photometry setup. Ca²⁺ transients were recorded from motoneurons of freely moving mice. DM, dichroic mirror; PMT, photomultiplier tube. Schematic was created with BioRender.com. **b** Raw traces of changes in GCaMP6 fluorescence that were related to flexion of the ankle joint of the hind limbs. Δ*F/F* represents the change in fluorescence from the mean level before the task. **c**, **e**, **g**, **i**, **k** Ca²⁺ signals associated with ankle flexion in a single mouse from the sham (**c**), untrained (**e**), 10-Hz EEMS (**g**), 15-Hz EEMS (**i**), and 20-Hz EEMS (**k**) groups. Upper, heatmap of Ca²⁺ signals aligned with the initiation of ankle flexion. Each row plots one flexion event, and a total of six flexion events are illustrated. The color scale at the left indicates Δ*F/F*. Lower, plot of the average Ca²⁺ transients. Thick lines indicate the mean, and shaded areas indicate SEM. The red line indicates the time of ankle flexion. **d**, **f**, **h**, **j**, **l** Mean Ca²⁺ transients associated with ankle flexion for the entire test group (*n* = 5 mice per group) for each condition: sham (**d**), untrained (**f**), 10-Hz EEMS (**h**), 15-Hz EEMS (**j**), and 20-Hz EEMS (**l**). **m** Average peak fluorescence for six ankle flexions (*n* = 5 mice per group). The curve represents the mean of 30 signals, and the shaded areas indicate SEM. **n** The AUC of the average fluorescence peak for six ankle flexions (*n* = 5 mice per group). Data represent the mean ± SEM; ns: no statistically significant difference, *$p < 0.05$, **$p < 0.01$, ***$p < 0.001$, one-way ANOVA followed by the Bonferroni post hoc test.

*Lbx1^cre* were male and housed in a specific pathogen-free environment at ambient temperature (24 ± 2 °C), air humidity 40-70% and 12 h dark/12 h light cycle. After surgery, mice were housed individually for 1 week before further experiments. Mice determined to be unhealthy or that died were excluded (<10%). Spinal T9 complete transection was judged to be successful when the BMS score was <1 in mice on day 7 after SCI. All mice received analgesic (carprofen, 5 mg/kg, Merck, Germany) before wound closure and every 12 h for at least 48 h after injury. The health of animals was monitored daily for at least 10 days after surgery, after which health was monitored weekly.

### Electrode parameter
For spinal epidural electrodes, we used a T-shaped FPC substrate with the following layers: polyimide material (PI), 9 μm; Glue, 4 μm; Cuprum (Cu), 12 μm; PI, 8 μm; Glue, 4 μm. The thickness of the gold coating on the electrode wire and contact surface is ignored, and thus the total electrode thickness was 37 μm.

The MS electrode for the mouse hindlimb was as follows: PI, 12.5 μm; Cu, 12 μm; Glue, 25 μm; PI, 12.5 μm. The thickness of the gold coating on the electrode wire and contact surface was ignored, and the total thickness was 62 μm.

### Electrode implantation
All experimental mice were anesthetized with pentobarbital (40 mg/kg, i.p., Sigma, Germany). First, a laminectomy was performed at vertebral level T9 to create an entry point for the implant. An insertable epidural needle electrode plate for mice was slowly inserted along the gap between the exposed spinal cord and vertebrae so that metal contacts covered the L2–L4 segment of the spinal cord, and electrophysiological testing was performed intraoperatively to fine-tune the positioning of electrodes during this process. The electrodes were sutured to the muscles adjacent to the vertebrae, ensuring secure attachment. Subsequently, the muscles, anadesma, and epidermis were sequentially sutured in place. Second, to record spinal cord evoked potential (SCEP) activity, the epidermis of the hind limb is incised and the muscle electrodes are securely affixed to the surface of the tibialis anterior muscle, while suturing is performed to secure the position of the electrodes.

Experiments were performed by an independent researcher who was blinded to the group allocation and the stimulation conditions.

### Spinal cord injury
A spinal cord transection was performed during electrode implantation. After each spinal cord electrode was implanted, the spinal cord was completely transected at T9 using microscissors as described[68,69]. Complete transection lesions were verified postmortem by confirming the absence of neural tissues throughout the dorsoventral extent of the spinal cord[68,69]. Each sham operation was performed as follows: the dorsal vertebral lamina at spinal T9 segment was removed and the electrode was implanted into the spinal cord, but the spinal cord itself was left intact. Subsequently, the muscle layers, fascia, and the skin were sutured. Urine was expressed by manual abdominal pressure twice daily until mice regained reflex bladder function. Surgeries were performed by an independent surgeon who was blinded to the group allocation and the stimulation conditions.

### Electrical stimulus parameters
We found that motoneurons innervating the TA muscle were mainly distributed in the L2-L4 segment of the spinal cord, so that a spinal cord electrode was implanted to cover L2–L4, and a muscle electrode was implanted in the TA muscle. Then, we calculated that the time from the start of spinal pulse to the end of muscle response was 13.34 ± 0.66 ms, and the time from the start of muscle pulse to the end of spinal response was 8.35 ± 0.49 ms. The stimulus frequency range we selected was 1–40 Hz. To prevent overlap of the dual stimulation effects, the sum of the time from the beginning of the spinal pulse to the end of the muscle response and the time from the beginning of the muscle pulse to the end of the spinal response was no more than 25 ms (1/40 s). Therefore, the interval between spinal stimulation and muscle stimulation was set at 15 ms to ensure that the SCEP was fully released. LR amplitudes of SCEPs peaked when the EES intensity was 100–140 μA, indicating that feedforward transmission mediated by spinal sensorimotor circuits was fully activated. Therefore, the EES intensity was adjusted between 100 μA and 140 μA for EEMS and EES. The threshold for MS was between 300 and 400 μA. In order to ensure that each MS could effectively activate the sensory circuit during the electrical stimulation training process, a 400 μA current was applied to the TA muscle, and the effective response (unimodal peak) was recorded in the spinal epidural within 10 ms. Thus, TA muscles received electrical stimulation of 400 μA in the EEMS and MS systems, transmitting sensory feedback to the spinal cord.

It has been reported that 210–400 μA EES promotes hind limb movement in SCI rats[8,62,70,71]. However, due to their smaller size, mice have a lower tolerance for stimulation intensity compared to rats. Our found that when the stimulation intensity exceeds 300 μA, the mice suffered violent twitching of the whole body and stiffness of the hind limbs, indicating that the mice were in an unstable state. Therefore, we set the intensity of the EES to 300 μA in the EEShc group. Experiments were performed by an independent researcher who was blinded to the group allocation and the stimulation conditions.

### Experimental groups
In all experiments and groups, the spinal cord electrode and the muscle electrode were implanted as mentioned in "Electrode Implantation". In the sham group, the spinal cord was intact, and electrical stimulation training was not implemented. In the untrained group, the spinal cord was completely transected, and electrical stimulation training was not implemented.

### EEMS
One week after SCI, awake mice with complete spinal cord transection were fixed in a simple fixation device for electrical stimulation, and a programmable small-animal motor function rehabilitation training device (Suzhou Institute of Biomedical Engineering and Technology, Chinese Academy of Sciences, Suzhou, Jiangsu, China) was connected

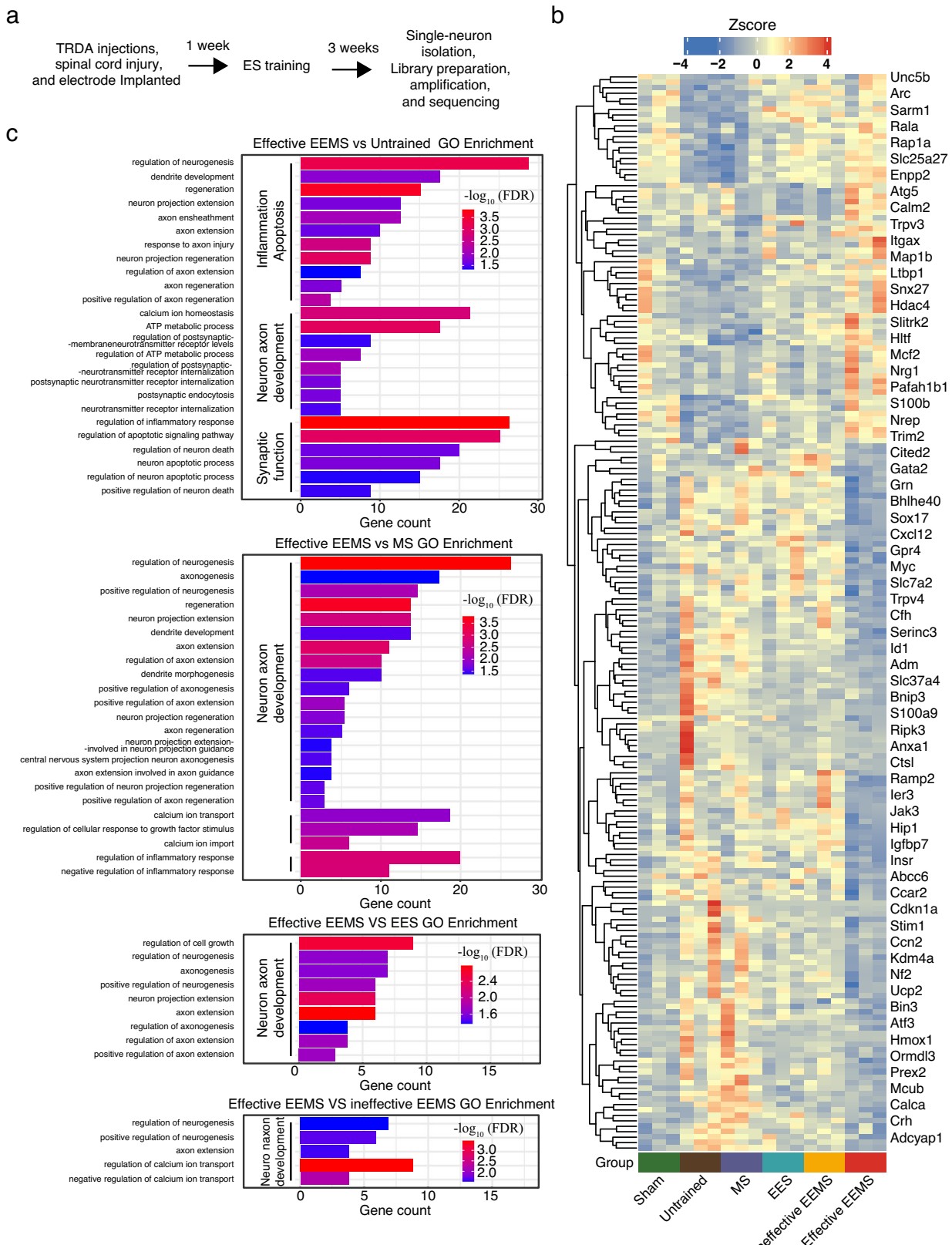

**Fig. 9 | Changes in mRNAs expressed in motoneurons in the spinal sensorimotor circuits of SCI mice after EEMS. a** Overview of RNA sequencing and analysis. **b** Heatmap showing the expression of selected differentially expressed genes (effective EEMS vs sham, effective EEMS vs untrained, effective EEMS vs EES, effective EEMS vs MS, and effective EEMS vs ineffective EEMS). **c** GO terms were generated for genes that were upregulated or downregulated in the effective EEMS groups relative to their levels in the untrained group, EES group, MS group, or ineffective EEMS (-log$_{10}$FDR > 1.5). The term GO is mainly associated with Neuron axon development, Synaptic function, and Inflammation and apoptosis.

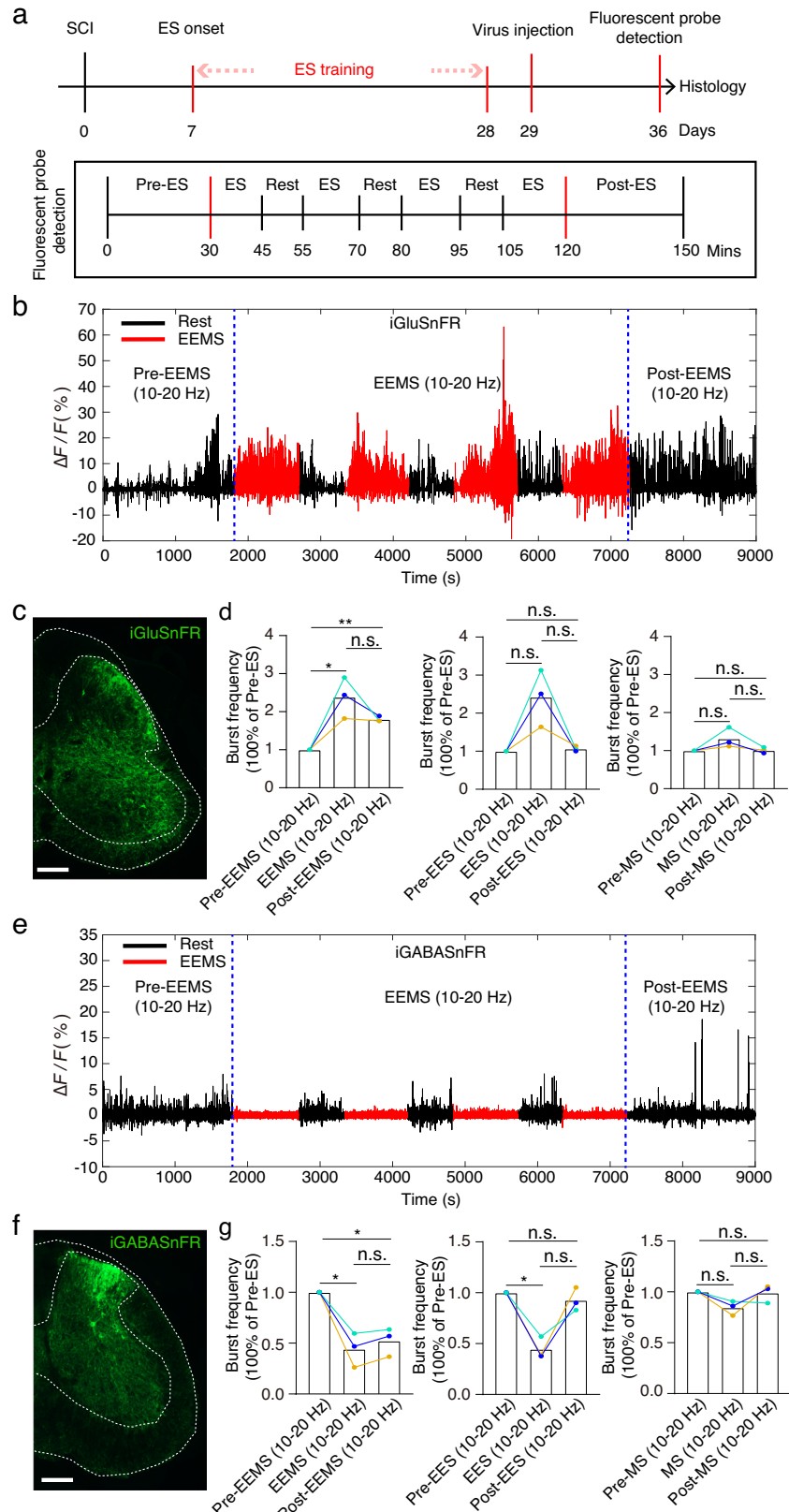

to the EES electrode and the MS electrode for stimulation training. The spinal cord was stimulated with 100–140 µA intensity, and after 15 ms, the TA muscle was stimulated with 400 µA intensity that comprised a single round of EEMS. The interval of each round of EEMS was based on the stimulation frequency, as follow: 25 ms (40 Hz), 33.33 ms (30 Hz), 50 ms (20 Hz), 66.66 ms (15 Hz), 100 ms (10 Hz), 500 ms (5 Hz), and

1000 ms (1 Hz). During electrical stimulation training, each mouse was rested for 10 min after 15 min of stimulation training. Different experimental groups were trained with different stimulus frequencies for a total of 3 weeks, with training taking place 6 days per week. Mice in the EEMS group were trained daily for 90 min (stimulation, 60 min; rest, 30 min), starting from 7 days after SCI.

**Fig. 10 | The impact of various electrical stimulation systems on the neurotransmitter flow of Glu and GABA. a** Flowchart illustrating the experimental setup for electrical stimulation training and fluorescent probe detection. **b** Burst curve of Glu flow pre, during and post EEMS (10–20 Hz). **c** Fluorescence image of glutamatergic neurons labeled by iGluSnFR. Scale bar: 200 µm. The experiment was repeated 3 times independently with similar results. **d** Left: The relative change in the burst frequency of Glu flow during and post EEMS (10–20 Hz) was compared to that pre-EEMS (10–20 Hz) ($n = 3$ mice per group). Middle: The relative change in the burst frequency of Glu flow during and post EES (10–20 Hz) was compared to that pre-EES (10–20 Hz) ($n = 3$ mice per group). Right: The relative change in the burst frequency of Glu flow during and post MS (10–20 Hz) was compared to that pre-MS (10–20 Hz) ($n = 3$ mice per group). **e** Burst curve of GABA flow before, during and after EEMS (10–20 Hz). **f** Fluorescence image of GABAergic neurons labeled by iGABASnFR. Scale bar: 200 µm. The experiment was repeated 3 times independently with similar results. **g** Left: The relative change in the burst frequency of GABA flow during and post EEMS (10–20 Hz) was compared to that pre-EEMS (10–20 Hz) ($n = 3$ mice per group). Middle: The relative change in the burst frequency of GABA flow during and post EES (10–20 Hz) was compared to that pre-EES (10–20 Hz) ($n = 3$ mice per group). Right: The relative change in the burst frequency of GABA flow during and post MS (10–20 Hz) was compared to that pre-MS (10–20 Hz) ($n = 3$ mice per group). Data represent the mean ± SEM, ns: no statistically significant difference, $*p < 0.05$, $**p < 0.01$, statistical analysis was carried out with a two-tailed paired $t$-test (**d** and **g**).

## EES
The stimulation conditions for the group were the same as for the EEMS group except that the TA muscle was not stimulated.

## MS
The stimulation conditions for each experimental group were the same as for the EEMS group except that the EES was not performed.

## Epidural Electrical Stimulation-high current (EEShc)
The stimulation conditions for each experimental group were the same as for the *EES* group except that the current intensity of the epidural electrical stimulation was 300 µA.

## Electrical stimulation training
In mice of the sham group, the electrodes were implanted during a sham operation, but electrical stimulation training was not performed. In mice of the untrained group, electrodes were implanted in the spinal cord during the T9 segment transection surgery to avoid secondary damage due to electrode implantation, but the electrical stimulation training was not performed. Mice in the 1- to 40- Hz EEMS, 1- to 40- Hz MS, 1-to 40- Hz EES, and 1- to 40- Hz EEShc groups were implanted with stimulating electrodes during the T9 segment transection surgery, and the mice were trained with electrical stimulation 7 days later. The mice underwent electrical stimulation training for 3 weeks. SCEP was recorded at 7 and 29 days after electrode implantation in all experimental groups. All mice were implanted with the stimulation electrodes on the same day.

Experiments were performed by an independent researcher who was blinded to the group allocation. Mice were randomly assigned to each group to receive training.

## SCEP recording
For each anesthetized mouse, a 16-channel physiological signal recording and analysis system (BIOPAC, USA) was used to record SCEPs. The epidural stimulation electrode was connected to the STM 100 C stimulation module, and the muscle electrode was connected to the EMG 100 C recording module. AcqKnowledge software (BIOPAC, USA) was used to set the parameters of the STM 100 C stimulation module, including a stimulation pulse frequency of 1 Hz and a pulse width of 0.2 ms. When the EMG 100 C recording module was used to collect the signals of TA on the right hindlimb, the following filter settings were used: Band Stop channel 1 parameter, 50 Hz; Band Stop channel 2 parameter, 100 Hz; and Low Pass channel parameter, 2000 Hz.

Fifty SCEPs were recorded for each mouse, and data were exported in TXT format. For analyzing SCEPs, MATLAB software (MathWorks, Inc., USA) which was developed in collaboration with Dr. Shouyan Wang (Institute of Brain-like Intelligent Science and Technology, Fudan University, Shanghai, China), was used to divide and align the data. A series of continuous SCEPs were divided into single waveforms of the same duration, and then 50 single waves were aligned and superimposed. A baseline was calibrated for each SCEP, and waveforms were averaged to calculate the peak value of the waveform.

Experiments were performed by an independent researcher who was blinded to the group allocation and the stimulation conditions.

## Magnetic Resonance Imaging
After undergoing anesthetized mice were positioned supine and securely immobilized using medical tape. The spinal cord was imaged using a high-resolution multi-directional T2-weighted imaging (T2WI) technique with a 11.7 T nuclear magnetic resonance imaging (BioSpec 11.7 /16, Brugg, Germany) employing fast spin echo (FSE). The imaging parameters were as follows: repetition time (TR) = 2000 ms, echo time (TE) = 26 ms, field of view (FOV) = 3 cm × 3 cm, matrix size =128 × 128, slice thickness = 0.5 mm, and cumulative scans performed four times. To minimize the influence of respiratory motion on the results, a respiratory gating technique was employed. Images from MRI scans were used directly (without processing) to determine the position of the electrodes in the spinal cord.

## Electrophysiological and pharmacological experiments
Pentobarbital (40 mg/kg, i.p.) was injected to anesthetize the mice implanted with electrodes. The following experiments were then conducted: (1) EES (100-140 µA) was applied to the spinal cord using a 1-Hz frequency and 0.2 ms pulse width so that SCEPs were recorded in the TA muscle, and (2) pharmacological modulation of EES-evoked motor responses was tested 30 min after using the sodium channel blocker TTX (1 µM, Sigma, Germany); the 2-adrenergic receptor agonist tizanidine (1 mM, MedChemExpress, USA); PTZ (5 mM, Sigma, Germany), an inhibitor of γ-aminobutyric acid (GABA) and the GABA-A receptor; AP5 (0.5 mM, Sigma, Germany), an NMDA receptor antagonist; and CNQX (50 µM, MedChemExpress, USA), a potent and competitive AMPA/kainate receptor antagonist. All drugs (10 µl) were delivered intrathecally to spinal segment L5−L6. Detailed information of the drugs could be found in Supplementary Table 1. Experiments were performed by an independent researcher who was blinded to the group allocation and the stimulation conditions.

## Spinal injections
For spinal cord−targeted viral deliveries, we performed stereotaxic injections using high-precision instruments (Stoelting, USA) while the mice were under pentobarbital anesthesia. The vertebrae of the L2−L4 segment of the spinal cord were exposed. A small hole was drilled in the middle of each of the three vertebrae in a rostral-to-caudal orientation, and a pulled calibrated glass pipette (Hamilton, USA) was used for local infusion of 500 nl of a virus solution (see below) using a single-channel microinjection-withdrawal pump (KDS, USA). The glass pipette was retracted after 10 min. The coordinates used to locate the spinal cord were 35.5 degrees from the midline of the spinal cord and a depth of 1 mm. Experiments were performed by an independent researcher who was blinded to the group allocation and the stimulation conditions.

## Muscle injections

The TA muscle was selected as the target muscle for injection of virus or tracer. A pulled calibrated glass pipette and a microsyringe were connected using hot-melt adhesive. Virus or tracer was mixed with an appropriate amount of fast green dye (Sangon Biotech, China) before injection to facilitate observation. The TA muscle was divided into three regions from top to bottom, and 2 µl of virus or tracer was slowly injected into each of these regions (i.e., three injection sites); the glass pipette was retracted from each injection site after a 3 min period. Muscle injection specificity was always verified postmortem by the presence of fluorescence exclusively located in the targeted muscle. Experiments were performed by an independent researcher who was blinded to the group allocation and the stimulation conditions.

## Spinal Sensorimotor Circuit Tracing

We labeled multiple elements of the spinal sensorimotor circuits: the motoneurons, the premotor circuit, proprioceptive axons, dorsal root ganglion (DRG) fibers, sensory neuron−motoneuron connections, and sensory neuron-interneuron[lbx1] connections. For retrograde labeling of motoneurons, we injected AAV2/2Retro-hSyn-EGFP-WPRE-pA (Taitool Bioscience, China) into TA muscles of mice that underwent electrical stimulation training for 1 week, and continued the electrical stimulation training for 2 weeks before perfusion. The same injection method was used for delivery of Cholera toxin B subunit (Thermo Fisher Scientific, USA).

For G-deleted monosynaptic rabies premotor circuit tracing, AAV2/9-EF1α-DIO-oRVG-WPRE-hGH-pA (BrainVTA, China) was injected first into the ventral horn of the spinal cord of $ChAT^{cre}$ mice, and AAV2/2Retro-nEf1α-mCherry-WPRE-pA (BrainVTA, China) was injected into the TA muscle followed by T9 complete transection surgery. After 3 weeks of electrical stimulation training, RV-N2C (G)-ΔG-EGFP (BrainVTA, China) was injected into the TA muscle. Mice were sacrificed 2 weeks after injection.

To label DRG fibers, we pressure-injected 0.5 µl of pAAV-CAG-EGFP-3xF-LAG-WPRE (Obio Technology, China) into the DRG at lumbar region L2−L4 using a finely pulled glass micropipette (at a coordinate of 0.3 mm below the DRG surface). The micropipette was left in place for 3 min after the injection to avoid backflow. Finally, for all relevant procedures, the wounds were closed with layered sutures. After 1 week of electrical stimulation training, the DRG was injected with a virus, and the electrical stimulation training was continued for an additional 2 weeks before perfusion.

Approximately 2 µl of 10% TRDA (Invitrogen, USA) dissolved in double-distilled $H_2O$ was slowly injected into TA muscles using glass micropipettes to mark specific motoneurons. At the same time, DRG fibers were labeled as described above.

To label interneuron[lbx1], AAV-DIO-mCherry (Brain Case, China) was injected into the intermediate area of gray matter at the L2−L4 segment of $Lbx1^{cre}$ mice, followed by T9 complete transection surgery.

Detailed information of the virus could be found in Supplementary Table 1. Experiments were performed by an independent researcher who was blinded to the group allocation and the stimulation conditions.

## Tissue clearing technique

We first prepared the two CUBIC (clear, unobstructed brain/body imaging cocktails and computational analysis) solutions. Reagent 1 consisted of 12.5 g urea (Sigma, Germany), 12.5 g *N, N, N′, N′*-tetramethylethylenediamine (Sigma, Germany), and 17.5 ml water. This mixture was heated and stirred at 61 °C until completely dissolved; then, the solution was cooled to room temperature. Finally, 7.5 g Triton X-100 (MasterTech, USA) was added, and the solution was thoroughly stirred. Reagent 2 consisted of 25 g sucrose (Sinopharm, China), 12.5 g urea, and 7.5 ml water. This mixture was heated and stirred at 61 °C until completely dissolved; then, the solution was cooled to room temperature. Finally, 5 g 2,2′,2″-nitrilotriethanol (Aladdin, China) and 400 µl of 10% Triton X-100 were added to the solution, and the solution was thoroughly stirred. These solutions were prepared fresh before each experiment.

The mice were perfused using standard methods, and the spinal cord was removed and immersed in 4% paraformaldehyde (Sigma, Germany) at 4 °C for 12 h before proceeding to tissue clearing. To clear the tissue, the spinal dura mater of the L2−L4 segment was removed, and the tissue was transferred to a 50 ml centrifuge tube with 40 ml reagent 1 and placed in a 37 °C shaker for 4 days. The spinal cord was removed from reagent 1 and washed with phosphate-buffered saline (PBS; pH 7.4; Sinopharm, China) with three changes over a 3 h period, placed in 20% sucrose in PBS for dehydration for 12 h at 4 °C, and then the tissue was transferred to a 50 ml centrifuge tube with 40 ml reagent 2 and placed in darkness at room temperature for 4 days.

Experiments were performed by an independent researcher who was blinded to the group allocation and the stimulation conditions.

## Whole-mount immunohistochemical staining

The spinal cord was placed in 20% sucrose in PBS for dehydration for 12 h at 4 °C (see above), and then embedded in O.C.T. (Sakura Finetek, USA) and stored in a −80 °C freezer for 12 h. The next day, the tissue was washed with PBS (change three times within 3 h) and placed in goat anti-GFP (Ab6662; Abcam, USA) solution diluted with 2% BSA (Sangon Biotech, 1:50, China) for 3 days at room temperature. The tissue was then washed with 0.1% Triton X-100 in PBS (change three times within 3 h) and subsequently incubated with an appropriate secondary antibody (see below) and washed the same way. Next, the tissue was transferred to 20% sucrose in PBS and was dehydrated overnight at room temperature. Finally, the tissue was placed in reagent 2. Experiments were performed by an independent researcher who was blinded to the group allocation and the stimulation conditions.

## Immunohistochemistry

Pentobarbital (40 mg/kg, i.p.) was injected to anesthetize the mice before sacrificing them. Mice were perfused with 4% paraformaldehyde in PBS. Each spinal cord was dissected out and washed with PBS. The tissue was then dehydrated in 30% sucrose overnight to replace paraformaldehyde with sucrose. Then, the L2−L4 segments of spinal cord were embedded with O.C.T. for cryosectioning, and the resulting coronal sections (35 µm thick) were used for immunostaining. Primary antibodies diluted in BSA were incubated at 4 °C at the following concentrations: Go-GFP (Ab6662; Abcam, USA), 1:200; Rb-CTB (PA125635; Invitrogen, USA), 1:200; Rb-mCherry (Ab183628; Abcam, USA), 1:200; Mo-vGluT1 (MAB5502; Merck, Germany), 1:200; Rb-c-Fos (2250 s; Cell Signaling Technology, USA), 1:200; Rb-Akt (4691 s; Cell Signaling Technology, USA), 1:200; Rb-p-Akt (4060; Cell Signaling Technology, USA), 1:200; Mo-NF (Ab82259; Abcam, USA), 1:500; α-BTX-555 (B35451; Thermo Fisher Scientific, USA), 1:1000; Rb-syn (Ab32127; Abcam, USA), 1:500. The tissues were washed in PBS, incubated at 4 °C for 12 h with the appropriate Alexa Fluor−conjugated secondary antibody (1:800; Abcam, USA) diluted in BSA, and washed with PBS again. Images were acquired with a confocal microscope (LSM700; Zeiss, Germany) and processed and exported with Zen software (Zeiss, Germany). Experiments were performed by an independent researcher who was blinded to the group allocation and the stimulation conditions.

## Motoneuron reconstructions and quantification

Images of spinal cord sections (at least 30 layers each) were acquired using a LSM700 confocal microscope (20× objective) in Z-stack mode. Motoneurons were reconstructed using IMARIS software (Bitplane, Switzerland). The dendrite surface was reconstructed using the

IMARIS Surface tool, and the volume of each distal part of the moto-neuron dendrite was calculated. At the same time, the Filaments tool was used to reconstruct the dendrite branches of motoneurons and count the number of dendrite terminal points. For each experimental group ($n = 6$ mice per group), the visual field with the largest number of neurons in the spinal cord slice was selected to be imaged. Quantification was performed by an independent researcher who was blinded to the group allocation and the stimulation conditions.

### Analysis of Interneurons, DRG Fibers, and Motoneuron Connections
Images were acquired with a LSM700 confocal microscope (64× objective) in Z-stack mode. Complete motoneurons or dendrites were continuously scanned. The IMARIS Contact Area tool was used to assess the synaptic input of interneurons and DRG fibers to moto-neurons. Quantification was performed by an independent researcher who was blinded to the group allocation and the stimulation conditions.

### Quantification of synapses
High-resolution images of glutamatergic synaptic input to retro-gradely marked motoneurons were acquired with a LSM700 confocal microscope (64× objective) in Z-stack mode and quantified using IMARIS software. We used the IMARIS Point Detection tool to distinguish and quantify vGluT1 labeling with a diameter of 2 µm as an estimate of glutamatergic neuron input. The 'Find Spots Close to Surface' option of the Spot Detection tool was selected to distinguish and quantify connection between glutamatergic neurons and moto-neurons. For proprioceptive axon input analyses to the spinal cord, we used the IMARIS Spot Detection tool to differentiate and quantify axon inputs of >1.5 µm in diameter as an estimation of proprioceptive input. Quantifications were performed by an independent researcher who was blinded to the group allocation and the stimulation conditions.

### Analysis of dendritic complexity
Images of segment L2−L4 of the spinal cord were acquired with a LSM700 confocal microscope (10× objective) in Z-stack mode, and at least 150 layers of images were acquired per mouse. The Filament Tracing module in IMARIS was used to reconstruct the morphology of motoneuron dendrites in that area. The number of neurons and total length of dendrites were measured, and the average dendrite length of each motoneuron was calculated as an assessment of neuron complexity. Quantification was performed by an independent researcher who was blinded to the group allocation and the stimulation conditions.

### Myofilament and Neuromuscular Junction (NMJ) Analyses
The TA muscle was fixed with 4% paraformaldehyde at 4 °C and washed three times in PBS for 10 min each time. Subsequently, the TA muscle was dehydrated by incubation in 20% sucrose in PBS and embedded in O.C.T. Frozen sections of tissue were stained with hematoxylin and eosin for histological analysis. Images were acquired with a LSM700 confocal microscope (20× objective) and processed and exported with Zen software. Finally, the area of each myofilament was measured by ImageJ (NIH), and Prism8 (Graph Pad, La Jolla, CA, USA) was used to analyze differences among groups.

To stain the NMJs, the TA muscle was fixed with 4% paraformaldehyde at 4 °C for 12 h. After washing in PBS, the muscle was separated into bundles of 5−10 muscle fibers under a microscope and placed in PHT (2% BSA and 1% Triton X-100 dissolved in PBS) at room temperature for 1 h. The antibodies Mo-NF, Rb-syn, and α-BTX were used to label the NMJs. Images were acquired using a LSM700 confocal microscope in Z-stack mode. The percentage of pre-synaptic (syn) area to post-synaptic (α-BTX) area was calculated for each NMJ[72].

Experiments were performed by an independent researcher who was blinded to the group allocation and the stimulation conditions.

### Fiber Photometry
AAV2/2Retro-EF1a-GCaMP6m-WPRE-hGH-pA virus (BrainVTA, China) was injected into the TA muscle to express GCaMP in the corresponding motoneurons in the spinal cord. After 3 weeks of electrical stimulation training, vertebrae in spinal cord segment L3 were exposed, and the right side of the vertebrae was opened with a skull drill. An optical fiber (250 µm O.D., 0.37 NA; Shanghai Fiblaser, China) was placed in a ceramic ferrule. The ferrule was secured to the stereotaxic instrument, and then the optical fiber was implanted in the spinal cord (0.3 mm to the right of the midline of the spinal cord at an angle of 20 degrees; implantation depth, 0.7−0.8 mm). Biological tissue glue (3 M Vetbond, USA) and dental cement (Peolankg, China) were used to attach the ceramic ferrule to the vertebra. In addition, rAAV-hSyn-GCaMP6f (Brain Case, China) was injected into the right dorsal horn of the spinal cord to induce the dorsal-horn neurons to express GCaMP6. At the same time, optical fibers were implanted into the dorsal horn of the spinal cord and fixed in place. Mice were allowed to recover for 1 week after fiber implantation. They were handled and habituated to the environment of the behavioral tests for 3 days. In all experiments, mice with incorrect injection sites were excluded from further experimentation.

To record fluorescence signals, a fiber optic patch cord guided the light between a commutator (Doric Lenses, Canada) and the implanted optical fiber. The laser power was adjusted such that the light emitted at the tip of the optical fiber was at a low level of 10−30 µW to minimize bleaching. Analog voltage signals were digitized at 200−500 Hz and recorded with CamFiberPhotometry software (Thinker Tech Nanjing Biotech Limited, China).

In the awake state, GCaMP responses to ankle flexion were recorded using fluorescence values obtained before ankle flexion (i.e., −1 s to time 0) and after ankle flexion (time 0 to +3 s). The $Ca^{2+}$ signaling associated with ankle joint flexion in mice was recorded six times. The data were segmented based on the behavioral events within the individual trials. As a baseline value, $\Delta F/F$ was calculated 1 s before ankle flexion. Photometry data were analyzed with custom-written MATLAB codes (MATLAB R2018a, MathWorks, Inc., China).

To calculate the GCaMP6m signal, the relative fluorescence changes of $\Delta F/F$ were calculated to determine the $Ca^{2+}$ signal as follows:

$$\Delta F/F = (F - F0)/F0 \qquad (1)$$

where $F_O$ is the average value of baseline value at the reference time point and $F$ is the fluorescent signal collected during the experiment. Experiments and quantification were performed by an independent researcher who was blinded to the group allocation and the stimulation conditions.

### Neurotransmitter probe detection
After 29 days of electrode implantation, rAAV-hSyn-iGluSnFR3.v857.GPI (Brain Case, China) and rAAV-hSyn-iGABASnFR (Brain Case, China) were injected into the L4 segment of the spinal cord (right side of midline: 0.5 mm, depth: 0.6 mm), that these plasmids encode a reporter that fluoresces in response to binding of the appropriate neurotransmitter. At the same time, optical fibers were implanted at the injection site and fixed. The awake mice were fixed on a simple fixed frame, and the fluorescence changes corresponding to Glu and GABA neurotransmitter release in the spinal cord of the sham group and the untrained group were detected by a neural signal recorder (Thinker Tech Nanjing Biotech Limited, China). For the 10−20 Hz EES, 10−20 Hz MS and 10−20 Hz EEMS groups, fluorescence

changes were detected before, during, and after stimulation. Each mouse was continuously recorded for 2.5 h (0.5 h before stimulation, 1.5 h during stimulation, and 0.5 h after stimulation).

## Basso Mouse Scale (BMS) Scoring

Mice were tested for right hindlimb functional deficits at 0, 1, 3, 7, 14, 21, and 28 days ($n = 10$ mice per group) after SCI. Hindlimb locomotor recovery was assessed in an open field using the BMS, which was previously described in detail[73]. This scale ranges from 0, indicating complete paralysis, to 9, indicating normal hindlimb movement. Quantification was performed by an independent researcher who was blinded to the group allocation and the stimulation conditions.

## Electromyography (EMG)

Each group included six mice. In the sham group, the electrodes were implanted during the sham operation, but electrical stimulation training was not performed. In the untrained group, electrodes were implanted in the spinal cord during the T9 segment transection surgery to avoid secondary damage due to electrode implantation, but electrical stimulation training was not performed. Mice in 10- to 20- Hz EEMS, 10- to 20- Hz MS, 10- to 20- Hz EES and 10- to 20- Hz EEShc groups were implanted with stimulating electrodes during the T9 segment transection surgery, and the mice were trained with electrical stimulation 7 days later. The mice underwent electrical stimulation training for 3 weeks, and then the TA muscle myoelectric burst and joint activity trajectory were detected when the mice were moving on a treadmill.

Starting 4 weeks postinjury, training took place on a body weight-supporting (90%-, 2 m·min$^{-1}$) treadmill (SANS, China). Due to the absence of a body weight-supported automatic adjustment system in the treadmill used in our study, a research assistant was required to manually adjust the support arm to ensure that the hind limbs of mice made proper contact with the treadmill. This resulted in undetermined information about the weight load of each mouse during training. The mice were attached to the treadmill while awake, and the muscle electrodes on the right hindlimb were connected to an EMG recording module (BIOPAC, USA). The mice were acclimated to the treadmill for 1 h before surface EMG testing. At the same time, the research assistant needed to adjust the support arm in real time to maintain the balance of the mice's body when the mice were moving on the treadmill, which resulted in the movement of the support arm. Nonetheless, the research assistant did not subjectively adjust the support arm to help the mice move. To minimize the impact on each mouse, research assistants were asked to keep the mouse's body balanced while minimizing adjustments to the support arm. For each mouse, recording began once a stable hind limb gait was achieved. An EMG-100C amplifier (BIOPAC, USA) continuously recorded a 30-second EMGs (filter settings: Band Stop channel 1 parameter, 50 Hz; Band Stop channel 2 parameter, 100 Hz; and Low Pass channel parameter, 2000 Hz), and its traces were analyzed via Acknowledge (BIOPAC, USA). There were three identical treadmills in the laboratory for the experiment, and six technical assistants were involved. Each group contained six mice, which were randomly assigned to six research assistants, ensuring that the mice of different stimulus groups were spread equally among different research assistants. Research assistants were blinded to the stimulation condition and grouping of the assigned mice.

## Quantification of motor function

The TA muscle is a flexor muscle that controls the ankle joint. When the TA muscle contracts, the ankle joint flexes. Here, muscle contractility refers to the stretching and contracting of muscles, and also represents the flexion process of the ankle joint. A pressure-sensitive sensor was attached to the skin overlying the TA muscle of each mouse's right hindlimb in the EEMS, EES, EEShc, and MS groups, and TA muscle

contractility was measured as described[39]. Briefly, after the 3-week training protocol for each of EEMS, EES, EEShc, and MS mice, the muscle electrode was removed, and the skin was sutured. A pressure-sensitive sensor (Suzhou Institute of Nano-tech and Nano-bionics, Chinese Academy of Sciences, Suzhou, China) was attached to the TA muscle epidermis, and then 1-V constant voltage was applied to the sensor. Mice moved freely in the open field, and when the TA muscle relaxed, the squeezing force of the muscle on the sensor was small, resulting in a large sensor resistance, and the current passing through was denoted $I_O$; when the TA muscle contracted, the squeezing force of the muscle on the sensor was large, resulting in a reduction of sensor resistance, and the current passing through was denoted $I$. The relative current through the sensor was as follows: (2) $\Delta I/I_O = (I - I_O) / I_O$. $\Delta I/I_O$ was applied to quantify the degree of contraction of the TA muscle.

Quantification was performed by an independent researcher who was blinded to the group allocation and the stimulation conditions.

## Transcriptome analysis

Three weeks after electrode implantation, the TA motoneurons of mice in the sham ($n = 3$ mice), untrained ($n = 3$ mice), 1- to 40- Hz EEMS ($n = 7$ mice), 10- to 20- Hz EES group ($n = 3$ mice), 10- to 20- Hz MS group ($n = 3$ mice) groups were labeled by TA intramuscular injection of TRDA, as described above. One week later, the mice were deeply anesthetized and fresh intact lumbar spinal cord tissue was collected after each mouse was sacrificed. Each freshly isolated spinal cord was immobilized in an AGAR block "V" shaped groove and sliced on a Vibratome (Leica, Germany) into 200- to 300-μm-thick coronal sections under ice bath and oxygenated conditions. The spinal sections were transferred to a patch-clamp bath for continuous oxygenation. TRDA-labeled motoneurons were aspirated under a fluorescence microscope (Zeiss, Germany) with a glass electrode at negative pressure using a microoperating system (Eppendorf, Germany). No less than 50 motoneurons from each mouse were aspirated for single-cell transcriptome sequencing. TA motor neurons were placed in single-cell lysis solution (Annoroad, China).

TRDA-labeled neurons were individually homogenized in 1 ml TRIzol (Invitrogen, USA), and RNA was extracted with a RNeasy Mini kit (Qiagen, Germany) or equivalent. Total RNA (1 μg) was used to synthesize double-stranded cDNA. First, reference genome sequences and gene model annotation files of related species were downloaded from genome websites such as UCSC, NCBI, and ENSEMBL. Second, Hisat2 software (v2.0.1, http://daehwankimlab.github.io/hisat2/) was used to index the reference genome sequence. Finally, clean data were aligned to the reference genome via Hisat2. Then, transcripts in FASTA format were converted from known gff annotation files and indexed. Then, with the file that served as the reference genome file, HTSeq (v0.6.1, https://htseq.readthedocs.io/en/master/) was used to estimate gene and isoform expression levels from the paired-end clean data. Differential expression analysis was performed using the DESeq2 Bioconductor package, a model based on a negative binomial distribution. The estimates of dispersion and logarithmic fold changes incorporated data-driven prior distributions[74]. We adjusted $p$-values for multiple testing in R using Benjamini and Hochberg[75] method to control the false discovery rate (FDR). Genes with an adjusted $p$-value ≤ 0.1 and a fold change ≥ 1.5 were considered significant. Because it is inevitable that other types of cells will be aspirated along with the targeted neurons, we analyzed the expression of the motor neuron-specific genes *ChAT*, *Isl1*, and *Isl2* in the samples. *ChAT*, *Isl1*, and *Isl2* were consistently expressed in each sample, which indicated that this method of motor neuron aspiration was reliable and ensured the validity of the data.

Raw count matrix for differential analysis was obtained by transcriptome sequencing (HTSeq software). A threshold value of fold change ≥ 1.5 and FDR ≤ 0.1 was set to identify differentially genes (effective EEMS vs untrained, effective EEMS vs EES, effective EEMS vs

MS, and effective EEMS vs ineffective EEMS). GO enrichment analysis was performed on all differentially expressed genes, which resulted in three functional classifications: molecular function (MF), cell component (CC), and biological process (BP). Each classification consisted of a series of GO terms, and each GO term was associated with a group of differential genes. Genes related to axon development, Synaptic function, Inflammation and apoptosis are involved in the regulation of neuronal plasticity[44]. Because the specific frequency of EEMS enhanced the plasticity of motoneurons, we focused on the functional terms Neuron axon development, Synaptic function, and Inflammation and apoptosis. GO terms related to these three functional terms were screened, and the differentially expressed genes associated with these GO terms are listed in Supplementary Table 2.

GOSeq (v1.34.1, http://bioconductor.org/packages/release/bioc/html/goseq.html) was used to identify gene ontology (GO) terms that were associated with the list of enriched genes with a significant adjusted p-value or padj value of <0.1. In addition, topGO was used to plot directed acyclic graph[76]. Kyoto Encyclopedia of Genes and Genomes (KEGG) was a collection of databases associated with genomes, biological pathways, diseases, drugs, and chemical substances (http://en.wikipedia.org/wiki/KEGG). We used in-house scripts to identify significantly differentially expressed genes that were represented in the KEGG database.

Experiments and quantification were performed by an independent researcher who was blinded to the group allocation and the stimulation conditions.

### RNA isolation and quantitative real-time polymerase chain reaction

Total RNA was extracted using Tissue RNA Purification Plus Kit Plus (YiShan Biotechnology, China). Total RNA (1 μg) was used to synthesize cDNA by reverse transcription using the HiScript III All-in-one RT SuperMix Perfect for qPCR Kit (Vazyme, China). Quantitative real-time PCR (qPCR) test was conducted with 2× SYBR Green qPCR Master Mix (Low ROX; Bimake, USA) using a real-time PCR Detection System (ABI 7500, Life Technologies, USA). The cycling conditions included a 10-min initial denaturation step at 95 °C followed by 40 cycles of 15 s at 95 °C, 30 s at 60 °C and 30 s at 72 °C. Target gene expression was normalized to that of the housekeeping gene GAPDH. Relative fold difference in expression was calculated using the $2^{-\Delta\Delta Ct}$ method after normalization to GAPDH expression. Primers (Genewiz and Sangon Biotech, China) are shown in Supplementary Data 1. Experiments and quantifications were performed by an independent researcher who was blinded to the group allocation and the stimulation conditions.

### Statistical analysis

All statistical analyses and plots were executed using Prism software or MATLAB. All computed parameters were quantified and compared between groups. Statistics were performed on averaged values per mouse. All data are reported as the mean ± SEM. Normality was initially assessed for all data before analysis. Significance was determined using the Mann Whitney test, two-sample t-test, or one-way analysis of variance followed by the Bonferroni test for multiple comparisons. Two-way ANOVA was also used and was followed by the Tukey post hoc test for multiple comparisons. Specific tests are indicated below with their relevant figures. No randomization was used in our experiments. The criterion for statistical significance was p < 0.05.

We used a two tailed unpaired t-test for the analyses in Fig. 1h, Fig. S10b–e, and Fig. S10h–j comparing the relative variability in the amplitude of spinal cord-evoked potentials before and after SCI as well as before and after dosing and in Fig. S12a comparing the relative expression of different genes in the untrained group and the 10- to 20-Hz EEMS groups; in Fig. S3c, the proportion of c-Fos positive neurons in the ipsilateral and contralateral regions following stimulation was compared.

In addition, in Fig. S3h, two tailed paired t-test was used to compare the latency of spinal cord response and muscle response induced by muscle stimulation; in Fig. 10d, g, the two tailed paired t-test was used to analyze burst frequency of neurotransmitter flow changes pre and during stimulation, pre and post stimulation, and during and post stimulation in the same mouse.

One-way ANOVA with Bonferroni correction was used for Fig. 2c, d, e, Fig. S5a, b, Figs. 3g, h, o, p, 4c, 5c, f, g, i, j, 6c–e, h, 7d, e, h, 8m, n, and Fig. S12d, e with the untrained group as the comparison group. In addition, one-way ANOVA with Bonferroni correction was used for Figs. 4c, 5c, f, g, i, j, 6c–e, h and 7d, e, h for multiple comparisons between the sham group and the ES onset group and between the sham group and the 10–20 Hz EEMS group.

We used a two-way ANOVA for Fig. S6b–e, and the untrained group was used as the comparison group to conduct multiple comparisons with Tukey's post hoc test. A similar analysis was used for Fig. S6f, with the 10–20 Hz EEMS group as the comparison group.

### Reporting summary
Further information on research design is available in the Nature Portfolio Reporting Summary linked to this article.

## Data availability
The single cell transcriptome sequencing data generated in this study have been deposited in the GEO database under accession code "GSE243038". All other data to support the findings of this study are included in the paper and supplementary information. Source data are provided with this paper. Any additional requests for information can be directed to, and will be fulfilled by, the corresponding authors. Source data are provided with this paper.

## Code availability
The code used to operate the device in this study have been deposited in the Center for Open Science database that can be accessed at https://doi.org/10.17605/OSF.IO/3MG8C.

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

## Acknowledgements

We thank Dr. Zilong Qiu for the gift of *ChAT^cre* knock-in mice. Our study was financially supported by the National Key Research and Development Program of China (2023YFC2412502, 2023YFC2306502 to Y.L.), the National Natural Sciences Foundation of China (82171376, 81971164, 81330026, 81771330 to Y.L.), a project funded by the Priority Academic Program Development of Jiangsu Higher Education Institutions, and the Key Research and Development Plan of Jiangsu Province (BE2023701, BE2018654 to Y.L.), Shanghai Municipal Science and Technology Major Project (No.2018SHZDZX01 to S.W.), ZJ Lab, and Shanghai Center for Brain Science and Brain-Inspired Technology.

## Author contributions

Conceptualization: Y.L., K.Z., and D.Y. methodology: Y.L., K.Z., D.Y., T.Z., Y.Z., Y.N., and S.W.; experimentation: K.Z., D.Y., W.W., H.Z., W.Y., and M.H.; visualization: K.Z., D.Y., W.W., H.Z., P.W., H.R., and W.Y.; supervision: Y.L.; writing—original draft: K.Z., W.W., and H.Z.; writing—review and editing: Y.L., K.Z., W.W., and H.Z.

## Competing interests

The authors declare no competing interests.
