## [Peer Review File · Nature Communications]

Dual electrical stimulation at spinal-muscular interface reconstructs spinal sensorimotor circuits after spinal cord injuryReviewers' Comments:

Reviewer #1:

Remarks to the Author:

My comments on the manuscript by Zhou et al. are limited to the transcriptomic analysis shown in Fig. 11 and Supplementary Fig. 11. I leave it to other Reviewers to evaluate the other data, analysis and interpretations in this study.

Transcriptome analysis (RNA sequencing) was used to identify changes in gene expression following SBES treatment in motoneurons. Notably, the authors used retrograde tracing and manual aspiration of individual neurons to purify the population of interest. I have several major comments on this analysis:

1 - It would be helpful to have more methodological detail about the material collection. They collected $n=3$ replicates from each of the three conditions (sham, untrained, SBES). How many neurons were collected from each mouse? What quality control/assurance was applied to ensure the validity of the data, e.g. expression of appropriate motoneuron marker genes?

2 - The Methods (line 984) states "the adjusted p-value of the genes was set to <0.05 to detect differentially expressed genes" using DESeq2. The phrasing of this sentence is ambiguous but I presume the authors mean that they define DE genes as those for which the adjusted p-value is <0.05 ($FDR < 0.05$). However, the main text (line 479) states "Transcriptome analysis identified 1,345 480 genes for which transcription had been altered (fold change ≥ 2.0 ; $p < 0.05$)". If this analysis used the adjusted p-value, that should be clarified in the main text and the method of FDR correction should be cited (e.g. Benjamini-Hochberg?). If not (i.e. if they used a nominal p-value <0.05), that is inappropriate since it is likely to lead to a substantial number of false positives and to pollute the subsequent pathway analysis.

3 - Fig. 11B - what are the units of the color scale? Is this showing some kind of z-score or normalized expression? How were these genes chosen for display?

Fig. 11C - What is the meaning of "RichFactor"? Is this a measure of enrichment?

Supplementary Fig. 11A - the x-axis is labeled " $-\log_{10}(p)$ ". Does this show a raw p-value, or is it showing the adjusted p-value? If this is showing raw p-values then it suggests that the DE analysis is greatly inflated.

Reviewer #2:

Remarks to the Author:

Comments and replies the standard questions (bulleted):

- What are the noteworthy results?

This study asked the question as to what changes can be identified in terms of TA muscle activity, synaptic properties around TA MNs and gene expression in TA MNs in response to epidural spinal cord stimulation combined with TA muscle stimulation or alone. The paper claims that there has been a unique frequency range of 10-20 Hz seemingly resulting in stronger "positive plasticity" as other frequencies; albeit the data does not allow for this conclusion.

Unsure, clarifications are necessary to understand what was actually done. However, the paper has potential in terms of describing molecular biological responses to combined spinal and muscle stimulation after SCI.

- Will the work be of significance to the field and related fields? How does it compare to the established literature? If the work is not original, please provide relevant references.

Potentially, some aspects of this work are original: TA muscle recording and the type of SC stimulation

electrode and gene sequencing data complementing the combined stimulation of cord and muscle post-SCI. However, first the Authors need to clarify several issues related to data collection.

- Does the work support the conclusions and claims, or is additional evidence needed?

No, cannot conclude anything until clarifications are provided.

- Are there any flaws in the data analysis, interpretation and conclusions? Do these prohibit publication or require revision?

- Is the methodology sound? Does the work meet the expected standards in your field?

- Is there enough detail provided in the methods for the work to be reproduced?

There are major problems with data collection and analysis:

There are contradictions in the methodologies explained and the displayed results. For example, firstly, the application of spinal stimulation prior to injury as shown in Fig 2. The first row of recordings would have not been possible if the spinal stimulation electrodes were implanted at the time of spinalization surgeries as stated on line 638 in the Methods. Please, explain when the implants occurred in order to collect the data shown in the first row.

Secondly, on line 1219, it is stated that SCS signal reached TA muscle after 15 ms, however, the Figures such as Fig. 1, Fig 2 and Fig 2-S1 all show that the latency from the end of the pulse is about 3-6 ms while from the start of the pulse about 8-10 ms. Clarify, how was this 15 ms deducted.

Latency is not defined for the 3 components, early, medium and late responses (ER, MR and LR).

Some of the illustrations show that shorter latency responses are considered ER under one vs. another condition e.g. 30 Hz vs. untrained. In Fig. 2 Suppl. The duration of the ER is different in trained vs untrained. It is critical to identify and explain as well as analyze with more care the latency of these responses. Clarify please, how these responses were categorized, and if the latency was identical when measuring response sizes under different amplitudes. Moreover, this information is critical if the work is to be reproduced.

The sample size of the recordings i.e. single trials, or averages of 10 or 1000 trials per animal, are not listed in Fig.2, nor in Fig2-supl. 1. There is no clear explanation of which cohorts were used for multiple analysis. Summarize the different cohorts of animals and the different Figures in which summative results are shown to indicate if some animals were part of different experimental series.

Equally problematic is that this paper contains several erroneous use of neurophysiological expressions and concepts, for example: Line 67-68: In this system, the conduction time of feedforward and feedback signals in the sensorimotor neural circuit of intact mouse was used as the stimulation interval" – this is unclear. Clarify and explain what is meant under "conduction time". Line 70: spinal cord stimulation of SBES may gain specificity – this is nonsense, clarify. Line 110: elicited spinal cord evoked electromyography (SCEE) in TA muscles

English grammar is also in need of major revisions, e.g. Line 74: this study moves closer toward fully or Line 185: group was not improvement, and so on.

This paper requires major revisions before it can be considered as a decent work for potential publication.

Reviewer #3:

Remarks to the Author:

Zhou and collaborators present a detailed investigation on the biological effects of a dual site electrical stimulation of the lumbar spinal cord and the ankle flexor muscle following spinal cord injury in mice. Overall, there is a fair novelty in combining spinal and muscular stimulation. Although, both interventions have been known for long time, an effective combination has not been readily proposed. Strengths of the study are its multilayered, state-of-the-art readouts on the biological effect of dual site stimulation in combination with treadmill rehabilitation. This includes readouts for electrophysiology, kinematics, force measurements, structural plasticity, various viral tracing techniques and single cell transcriptomics. The clear weakness of the study is its lacking physiological foundation, why a combination of spinal and muscular stimulation should be qualitatively different from either intervention alone, and, why the combined stimulation at 10-20 Hz drastically outperforms any other presented treatment modality.

I suggest the following major revisions and points for clarification.

1) The term 'bidirectional stimulation' seems inaccurate.

Spinal stimulation, as known from previous literature, and shown in figure 1a, mediates an activation of sensory fibres at the entry into the spinal cord. This activation causes a feedforward volley of neural activity onto motor units, that induce muscle contractions and subsequent sensory feedback from muscle spindles. Feedback from muscle spindles is required for the therapeutic action of spinal cord stimulation, as demonstrated in Moraud et al., 2016. Additionally, spinal stimulation also causes an antidromic activity along the sensory fibres, which is considered as a limiting side effect, first described in Formento et al., 2020. At higher current amplitudes, spinal stimulation also directly activates the motor axons, which leads to the early response component in the EMG (compare Capogrosso et al., 2013). Also, electrical muscle stimulation leads to a feedforward contraction of the muscle, with subsequent sensory feedback from muscle spindles. A possible additional activation of sensory fibres is plausible, as electrical fields are typically not sufficiently selective for either motor or sensory axons. Therefore, neither intervention, spinal nor muscular stimulation, should be considered to be specific to either part of the spinal sensorimotor loop.

Following electrical stimulation, both interventions mediate their therapeutic effect through activation of the spinal sensorimotor loop in its natural feed forward direction. Therefore, the concepts of a 'bidirectional stimulation' seems inaccurate. In my opinion a title like 'Dual electrical stimulation at the spinal and muscular interface reconstructs spinal sensorimotor circuits after spinal cord injury' would be more suitable for the paper.

1a) Side notes on the activation of sensory pathways with muscular stimulation

The authors provide no literature reference or experimental proof, that electrical muscle stimulation activates sensory fibres directly, as shown in the conceptual diagram in Figure 1a. In Figure 1j, the authors report a sensory evoked response recorded from the epidural electrode following muscle stimulation. However, this observation does not prove the direct electrical activation of sensory fibres. The epidural response could as well have resulted from muscle spindle feedback, evoked by the electrically induced muscle fibre contraction. No recordings of the epidural recording are shown for Figure 1j. These types of signals are usually artefact prone. Therefore, the reported values in Figure 1j would benefit from presenting some form of raw data for the epidural recordings, and synchronous EMG data to rule out coincident muscle contractions.

2) The physiological foundation for the synergistic effect of spinal and muscular stimulation remains elusive and should be investigated in more detail.

The novelty of the study conceptually boils down to the combination of spinal and muscle stimulation, and its possible therapeutic advantage over either intervention alone. It would be important to highlight the physiological reasons, why the dual stimulation results in such a striking qualitative difference in the therapy.

a) First, since spinal and muscular stimulation act on the same sensorimotor loop, it is possible that the synergistic effect simply results from a coinciding activation of muscular or spinal structures. This interpretation would be supported by the 15ms interval between spinal and muscle stimulation, that corresponds to the reported 15 ms interval for the spinal stimulation induced activation of the muscle (although in figure 1e, the interval for the M response looks more like 8 ms). A coinciding activation of the EMG may simply enhance the EMG amplitude. Could the authors please provide corresponding EMG recordings for paired spinal and muscle stimulation at different interstimulus intervals, ideally in the healthy state and the early stages after spinal cord injury? These insights would possibly explain the enhanced limb flexion, that results in a significantly enhanced training effect over 3 weeks.

b) If the authors have additional ideas how to highlight the physiological differences between spinal-, muscular stimulation, and their combination this will be appreciated. Establishing the physiological

foundations for the difference between the different types of stimulations will be crucial to convince the scientific community, that the dual site stimulation can be as superior as presented in the many readouts throughout figure 2 to 11.

c) Third, it is odd, that spinal stimulation showed no therapeutic effect. This seems discrepant with existing literature. One possibility might be, that the authors used only the stimulation of proximal lumbar segments. The state-of-the-art treatment, would require a stimulation of extensor segments as well (e.g. Asboth et al., 2018). If the authors could comment, why they believe that their spinal stimulation underperforms in comparison to previous literature. This would help external readers to put the findings better into perspective.

d) Fourth, would it be possible that the authors provide supplementary Videos to the article for the first few days of training and the final days of training with no stimulation, spinal stimulation, muscle stimulation and their combination. This would be important to rule out confounding effects. For example, in Figure 3h, there is a pronounced hip flexion for the SCS and MS conditions in comparison to the dual site stimulation. A simple change in the biomechanical placement of the leg on the treadmill could easily confound the results of the study. Therefore, sharing representative Videos for at least the 20 to 40 Hz SCS, the 20 to 40Hz MS and the optimal dual site stimulation seem important.

e) A discussion on the interaction of muscle stimulation and the antidromic response of spinal stimulation (Formento et al., 2020) should be included in the discussion.

I complement the authors on the extraordinary amount of group comparisons and refined experimental readouts.

3) Further minor comments:

a) If the authors could add in the description to the methods, a summary how many animals they used for the study and how they managed to train these large amounts of animals in Figure 2, this would help readers to estimate the feasibility of the study and its reproducibility. For example, how many treadmills were run in parallel? Did this require the attendance of several technical assistants, or was the training fully automatized? How was the bodyweight support adjusted on an individual basis? Was the training blinded?

b) Figure 11f, the y-axis should be converted to 100 percent

c) The majority of the figures deal with readouts from the same experiment, and often the same data from the healthy and sham groups are presented multiple times. Maybe there are ways to combine some of the figures in a more concise way? Similar articles from the journal could serve as a figure template.

d) In the text, multiple times the same statistical test is spelled out and interrupts the reading process. Maybe the authors could find a way, to reduce this text and put it into the methods section and figure legends?

e) Could the authors provide a photo of the implants from Supplementary figure 1, and reference the applied materials and production processes? Possibly, also say whether these implants are available to the scientific community. These questions could facilitate reproducibility of results.

f) In the abstract, the first part of the following sentence sounds like a tautology and should be simplified: Different parameters of electrical stimulation encode diverse characteristics of electrical signals, which leads to various effects on the remodelling of neural circuits, indicating that the stimulation parameters are related to different neural activities.

Citations:

Asboth L, Friedli L, Beauparlant J, Martinez-Gonzalez C, Anil S, Rey E, Baud L, Pidpruzhnykova G, Anderson MA, Shkorbatova P, Batti L, Pagès S, Kreider J, Schneider BL, Barraud Q, Courtine G. Cortico-reticulo-spinal circuit reorganization enables functional recovery after severe spinal cord contusion. *Nat Neurosci.* 2018 Apr;21(4):576-588. doi: 10.1038/s41593-018-0093-5. Epub 2018 Mar 19. PMID: 29556028.

Capogrosso M, Wenger N, Raspopovic S, Musienko P, Beauparlant J, Bassi Luciani L, Courtine G, Micera S. A computational model for epidural electrical stimulation of spinal sensorimotor circuits. *J Neurosci.*

2013 Dec 4;33(49):19326-40. doi: 10.1523/JNEUROSCI.1688-13.2013. PMID: 24305828; PMCID: PMC6618777.

Formento E, Minassian K, Wagner F, Mignardot JB, Le Goff-Mignardot CG, Rowald A, Bloch J, Micera S, Capogrosso M, Courtine G. Electrical spinal cord stimulation must preserve proprioception to enable locomotion in humans with spinal cord injury. *Nat Neurosci*. 2018 Dec;21(12):1728-1741. doi: 10.1038/s41593-018-0262-6. Epub 2018 Oct 31. PMID: 30382196; PMCID: PMC6268129.

Morand EM, Capogrosso M, Formento E, Wenger N, DiGiovanna J, Courtine G, Micera S. Mechanisms Underlying the Neuromodulation of Spinal Circuits for Correcting Gait and Balance Deficits after Spinal Cord Injury. *Neuron*. 2016 Feb 17;89(4):814-28. doi: 10.1016/j.neuron.2016.01.009. Epub 2016 Feb 4. PMID: 26853304.

REVIEWER COMMENTS

Reviewer #1 (Remarks to the Author):

My comments on the manuscript by Zhou et al. are limited to the transcriptomic analysis shown in Fig. 11 and Supplementary Fig. 11. I leave it to other Reviewers to evaluate the other data, analysis and interpretations in this study.

Transcriptome analysis (RNA sequencing) was used to identify changes in gene expression following SBES treatment in motoneurons. Notably, the authors used retrograde tracing and manual aspiration of individual neurons to purify the population of interest. I have several major comments on this analysis:

1 - It would be helpful to have more methodological detail about the material collection. They collected $n=3$ replicates from each of the three conditions (sham, untrained, SBES). How many neurons were collected from each mouse? What quality control/assurance was applied to ensure the validity of the data, e.g. expression of appropriate motoneuron marker genes?

2 - The Methods (line 984) states "the adjusted p-value of the genes was set to <0.05 to detect differentially expressed genes" using DESeq2. The phrasing of this sentence is ambiguous but I presume the authors mean that they define DE genes as those for which the adjusted p-value is <0.05 ($FDR < 0.05$). However, the main text (line 479) states "Transcriptome analysis identified 1,345 480 genes for which transcription had been altered (fold change ≥ 2.0 ; $p < 0.05$ "). If this analysis used the adjusted p-value, that should be clarified in the main text and the method of FDR correction should be cited (e.g. Benjamini-Hochberg?). If not (i.e. if they used a nominal p-value <0.05), that is inappropriate since it is likely to lead to a substantial number of false positives and to pollute the subsequent pathway analysis.

3 - Fig. 11B - what are the units of the color scale? Is this showing some kind of z-score or normalized expression? How were these genes chosen for display?

Fig. 11C - What is the meaning of "RichFactor"? Is this a measure of enrichment?

Supplementary Fig. 11A - the x-axis is labeled " $-\log_{10}(p)$ ". Does this show a raw p-value, or is it showing the adjusted p-value? If this is showing raw p-values then it suggests that the DE analysis is greatly inflated.

Reviewer #2 (Remarks to the Author):

Comments and replies the standard questions (bulleted):

- What are the noteworthy results?

This study asked the question as to what changes can be identified in terms of TA muscle activity, synaptic properties around TA MNs and gene expression in TA MNs in response to epidural spinal cord stimulation combined with TA muscle stimulation or alone. The paper claims that there has been a unique frequency range of 10-20 Hz seemingly resulting in stronger "positive plasticity" as other frequencies; albeit the data does not allow for this conclusion.

Unsure, clarifications are necessary to understand what was actually done. However, the paper has potential in terms of describing molecular biological responses to combined spinal and muscle stimulation after SCI.

- Will the work be of significance to the field and related fields? How does it compare to the established literature? If the work is not original, please provide relevant references. Potentially, some aspects of this work are original: TA muscle recording and the type of SC stimulation electrode and gene sequencing data complementing the combined stimulation of cord and muscle post-SCI. However, first the Authors need to clarify several issues related to data collection.

- Does the work support the conclusions and claims, or is additional evidence needed? No, cannot conclude anything until clarifications are provided.

- Are there any flaws in the data analysis, interpretation and conclusions? Do these prohibit publication or require revision?

- Is the methodology sound? Does the work meet the expected standards in your field?

- Is there enough detail provided in the methods for the work to be reproduced?

There are major problems with data collection and analysis:

There are contradictions in the methodologies explained and the displayed results. For example, firstly, the application of spinal stimulation prior to injury as shown in Fig 2. The first row of recordings would have not been possible if the spinal stimulation electrodes were implanted at the time of spinalization surgeries as stated on line 638 in the Methods. Please, explain when the implants occurred in order to collect the data shown in the first row.

Secondly, on line 1219, it is stated that SCS signal reached TA muscle after 15 ms, however, the Figures such as Fig. 1, Fig 2 and Fig 2-S1 all show that the latency from the end of the pulse is about 3-6 ms while from the start of the pulse about 8-10 ms. Clarify, how was this 15 ms deducted.

Latency is not defined for the 3 components, early, medium and late responses (ER, MR and LR). Some of the illustrations show that shorter latency responses are considered ER under one vs. another condition e.g. 30 Hz vs. untrained. In Fig. 2 Suppl. The duration of the ER is different in trained vs untrained. It is critical to identify and explain as well as analyze with more care the latency of these responses. Clarify please, how these responses were categorized, and if the latency was identical when measuring response sizes under different amplitudes. Moreover, this information is critical if the work is to be reproduced.

The sample size of the recordings i.e. single trials, or averages of 10 or 1000 trials per animal, are not listed in Fig.2, nor in Fig2-supl. 1. There is no clear explanation of which cohorts were used for multiple analysis. Summarize the different cohorts of animals and the different Figures in which summative results are shown to indicate if some animals

were part of different experimental series.

Equally problematic is that this paper contains several erroneous use of neurophysiological expressions and concepts, for example: Line 67-68: In this system, the conduction time of feedforward and feedback signals in the sensorimotor neural circuit of intact mouse was used as the stimulation interval" – this is unclear. Clarify and explain what is meant under "conduction time". Line 70: spinal cord stimulation of SBES may gain specificity – this is nonsense, clarify. Line 110: elicited spinal cord evoked electromyography (SCEE) in TA muscles

English grammar is also in need of major revisions, e.g. Line 74: this study moves closer toward fully or Line 185: group was not improvement, and so on.

This paper requires major revisions before it can be considered as a decent work for potential publication.

Reviewer #3 (Remarks to the Author):

Zhou and collaborators present a detailed investigation on the biological effects of a dual site electrical stimulation of the lumbar spinal cord and the ankle flexor muscle following spinal cord injury in mice. Overall, there is a fair novelty in combining spinal and muscular stimulation. Although, both interventions have been known for long time, an effective combination has not been readily proposed. Strengths of the study are its multilayered, state-of-the-art readouts on the biological effect of dual site stimulation in combination with treadmill rehabilitation. This includes readouts for electrophysiology, kinematics, force measurements, structural plasticity, various viral tracing techniques and single cell transcriptomics. The clear weakness of the study is its lacking physiological foundation, why a combination of spinal and muscular stimulation should be qualitatively different from either intervention alone, and, why the combined stimulation at 10-20 Hz drastically outperforms any other presented treatment modality.

I suggest the following major revisions and points for clarification.

1)The term 'bidirectional stimulation' seems inaccurate.

Spinal stimulation, as known from previous literature, and shown in figure 1a, mediates an activation of sensory fibres at the entry into the spinal cord. This activation causes a feedforward volley of neural activity onto motor units, that induce muscle contractions and subsequent sensory feedback from muscle spindles. Feedback from muscle spindles is required for the therapeutic action of spinal cord stimulation, as demonstrated in Moraud et al., 2016. Additionally, spinal stimulation also causes an antidromic activity along the sensory fibres, which is considered as a limiting side effect, first described in Formento et al., 2020. At higher current amplitudes, spinal stimulation also directly activates the motor axons, which leads to the early response component in the EMG (compare Capogrosso et al., 2013). Also, electrical muscle stimulation leads to a feedforward contraction of the muscle, with subsequent sensory feedback from muscle spindles. A possible additional activation of sensory fibres is plausible, as electrical fields are typically not sufficiently

selective for either motor or sensory axons. Therefore, neither intervention, spinal nor muscular stimulation, should be considered to be specific to either part of the spinal sensorimotor loop.

Following electrical stimulation, both interventions mediate their therapeutic effect through activation of the spinal sensorimotor loop in its natural feed forward direction. Therefore, the concepts of a 'bidirectional stimulation' seems inaccurate. In my opinion a title like 'Dual electrical stimulation at the spinal and muscular interface reconstructs spinal sensorimotor circuits after spinal cord injury' would be more suitable for the paper.

1a) Side notes on the activation of sensory pathways with muscular stimulation

The authors provide no literature reference or experimental proof, that electrical muscle stimulation activates sensory fibres directly, as shown in the conceptual diagram in Figure 1a. In Figure 1j, the authors report a sensory evoked response recorded from the epidural electrode following muscle stimulation. However, this observation does not prove the direct electrical activation of sensory fibres. The epidural response could as well have resulted from muscle spindle feedback, evoked by the electrically induced muscle fibre contraction. No recordings of the epidural recording are shown for Figure 1j. These types of signals are usually artefact prone. Therefore, the reported values in Figure 1j would benefit from presenting some form of raw data for the epidural recordings, and synchronous EMG data to rule out coincident muscle contractions.

2) The physiological foundation for the synergistic effect of spinal and muscular stimulation remains elusive and should be investigated in more detail.

The novelty of the study conceptually boils down to the combination of spinal and muscle stimulation, and its possible therapeutic advantage over either intervention alone. It would be important to highlight the physiological reasons, why the dual stimulation results in such a striking qualitative difference in the therapy.

a) First, since spinal and muscular stimulation act on the same sensorimotor loop, it is possible that the synergistic effect simply results from a coinciding activation of muscular or spinal structures. This interpretation would be supported by the 15ms interval between spinal and muscle stimulation, that corresponds to the reported 15 ms interval for the spinal stimulation induced activation of the muscle (although in figure 1e, the interval for the M response looks more like 8 ms). A coinciding activation of the EMG may simply enhance the EMG amplitude. Could the authors please provide corresponding EMG recordings for paired spinal and muscle stimulation at different interstimulus intervals, ideally in the healthy state and the early stages after spinal cord injury? These insights would possibly explain the enhanced limb flexion, that results in a significantly enhanced training effect over 3 weeks.

b) If the authors have additional ideas how to highlight the physiological differences between spinal-, muscular stimulation, and their combination this will be appreciated. Establishing the physiological foundations for the difference between the different types of

stimulations will be crucial to convince the scientific community, that the dual site stimulation can be as superior as presented in the many readouts throughout figure 2 to 11.

c) Third, it is odd, that spinal stimulation showed no therapeutic effect. This seems discrepant with existing literature. One possibility might be, that the authors used only the stimulation of proximal lumbar segments. The state-of-the-art treatment, would require a stimulation of extensor segments as well (e.g. Asboth et al., 2018). If the authors could comment, why they believe that their spinal stimulation underperforms in comparison to previous literature. This would help external readers to put the findings better into perspective.

d) Fourth, would it be possible that the authors provide supplementary Videos to the article for the first few days of training and the final days of training with no stimulation, spinal stimulation, muscle stimulation and their combination. This would be important to rule out confounding effects. For example, in Figure 3h, there is a pronounced hip flexion for the SCS and MS conditions in comparison to the dual site stimulation. A simple change in the biomechanical placement of the leg on the treadmill could easily confound the results of the study. Therefore, sharing representative Videos for at least the 20 to 40 Hz SCS, the 20 to 40Hz MS and the optimal dual site stimulation seem important.

e) A discussion on the interaction of muscle stimulation and the antidromic response of spinal stimulation (Formento et al., 2020) should be included in the discussion.

I complement the authors on the extraordinary amount of group comparisons and refined experimental readouts.

3) Further minor comments:

a) If the authors could add in the description to the methods, a summary how many animals they used for the study and how they managed to train these large amounts of animals in Figure 2, this would help readers to estimate the feasibility of the study and its reproducibility. For example, how many treadmills where run in parallel? Did this require the attendance of several technical assistants, or was the training fully automatised? How was the bodyweight support adjusted on an individual basis? Was the training blinded?

b) Figure 11f, the y-axis should be converted to 100 percent

c) The majority of the figures deal with readouts from the same experiment, and often the same data from the healthy and sham groups are presented multiple times. Maybe there are ways to combine some of the figures in a more concise way? Similar articles from the journal could serve as a figure template.

d) In the text, multiple times the same statistical test is spelled out and interrupts the reading process. Maybe the authors could find a way, to reduce this text and put it into the methods section and figure legends?

e) Could the authors provide a photo of the implants from Supplementary figure 1, and reference the applied materials and production processes? Possibly, also say whether these implants are available to the scientific community. These questions could facilitate reproducibility of results.

f) In the abstract, the first part of the following sentence sounds like a tautology and

should be simplified: Different parameters of electrical stimulation encode diverse characteristics of electrical signals, which leads to various effects on the remodelling of neural circuits, indicating that the stimulation parameters are related to different neural activities.

Citations:

Asboth L, Friedli L, Beauparlant J, Martinez-Gonzalez C, Anil S, Rey E, Baud L, Pidpruzhnykova G, Anderson MA, Shkorbatova P, Batti L, Pagès S, Kreider J, Schneider BL, Barraud Q, Courtine G. Cortico-reticulo-spinal circuit reorganization enables functional recovery after severe spinal cord contusion. *Nat Neurosci.* 2018 Apr;21(4):576-588. doi: 10.1038/s41593-018-0093-5. Epub 2018 Mar 19. PMID: 29556028.

Capogrosso M, Wenger N, Raspopovic S, Musienko P, Beauparlant J, Bassi Luciani L, Courtine G, Micera S. A computational model for epidural electrical stimulation of spinal sensorimotor circuits. *J Neurosci.* 2013 Dec 4;33(49):19326-40. doi: 10.1523/JNEUROSCI.1688-13.2013. PMID: 24305828; PMCID: PMC6618777.

Formento E, Minassian K, Wagner F, Mignardot JB, Le Goff-Mignardot CG, Rowald A, Bloch J, Micera S, Capogrosso M, Courtine G. Electrical spinal cord stimulation must preserve proprioception to enable locomotion in humans with spinal cord injury. *Nat Neurosci.* 2018 Dec;21(12):1728-1741. doi: 10.1038/s41593-018-0262-6. Epub 2018 Oct 31. PMID: 30382196; PMCID: PMC6268129.

Moraud EM, Capogrosso M, Formento E, Wenger N, DiGiovanna J, Courtine G, Micera S. Mechanisms Underlying the Neuromodulation of Spinal Circuits for Correcting Gait and Balance Deficits after Spinal Cord Injury. *Neuron.* 2016 Feb 17;89(4):814-28. doi: 10.1016/j.neuron.2016.01.009. Epub 2016 Feb 4. PMID: 26853304.

We are grateful for the opportunity to revise our manuscript and wish to thank the reviewers for their encouraging feedback. Their insightful comments and suggestions for further data analyses have helped us to further improve the manuscript by supporting our results and refining our conclusions. Below, we have responded point by point to each of the questions raised. (*Our responses are in blue text*)

Reviewer #1 (Remarks to the Author):

My comments on the manuscript by Zhou et al. are limited to the transcriptomic analysis shown in Fig. 11 and Supplementary Fig. 11. I leave it to other Reviewers to evaluate the other data, analysis and interpretations in this study.

Transcriptome analysis (RNA sequencing) was used to identify changes in gene expression following SBES treatment in motoneurons. Notably, the authors used retrograde tracing and manual aspiration of individual neurons to purify the population of interest. I have several major comments on this analysis:

We thank the reviewer for the comments. We have used the reviewer's suggestions to meticulously revise our manuscript.

- It would be helpful to have more methodological detail about the material collection. They collected n=3 replicates from each of the three conditions (sham, untrained, SBES). How many neurons were collected from each mouse? What quality control/assurance was applied to ensure the validity of the data, e.g. expression of appropriate motoneuron marker genes?

Response: Based on the reviewer's suggestion, we added detail about motor neurons collection and quality control to the Methods section.

“Three weeks after electrode implantation, the TA motoneurons of mice in the sham group (n = 3 mice), untrained group (n = 3 mice), 1-40 Hz DES group (n = 7 mice), 10-20 Hz SCS group (n = 3 mice), 10-20 Hz MS group (n = 3 mice), were labeled by TA intramuscular injection of TRDA, as described previously. One week later, the mice were deeply anesthetized and fresh intact lumbar spinal cord tissue was collected. The freshly isolated spinal cord was immobilized in an AGAR block “V” shaped groove and sliced on a Vibratome (Leica, Germany) into 200- to 300- μ m-thick coronal sections under ice bath and oxygen conditions. The spinal sections were transferred to a patch-clamp bath for continuous oxygenation. TRDA-labeled motoneurons were aspirated under a fluorescence microscope (Zeiss) with a glass electrode at negative pressure using a micro-operating system (Eppendorf, Germany). No less than 50 motoneurons from each mouse were aspirated for single-cell transcriptome sequencing. TA motor neurons were placed in single cell lysis solution

(Genewiz, China).” (Pages 54-55, lines 1117-1130)

“In study, when the target neuron is isolated by this aspiration method, it is inevitable that other types of cells will be aspirated. In order to ensure that the isolated cells mainly contained motor neurons, motor neuron specific expression genes, including ChAT, Isl1, and Isl2, were detected in the samples. The results showed that ChAT, Isl1, and Isl2 genes stably expressed in each group, which indicated that the method of motor neuron aspiration was reliable and ensured the validity of the data.” (Page 56, lines 1145-1150)

- The Methods (line 984) states "the adjusted p-value of the genes was set to <0.05 to detect differentially expressed genes" using DESeq2. The phrasing of this sentence is ambiguous but I presume the authors mean that they define DE genes as those for which the adjusted p-value is <0.05 (FDR<0.05). However, the main text (line 479) states "Transcriptome analysis identified 1,345 genes for which transcription had been altered (fold change ≥ 2.0 ; $p < 0.05$)". If this analysis used the adjusted p-value, that should be clarified in the main text and the method of FDR correction should be cited (e.g. Benjamini-Hochberg?). If not (i.e. if they used a nominal p-value <0.05), that is inappropriate since it is likely to lead to a substantial number of false positives and to pollute the subsequent pathway analysis.

Response: We greatly appreciate your valuable suggestions. We completely agree with the point that you make above. In the original analysis, nominal p -values were used to detect differentially expressed genes, which, as you say, is inappropriate and can lead to a substantial number of false positives, thus confounding any subsequent pathway analysis. Therefore, to elucidate the alterations in motor neuron gene expression following training with different electrical stimulation systems, we employed Smart-seq2 sequencing on the motor neuron once again (including sham, untrained, 10-20 Hz MS, 10-20 Hz SCS, ineffective DES, and effective DES groups). we used fold change ≥ 1.5 , and a false discovery rate (FDR) ≤ 0.1 (the adjustment methods include the Bonferroni correction ("bonferroni") in which the p -values are multiplied by the number of comparisons. Less conservative corrections are also included by Holm (1979) ("holm"), Hochberg (1988) ("hochberg"), Hommel (1988) ("hommel"), Benjamini & Hochberg (1995) ("BH" or its alias "FDR"), and Benjamini & Yekutieli (2001) ("BY"), respectively. A pass-through option ("none") is also included. The set of methods are contained in the p-adjust. Methods vector for the benefit of methods that need to have the method as an option and pass it on to p-adjust) as the new detecting criteria for differential genes (effective DES vs MS, effective DES vs SCS, effective DES vs Untrained, and effective DES vs ineffective DES). We modified the manuscript accordingly. (Page 24, lines 513-516 and page 56, line 1151-1154)

- Fig. 11B - what are the units of the color scale? Is this showing some kind of z-score or normalized expression? How were these genes chosen for display?

Response: We now realize this created some confusion in the original manuscript. The color scale indicates normalized gene expression. In the revised manuscript, we have explained how the normalized expression of each gene was calculated. The FPKM (FPKM, expected number of Fragments Per Kilobase of transcript sequence per Millions base pairs sequenced) value of the gene was read and \log_{10} transformed. We then obtained the $\log_{10}(\text{FPKM} + 1)$ value for each gene. Normalized gene expression was then calculated by subtracting the mean value of $\log_{10}(\text{FPKM} + 1)$ for all genes in the group from the $\log_{10}(\text{FPKM} + 1)$ value for that particular gene. Negative values for normalized gene expression indicate low expression of that gene, and positive values indicate high expression, as indicated by the different colors (red indicates high expression and blue indicates low expression). The data in the figure show the normalized expression of the gene. In the revised manuscript, we have opted to present the up-regulation and down-regulation of gene expression using z-scores as a measure (See Figure 11).

We have added details about these to the Materials and Methods as follows:

“Raw count matrix for differential analysis was obtained by transcriptome sequencing (HTSeq software). A threshold value of Fold Change ≥ 1.5 and FDR ≤ 0.1 was set to screen differential genes (effective DES vs MS, effective DES vs SCS, effective DES vs untrained, and effective DES vs ineffective DES). GO enrichment analysis was performed on all differentially expressed genes, and three functional classifications were obtained: molecular function (MF), cell component (CC) and biological process (BP). Each classification contains a series of GO terms, and each GO term contains a series of differential genes. Recent studies have revealed that genes related to Neuron axon development, Synaptic function, Inflammation and apoptosis are involved in the regulation of neuronal plasticity (Gunnar H. D. Poplawski et al., 2020). Furthermore, our study shows that specific frequency of DES enhances the plasticity of motoneurons. These results suggest that genes related to Neuron axon development, Synaptic function, Inflammation and apoptosis may be involved in the regulation of motoneuron plasticity. Therefore, we defined three functional terms: Neuron axon development, Synaptic function, Inflammation and apoptosis; Subsequently, GO terms related to these three functional terms were screened, and the differential genes under GO terms were listed for display.” (Page 56, lines 1151-1164)

Fig. 11C - What is the meaning of "RichFactor"? Is this a measure of enrichment?

Response: In Figure 11C of original manuscript, RichFactor is the ratio of the number of differentially expressed genes annotated in the pathway term relative to the number

of all genes annotated in that pathway term. The larger the RichFactor value, the greater the degree of enrichment. Indeed, RichFactor is a measure of enrichment. The revised manuscript replaces RichFactor with GeneRatio for enhanced clarity of comprehension (See Figure 11-figure supplement 2B).

Supplementary Fig. 11A - the x-axis is labeled "-log10(p)". Does this show a raw p-value, or is it showing the adjusted p-value? If this is showing raw p-values then it suggests that the DE analysis is greatly inflated.

Response: We thank the reviewer for this critical information. We apologize for our inappropriate data analysis. The raw p -values shown in the original Supplementary Figure 11A indicate, as you said, that the differential expression analysis has been greatly inflated. Therefore, we performed single-cell transcriptome sequencing of the motor neurons once again and analyzed the sequencing results, using fold change ≥ 1.5 , FDR ≤ 0.1 as the new detecting criteria for differentially expressed genes in the revised manuscript (See Figure 11-figure supplement 1).

Reviewer #2 (Remarks to the Author):

Comments and replies the standard questions (bulleted):

• What are the noteworthy results?

This study asked the question as to what changes can be identified in terms of TA muscle activity, synaptic properties around TA MNs and gene expression in TA MNs in response to epidural spinal cord stimulation combined with TA muscle stimulation or alone. The paper claims that there has been a unique frequency range of 10-20 Hz seemingly resulting in stronger “positive plasticity” as other frequencies; albeit the data does not allow for this conclusion.

Unsure, clarifications are necessary to understand what was actually done. However, the paper has potential in terms of describing molecular biological responses to combined spinal and muscle stimulation after SCI.

• Will the work be of significance to the field and related fields? How does it compare to the established literature? If the work is not original, please provide relevant references.

Potentially, some aspects of this work are original: TA muscle recording and the type of SC stimulation electrode and gene sequencing data complementing the combined stimulation of cord and muscle post-SCI. However, first the Authors need to clarify several issues related to data collection.

• Does the work support the conclusions and claims, or is additional evidence needed?

No, cannot conclude anything until clarifications are provided.

• Are there any flaws in the data analysis, interpretation and conclusions? Do these prohibit publication or require revision?

• Is the methodology sound? Does the work meet the expected standards in your field?

• Is there enough detail provided in the methods for the work to be reproduced?

We thank the reviewer for the comments. Based on the reviewer’s suggestions, we have meticulously revised our manuscript.

There are major problems with data collection and analysis:

There are contradictions in the methodologies explained and the displayed results. For example, firstly, the application of spinal stimulation prior to injury as shown in Fig 2. The first row of recordings would have not been possible if the spinal stimulation electrodes were implanted at the time of spinalization surgeries as stated on line 638 in the Methods. Please, explain when the implants occurred in

order to collect the data shown in the first row.

Response: We now realize that we did not adequately clarify this in the original manuscript, which confused the reviewer. Actually, the data in the first row were recorded 29 days after the electrodes were implanted in the sham group. Figure 2 has been revised to better clarify the data. This experimental design includes the sham group, untrained group, 1- to 40- Hz DES (dual electrical stimulation) group, 1- to 40- Hz MS group, 1- to 40- Hz SCS group and 1- to 40- Hz SCShc group (n = 8 mice per group). Electrical stimulation onset represents the point at which electrical stimulation training begins (7 days after electrode implantation). In mice of the sham group, the electrodes were implanted during the sham operation, but the electrical stimulation training was not performed; In mice of the untrained group, electrodes were implanted in the spinal cord during the T9 segment transection surgery to avoid secondary damage due to electrode implantation, but the electrical stimulation training was not performed. Mice in 1- to 40- Hz DES, 1- to 40- Hz MS, 1- to 40- Hz SCS, and 1- to 40- Hz SCShc groups were implanted with stimulating electrodes during the T9 segment transection surgery, and those mice were trained with electrical stimulation 7 days later. The mice underwent electrical stimulation training for 3 weeks. SCEP was recorded at 7 and 29 days after electrode implantation in all experimental groups. All mice were implanted with the stimulation electrodes on the same day (Figure 2 and Page 41, lines 817-827).

Secondly, on line 1219, it is stated that SCS signal reached TA muscle after 15 ms, however, the Figures such as Fig. 1, Fig 2 and Fig 2-S1 all show that the latency from the end of the pulse is about 3-6 ms while from the start of the pulse about 8-10 ms. Clarify, how was this 15 ms deducted.

Response: Thank you; we realize we did not accurately describe this latency. Line 1219 of the original manuscript, “it is stated that SCS signal reached TA muscle after 15 ms”, conveys the wrong meaning. The sentence has been revised as follows: “The duration from the start of the spinal pulse to the end of the muscle response was no more than 15 ms, and the interval between spinal and muscle stimulation was determined to be 15 ms in order to prevent overlap of stimulus effects.” (Page 67, lines 1433-1436)

In the revised manuscript, we have also explained how this 15 ms latency was determined. In a dual electrical stimulation system, SCS and MS have a timing-sequence stimulation pattern. For example, first the SCS was administered, second the MS was administered, then the SCS was administered again, and so on. The stimulus frequency range we selected was 1-40 Hz, to prevent overlap of the dual stimulation effects, the sum of the time from the beginning of the spinal pulse to the end of the muscle response and the time from the beginning of the muscle pulse to the end of the spinal response was no more than 25 ms (1/40 s). Then, under normal physiological conditions, we calculated that the time from the start of spinal pulse to the end of

muscle response was 13.34 ± 0.66 ms (Figure 1-figure supplement 4A), and the time from the start of muscle pulse to the end of spinal response was 8.35 ± 0.49 ms (Figure 1-figure supplement 4B). Therefore, the interval between spinal stimulation and muscle stimulation was set 15 ms to ensure that the spinal-evoked muscle response was fully released.

In addition, according to the reviewer's comment, we reanalyzed the data in Figure 1, Figure 2 and Figure 2-figure supplement 1. More precisely, in the sham group, the latency from the end of the spinal pulse was 2.83 ± 0.22 ms (Figure 1-figure supplement 2B) whereas from the start of the spinal pulse was 5.83 ± 0.22 ms (Figure 1-figure supplement 2C).

Latency is not defined for the 3 components, early, medium and late responses (ER, MR and LR). Some of the illustrations show that shorter latency responses are considered ER under one vs. another condition e.g. 30 Hz vs. untrained. In Fig. 2 Suppl. The duration of the ER is different in trained vs untrained. It is critical to identify and explain as well as analyze with more care the latency of these responses. Clarify please, how these responses were categorized, and if the latency was identical when measuring response sizes under different amplitudes. Moreover, this information is critical if the work is to be reproduced.

Response: Thanks for pointing out that the original manuscript did not provide adequate definitions for these terms. According to the literature, spinal stimulation directly activates motor axons, which leads to the ER component of the SCEP. Spinal stimulation also activates the interneurons – motor neurons – muscle, which leads to the MR component of the SCEP. Spinal stimulation activates sensory afferent fibers – the interneurons – motor neurons – muscle, which leads to the LR component of the SCEP (Capogrosso M et al., 2013; Tresch MC et al., 2000). Based on this evidence and our analysis of data obtained in the present study (Figure 1-figure supplement 2A), we defined the latencies associated with the early, medium, and late responses as 4–7 ms, 7–10 ms and 10–15 ms, respectively.

In response to the reviewer's question, we also reanalyzed our experimental results, which revealed was no statistically significant difference in the latency of ER in the sham, 1- 40 Hz DES, 1-40 Hz SCS, 1-40 Hz MS and 1-40 Hz SCShc groups compared with the untrained group, and almost all the early latency responses occurred within 4–7 ms (Figure 2-figure supplement 1A). This indicates that the latency of ER did not change under different experimental conditions.

Following the reviewer's comments, we collected SCEP data for different experimental groups and analyzed the duration of ER. The results showed that there was no statistically significant difference in the duration of ER in the 1-40 Hz DES, 1-40 Hz SCS, 1-40 Hz MS and 1-40 Hz SCShc groups compared with the untrained group (Figure 2-figure supplement 1B). Compared with the untrained group, the absolute value of the duration of ER in the sham group and 10-20 Hz DES groups was

lower, whereas the duration of ER did not differ significantly among the three groups (untrained group versus sham group, $p=0.1483$; untrained group versus 10-20 Hz group, $p=0.1434$) (Figure 2-figure supplement 1B). This indicates that the duration of ER was not affected by electrical stimulation training.

In addition, we analyzed the latency of ER under stimuli of differing amplitude. The latency did not change significantly under different amplitude stimulation conditions, although the absolute value of ER latency decreased gradually with an increase of stimulus intensity (Figure 1-figure supplement 2D,E).

The sample size of the recordings i.e. single trials, or averages of 10 or 1000 trials per animal, are not listed in Fig.2, nor in Fig2-supl. 1. There is no clear explanation of which cohorts were used for multiple analysis. Summarize the different cohorts of animals and the different Figures in which summative results are shown to indicate if some animals were part of different experimental series.

Response: We thank the reviewer for the comments. We added the sample size of the recordings and the number of experiments per animal in the legend of Figure 2 and Figure 2-figure supplement 1. Each experimental group included eight mice, each of which underwent recording with a continuous electrical signal (each electrical signal included 50 spinal cord evoked EMG). The waveform in the figure was the average (SD) of 50 stimuli.

We added the cohort information for multiple comparisons in the Materials and Methods. Figure 2-figure supplement 1 in the original manuscript was integrated into Figure 2 in the revised manuscript. The Bonferroni multiple comparison test was used to compare the untrained group with each experimental group. In addition, in Figure 2E, the 10-20 Hz DES groups were compared with the sham group by Bonferroni's multiple comparison test (Figure 2 and Page 58, lines 1209-1215).

We have added detailed information describing the animal experiments in the Materials and Methods. In Figure 2, mice in the sham, untrained, 1-40 Hz DES, 1-40 Hz MS, 1-40 Hz SCS, and 1-40 Hz SCS_{hc} groups were implanted with electrodes on the same day. After one week, SCEP was recorded for mice in all groups, followed by three weeks of electrical stimulation training in the electrical stimulation group only. At 4 weeks after electrode implantation, SCEP was recorded for all groups. In Figure 2C-E, all data were collected for analysis at 29 days after electrode implantation. The sham and untrained groups were part of different experiment series (DES, SCS, MS, and SCS_{hc} groups). Each experimental group included data from electrical stimulation onset (7 days after electrode implantation) and 29 days after electrode implantation, which belonged to different time points in the same group of mice (Page 41, lines 817-827).

Equally problematic is that this paper contains several erroneous use of

neurophysiological expressions and concepts, for example: Line 67-68: In this system, the conduction time of feedforward and feedback signals in the sensorimotor neural circuit of intact mouse was used as the stimulation interval” – this is unclear. Clarify and explain what is meant under “conduction time”.

Response: Thank you for pointing out certain incorrect expressions. We checked the entire manuscript carefully and did our best to correct these inaccuracies. For example: Line 67-68: “In this system, the conduction time of feedforward and feedback signals in the sensorimotor neural circuit of intact mouse was used as the stimulation interval”. In local neural circuits of intact mice, peripheral sensory information feedback involves spinal interneurons and motor neurons. Interneurons and motor neurons are involved in controlling muscle contractions, which then control peripheral muscle contractions, generating feedback signals, and so on (Arber S et al., 2012, 2018). To simulate the operation of sensorimotor neural circuits in mice with spinal cord injury, SCS was first performed to simulate motor commands. The spinal cord-evoked response in muscle ended after 15 ms, and then the MS was administered to simulate sensory feedback signals, and the muscle-evoked response in the spinal cord ended after 10 ms. In the dual stimulation system, the conduction time was defined as the sum of the signal conduction time of spinal-to-muscle and muscle-to-spinal.

To clarify this concept, we modified the text as follows: “In this system, in order to mimic the operation of sensorimotor neural circuits, the spinal cord was first given electrical stimulation signals to simulate the motor commands. Afterward, the muscle was then given electrical stimulation signals to simulate sensory feedback signals. To prevent overlap of the dual stimulation effects, MS was administered after the end of the spinal cord-evoked response. Therefore, the duration from the start of the spinal pulse to the end of the muscle response is used as the stimulation interval.” (Page 4; lines 70-75)

Line 70: spinal cord stimulation of SBES may gain specificity – this is nonsense, clarify.

Response: We agree with the reviewer in that this sentence was not correct. Actually, we want to express the meaning “spinal cord electrical stimulation paired with muscle electrical stimulation can more accurately modulate the target circuit”. It has been reported that electrical stimulation of the nervous system can augment the plasticity of spared or latent circuits through focal modulation after spinal cord injury. Pairing stimulation of two parts of a spared circuit can target modulation more specifically to the intended circuit (Yang Q et al., 2019; Mishra AM et al., 2017; Pal A et al., 2022).

For clarification, changes have been made in the text as follows: “As a kind of field potential stimulation, spinal cord electrical stimulation has a wide range of activation. Electrical stimulation of the anterior tibial muscle activates sensory

feedback pathways associated with TA muscles. By carrying out DES at the spinal and muscle interface, we can more specifically target the sensorimotor circuit for modulation.” (Page 4; lines 76-80)

Line 110: elicited spinal cord evoked electromyography (SCEE) in TA muscles

Response: Thank you for pointing out this phrase. We apologize for the confusion. To avoid misunderstanding, we clarified the text as follows: “The SCS was administered to evoke a response in the TA muscle. We define this evoked response recorded in the muscle as spinal cord–evoked potential (SCEP).” (Page 6; lines 118-120)

English grammar is also in need of major revisions, e.g. Line 74: this study moves closer toward fully or Line 185: group was not improvement, and so on.

This paper requires major revisions before it can be considered as a decent work for potential publication.

Response: We thank the reviewer for the comment. We employed a native English-speaker to correct the grammatical mistakes in this manuscript. The sentence in question now reads as follows:

“Briefly, this study leads to a deeper understanding of the role of neuromodulation in neural circuit reassembly and indicates a potential framework for the clinical application of neural trauma treatment.” (Page 4, lines 83-86)

“But under other frequencies of electrical stimulation, compared with the SCS, MS, and SCShc groups, the motor function of the hindlimbs of mice was not improved in the DES group” (Page 12; lines 242-244)

Reviewer #3 (Remarks to the Author):

Zhou and collaborators present a detailed investigation on the biological effects of a dual site electrical stimulation of the lumbar spinal cord and the ankle flexor muscle following spinal cord injury in mice. Overall, there is a fair novelty in combining spinal and muscular stimulation. Although, both interventions have been known for long time, an effective combination has not been readily proposed. Strengths of the study are its multilayered, state-of-the-art readouts on the biological effect of dual site stimulation in combination with treadmill rehabilitation. This includes readouts for electrophysiology, kinematics, force measurements, structural plasticity, various viral tracing techniques and single cell transcriptomics. The clear weakness of the study is its lacking physiological foundation, why a combination of spinal and muscular stimulation should be qualitatively different from either intervention alone, and, why the combined stimulation at 10-20 Hz drastically outperforms any other presented treatment modality.

We thank the reviewer for the positive comments on our manuscript. We have made meticulous revision based on the suggestions.

I suggest the following major revisions and points for clarification.

1)The term ‘bidirectional stimulation’ seems inaccurate.

Spinal stimulation, as known from previous literature, and shown in figure 1a, mediates an activation of sensory fibres at the entry into the spinal cord. This activation causes a feedforward volley of neural activity onto motor units, that induce muscle contractions and subsequent sensory feedback from muscle spindles. Feedback from muscle spindles is required for the therapeutic action of spinal cord stimulation, as demonstrated in Moraud et al., 2016. Additionally, spinal stimulation also causes an antidromic activity along the sensory fibres, which is considered as a limiting side effect, first described in Formento et al., 2020. At higher current amplitudes, spinal stimulation also directly activates the motor axons, which leads to the early response component in the EMG (compare Capogrosso et al., 2013). Also, electrical muscle stimulation leads to a feedforward contraction of the muscle, with subsequent sensory feedback from muscle spindles. A possible additional activation of sensory fibres is plausible, as electrical fields are typically not sufficiently selective for either motor or sensory axons. Therefore, neither intervention, spinal nor muscular stimulation, should be considered to be specific to either part of the spinal sensorimotor loop.

Following electrical stimulation, both interventions mediate their therapeutic effect through activation of the spinal sensorimotor loop in its natural feed forward direction. Therefore, the concepts of a ‘bidirectional stimulation’ seems inaccurate. In my opinion a title like ‘Dual electrical stimulation at the spinal and muscular interface reconstructs spinal sensorimotor circuits after spinal cord injury’ would be more suitable for the paper.

Response: We are grateful for the reviewer’s thoughtful suggestion. In the spinal sensorimotor loop, neural signals always run in a fixed direction, so the term “bidirectional electrical stimulation” is indeed not appropriate. Therefore, following the suggestion of the reviewer, we have revised the manuscript with the title “Dual electrical stimulation at the spinal and muscular interface reconstructs spinal sensorimotor circuits after spinal cord injury”.

*1a) Side notes on the activation of sensory pathways with muscular stimulation
The authors provide no literature reference or experimental proof, that electrical muscle stimulation activates sensory fibres directly, as shown in the conceptual diagram in Figure 1a. In Figure 1j, the authors report a sensory evoked response recorded from the epidural electrode following muscle stimulation. However, this observation does not prove the direct electrical activation of sensory fibres. The epidural response could as well have resulted from muscle spindle feedback, evoked by the electrically induced muscle fibre contraction. No recordings of the epidural recording are shown for Figure 1j. These types of signals are usually artefact prone. Therefore, the reported values in Figure 1j would benefit from presenting some form of raw data for the epidural recordings, and synchronous EMG data to rule out coincident muscle contractions.*

Response: Thank you very much for this comment. We provide several lines of evidence to address these questions.

First, we implanted stimulation electrodes in the TA muscles of the left and right hindlimbs of anesthetized mice. Then electrical stimulation was given only to the TA muscle of the right hindlimb. The whole process was stimulated for 90 minutes, with rest for 90 minutes (Figure 1-figure supplement 3A). Mice were then perfused and spinal cord L2–L4 segments were used for frozen sectioning. We found that there was a statistically significant increase in c-Fos expression in the ipsilateral and contralateral spinal cord after muscle stimulation, as demonstrated by immunofluorescence staining (Figure 1-figure supplement 3B). The results showed that the proportion of c-Fos⁺ neurons in the dorsal horn of the ipsilateral spinal cord was significantly greater than that on the contralateral side (Figure 1-figure supplement 3C). This suggests that electrical stimulation of muscle can directly activate sensory fibers, which in turn activate the dorsal horn cells of the spinal cord.

Second, we injected AAV-GCAMP6F virus into the dorsal horn of the spinal cord of mice to facilitate GCAMP6F expression in dorsal horn neurons. At the same time, the optical fiber was implanted into the dorsal corner of the spinal cord and fixed with dental cement. At 10 days after surgery, the right hindlimb of anesthetized mice was implanted with electrodes (Figure 1-figure supplement 3D). At the time of TA muscle stimulation, fluorescence changes in response to increased calcium signaling in dorsal horn neurons of the spinal cord were recorded simultaneously. We found that 40 consecutive MS pulses could stably enhance calcium signaling in dorsal horn neurons (Figure 1-figure supplement 3E). This result indicates that muscle stimulation activates sensory fibers, which in turn activate dorsal horn neurons.

Finally, we recorded evoked signals in the spinal cord and muscle during muscle stimulation (Figure 1-figure supplement 3F). The latencies of evoked signals recorded in the spinal cord and the muscle were measured (Figure 1-figure supplement 3G). The results showed that, in the same individual after muscle stimulation, evoked signals were first evident in the spinal cord, followed by evoked EMG in the muscle (Figure 1-figure supplement 3H). This suggests that the epidural reaction could not have resulted from muscle spindle feedback evoked by electrically induced muscle fiber contraction.

2)The physiological foundation for the synergistic effect of spinal and muscular stimulation remains elusive and should be investigated in more detail.

The novelty of the study conceptually boils down to the combination of spinal and muscle stimulation, and its possible therapeutic advantage over either intervention alone. It would be important to highlight the physiological reasons, why the dual stimulation results in such a striking qualitative difference in the therapy.

a) First, since spinal and muscular stimulation act on the same sensorimotor loop, it is possible that the synergistic effect simply results from a coinciding activation of muscular or spinal structures. This interpretation would be supported by the 15ms interval between spinal and muscle stimulation, that corresponds to the reported 15 ms interval for the spinal stimulation induced activation of the muscle (although in figure 1e, the interval for the M response looks more like 8 ms). A coinciding activation of the EMG may simply enhance the EMG amplitude. Could the authors please provide corresponding EMG recordings for paired spinal and muscle stimulation at different interstimulus intervals, ideally in the healthy state and the early stages after spinal cord injury? These insights would possibly explain the enhanced limb flexion, that results in a significantly enhanced training effect over 3 weeks.

Response: Thank you for pointing out this potential confusion. First, SCS was administered to the spinal cord. The time from the start of the spinal pulse to the end of the muscle response was 13.34 ± 0.66 ms (Figure 1-figure supplement 4A).

Afterward, feedback electrical stimulation was administered to the muscles. The interval between SCS and MS was set at 15 ms to ensure that the spinal cord–evoked muscle response was fully released. Therefore, SCS and MS were not performed at the same time, and the spinal cord and muscle structure were not activated at the same time. In addition, SCS evoked muscle contraction, generated EMG of approximately 0.18 mV, but this level of EMG may not be sufficient to elicit a spinal response.

We wish to further explain how this 15-ms interval was chosen. In our DES system, SCS and MS had a timing-sequence stimulation pattern. For example, the SCS was administered first, the MS was administered second, and then the SCS was administered again, and so on. The stimulus frequency range we selected was 1–40 Hz to prevent overlap of the dual stimulation effects; the sum of the time from the beginning of the spinal pulse to the end of the muscle response and the time from the beginning of the muscle pulse to the end of the spinal response was no more than 25 ms (1/40 s). Then, under normal physiological conditions, we calculated that the time from the start of the spinal pulse to the end of the muscle response was 13.34 ± 0.66 ms (Figure 1-figure supplement 4A), and the time from the start of muscle pulse to the end of spinal response was 8.35 ± 0.49 ms (Figure 1-figure supplement 4B). Therefore, the interval between spinal stimulation and muscle stimulation was set 15 ms to ensure that the spinal cord-evoked muscle response was fully realized.

In addition, according to the reviewer's opinion, we reanalyzed the data in Figure 1, Figure 2 and Figure 2-figure supplement 1. More precisely, in the sham group, the latency from the end of the spinal pulse was 2.83 ± 0.22 ms in the sham group (Figure 1-figure supplement 2B) whereas from the start of the spinal pulse it was approximately 5.83 ± 0.22 ms (Figure 1-figure supplement 2C).

As suggested by the reviewer, we now provide corresponding EMG recordings for paired spinal and muscle stimulation at different interstimulus intervals. The stimulation interval between spinal cord stimulation and muscle stimulation was set to 20 ms, 25 ms, and 30 ms, respectively. Each stimulation interval was used for 10–20 Hz DES. The results showed no change in EMG amplitude after paired spinal cord and muscle stimulation at different intervals compared with the 15 ms stimulation interval group (Figure 3N and Figure 3-figure supplement 2).

b) If the authors have additional ideas how to highlight the physiological differences between spinal-, muscular stimulation, and their combination this will be appreciated. Establishing the physiological foundations for the difference between the different types of stimulations will be crucial to convince the scientific community, that the dual site stimulation can be as superior as presented in the many readouts throughout figure 2 to 11.

Response: We thank the reviewer for the suggestions. It was recently reported that the balance of excitatory and inhibitory neurotransmitters in the spinal cord is crucial for the recovery of motor function after SCI (Bertels H et al., 2022). After incomplete

SCI, spinal cord inhibitory interneurons limit functional recovery (Chen B et al., 2018; Courtine G et al., 2018). By increasing the excitability of the spinal cord, individuals with incomplete spinal cord injury can regain autonomic motor function (Angeli CA et al., 2014). Therefore, increasing the excitability of the spinal cord and reducing inhibition of inhibitory neurons in the spinal cord may contribute to motor function recovery following a spinal cord injury. To explore the physiological mechanism by which 10- to 20- Hz DES at the spinal-muscle interface promotes neural circuit remodeling and functional improvement, we recorded the real-time in vivo flow of Glu and GABA in the spinal cord of mice before, during and after stimulation through neurotransmitter probes. After 29 days of stimulation training, viruses carrying Glu and GABA fluorescent probes were injected into the intermediate-dorsal region of the L4 spinal cord of mice in the 10-20 Hz SCS, 10-20 Hz MS, and 10-20 Hz DES groups (Figure 12A). One week later, the fluorescence changes of Glu and GABA neurotransmitters in the spinal cord of the SCS, the MS, and DES groups were detected in the awake state. The mice in an awakened state were subjected to a 30 minutes signal collection period prior to the commencement of stimulation. The stimulation phase lasted for a duration of 90 minutes, with each stimulation lasting for 15 minutes followed by a rest period of 10 minutes. Subsequent to the conclusion of the stimulation phase, signal recording was conducted for an additional 30 minutes. After recording the fluorescence probe, immunohistochemistry was performed on the L4 segment of the spinal cord to assess viral infection in neurons (Figure 12A,C, and F).

The results from the glutamate probe demonstrated a significant increase in burst frequency of glutamate flow during and post DES (10-20 Hz) compared to pre-DES (10-20 Hz), with higher burst frequency observed during stimulation than after (Figure 12B, D). On the contrary, there was no significant increase in burst frequency of glutamate flow during and post SCS (10-20 Hz) and MS (10-20 Hz) compared to before stimulation (Figure 12D and Figure 12-figure supplement 1A, B). Furthermore, it was found that both DES (10-20 Hz) and SCS (10-20 Hz) induced a higher burst frequency of glutamate flow compared to MS (10-20 Hz) (Figure 12D and Figure 12-figure supplement 1A, B).

The results of the GABA probe demonstrated a significant decrease in burst frequency of GABA flow during and post DES (10-20 Hz) compared to pre-DES (10-20 Hz) (Figure 12E, G). Similarly, the burst frequency of GABA flow during SCS (10-20 Hz) was significantly lower than pre-SCS (10-20 Hz), while no significant difference was observed in the burst frequency of GABA flow after cessation of stimulation compared to pre-SCS (10-20 Hz) levels (Figure 12G and Figure-figure supplement 1C, D). In addition, there was no statistically significant difference in the frequency of GABA stream bursts during and after MS (10-20 Hz) compared to pre-MS (10-20 Hz) (Figure 12G and Figure-figure supplement 1C, D).

In summary, both DES (10-20 Hz) and SCS (10-20 Hz) exhibited an excitatory effect on spinal neurons surpassing that of MS (10-20 Hz). Notably, following discontinuation of DES (10-20 Hz), its excitatory effects on the spinal cord persisted for a certain duration. The DES (10-20 Hz) and SCS (10-20 Hz) interventions both

effectively mitigate the inhibition of spinal neurons, surpassing the efficacy of MS (10-20 Hz). Furthermore, this reduction in inhibition persists for a considerable duration even after discontinuation of DES (10-20 Hz). Therefore, only 10-20 Hz DES can effectively maintain the excitability of spinal neurons post-stimulation cessation, while concurrently mitigating the inhibition of spinal neurons. Consequently, this facilitates easier activation of the spinal neural circuit and restoration of hind limb motor function, aligning with previously reported findings (Courtine, 2018; Chen B., 2018; Angeli CA., 2014).

c)Third, it is odd, that spinal stimulation showed no therapeutic effect. This seems discrepant with existing literature. One possibility might be, that the authors used only the stimulation of proximal lumbar segments. The state-of-the-art treatment, would require a stimulation of extensor segments as well (e.g. Asboth et al., 2018). If the authors could comment, why they believe that their spinal stimulation underperforms in comparison to previous literature. This would help external readers to put the findings better into perspective.

Response: We agree with the reviewer's points. Indeed, as stated by the reviewer, we only stimulated the proximal lumbar segments and only activated the motoneuron pool that innervates the flexor muscles. It is well known that a complete movement requires the synergistic activation of the extensor and flexor muscles. Therefore, the therapeutic effect of SCS in this study was not as superior as reported in the literature. We now discuss this point in the revised manuscript (Page 32, lines 677-686).

“Recent studies have shown recovery of motor function in paralyzed hindlimbs in spinal cord injury rats after continuous epidural electrical stimulation with L2 and S1 electrodes (Asboth L et al., 2018; van den Brand R et al., 2012; Bonizzato M et al., 2018). However, in our stimulation system, spinal stimulation underperforms in comparison to previous literature. The reason for this may be that electrical stimulation was given only to the L2-L4 segments of the spinal cord, where stimulation activates only the motor neuron pools that control the flexor muscles, but not the motor neuron pools that control the extensors. As we all know, a complete flexion and extension requires coordinated activation of the flexor and extensor muscles. It is suggested that the addition of spinal cord stimulation sites on the basis of spinal cord - muscle DES may further enhance the motor function of the hindlimb of mice with spinal cord injury.”

d)Fourth, would it be possible that the authors provide supplementary Videos to the article for the first few days of training and the final days of training with no stimulation, spinal stimulation, muscle stimulation and their combination. This would be important to rule out confounding effects. For example, in Figure 3h, there is a pronounced hip flexion for the SCS and MS conditions in comparison to

the dual site stimulation. A simple change in the biomechanical placement of the leg on the treadmill could easily confound the results of the study. Therefore, sharing representative Videos for at least the 20 to 40 Hz SCS, the 20 to 40Hz MS and the optimal dual site stimulation seem important.

Response: As suggested by the reviewer, we now provide supplementary videos for the untrained group, the SCS group (20-40 Hz), the MS group (20-40 Hz) and the DES group (10-20 Hz) at 3 and 18 days after stimulation training. Simultaneously, we have reconstructed the stick diagrams of hind limb movement in mice based on the video footage (See Figure 3 and Figure 3-figure supplement 1).

e) A discussion on the interaction of muscle stimulation and the antidromic response of spinal stimulation (Formento et al., 2020) should be included in the discussion.

Response: We agree with the reviewer's point. We now discuss this point in the revised manuscript (Page 32-33, lines 687-702).

“Epidural electrical stimulation (EES) of the spinal cord restores locomotion in animal models of spinal cord injury but is less effective in humans (Hunter JP et al., 1994; Su CF et al., 1992; Buonocore M et al., 2008). This interspecies discrepancy is due to interference between EES and proprioceptive information in humans (Formento E et al., 2018). This transient deafferentation prevents modulation of reciprocal inhibitory networks involved in locomotion and reduces or abolishes the conscious perception of leg position. In our study, electrical stimulation signals in the spinal cord may also produce a similar antidromic conduction, counteracting proprioceptive incoming information in mice. Since both the antidromic response of SCS signal and the forward conduction of MS signal are through sensory fibers, the conduction time of both signals should be less than 10 ms (Figure 1-figure supplement 4B). But the MS start 15 ms after SCS, so MS does not interact with the inverse response of SCS. In a DES system at the spinal cord and muscle interface, SCS was administered first, followed by MS after the end of the spinal-evoked response. Therefore, dual SCS and MS have a timing-sequence stimulation pattern, which avoids the mutual interference of reverse reactions and enhances sensory feedback. We can mitigate the antidromic response of spinal stimulation by DES at the spinal-muscular interface to a certain extent.”

I complement the authors on the extraordinary amount of group comparisons and refined experimental readouts.

3) Further minor comments:

a) If the authors could add in the description to the methods, a summary how many

animals they used for the study and how they managed to train these large amounts of animals in Figure 2, this would help readers to estimate the feasibility of the study and its reproducibility. For example, how many treadmills were run in parallel? Did this require the attendance of several technical assistants, or was the training fully automated? How was the bodyweight support adjusted on an individual basis? Was the training blinded?

Response: Following the reviewer's comments, we have added the relevant information in Materials and Methods.

“In Fig. 2, this experiment design includes sham group, untrained group, 1-40 Hz DES (Dual electrical stimulation) group, 1-40 Hz MS group, 1-40 Hz SCS group and 1-40 Hz SCShc group, each group includes 8 mice. Electrical stimulation onset represents the point at which electrical stimulation training begins (7 days after electrode implantation). In the sham group, the electrodes were implanted during sham operation, but the electrical stimulation training was not performed; In the untrained group, electrodes were implanted in the spinal cord during the T9 segment transection surgery to avoid secondary damage due to electrode implantation, but the electrical stimulation training was not performed; Mice in the 1-40 Hz DES, 1-40 Hz MS, 1-40 Hz SCS, and 1-40 Hz SCShc groups were implanted with stimulating electrodes during the T9 segment transection surgery, and the mice were trained with electrical stimulation 7 days later. The mice underwent electrical stimulation training for 3 weeks. SCEP was recorded at 7 and 29 days after electrode implantation in all experimental groups. All mice were implanted with the stimulation electrodes on the same day.” (Page 41, lines 817-827)

“In Figure 3, each group include six mice. In the sham group, the electrodes were implanted during sham operation, but the electrical stimulation training was not performed. In the untrained group, electrodes were implanted in the spinal cord during the T9 segment transection surgery to avoid secondary damage due to electrode implantation, but the electrical stimulation training was not performed. Mice in 10-20 Hz DES, 10-20 Hz MS, 10-20 Hz SCS and 10-20 Hz SCShc groups (2 mice for 10 Hz, 2 mice for 15 Hz, 2 mice for 20 Hz) were implanted with stimulating electrodes during the T9 segment transection surgery, and the mice were trained with electrical stimulation 7 days later. The mice underwent electrical stimulation training for 3 weeks, and then the TA muscle myoelectric burst and joint activity trajectory were detected when the mice were walking on a treadmill. There are three treadmills in the laboratory for the experiment, and six technical assistants are involved. The treadmill moves forward at a constant speed (2 m/min). The technicians adjust the body of the mice so that the hindlimb of the mice just touch the conveyor belt of the treadmill, and the technical assistants were blind to trained mice.” (Page 53, lines 1086-1100)

b) Figure 11f, the y-axis should be converted to 100 percent

Response: We thank the reviewer for the comment and have modified the y-axis label accordingly: “Intensity of Akt in MN (100% of untrained)” “Intensity of p-Akt in MN (100% of untrained)”, and "Intensity of PKAc in MN (100% of untrained)". (See Figure 11-figure supplement 2D,E)

c)The majority of the figures deal with readouts from the same experiment, and often the same data from the healthy and sham groups are presented multiple times. Maybe there are ways to combine some of the figures in a more concise way? Similar articles from the journal could serve as a figure template.

Response: We thank the reviewer for this critical information. Accordingly, the figure layout has been improved. (Please see Figures. 2, 3, 4, 5)

d)In the text, multiple times the same statistical test is spelled out and interrupts the reading process. Maybe the authors could find a way, to reduce this text and put it into the methods section and figure legends?

Response: Following the reviewer’s comment, in the text, repeated statistical methods have been placed into Materials methods.

“We used a two-sample unpaired t-test for the analyses in Figure 1H, Figure 9I-L, and Figure 9Q-S comparing the relative variability in the amplitude of spinal cord-evoked EMGs before and after SCI as well as before and after dosing and in Figure 11A comparing the relative expression of different genes in the untrained group and the 10- to 20- Hz DES groups; in supplementary Figure 1C, the proportion of c-Fos positive neurons in the ipsilateral and contralateral regions following stimulation was compared (Pages 58-59, lines 1198-1219).

In addition, in Figure1-figure supplement 3H, paired t-test was used to compare the latency of spinal cord response and muscle response induced by muscle stimulation; in Figure 12D,G, the paired t-test was used to analyze frequency changes pre and during stimulation, pre and post stimulation, and during and post stimulation in the same mouse.

One-way ANOVA with Bonferroni correction was used for Figure 2C-E, Figure 2 Supplementary Figure 1A,B, Figure 3P,Q, Figure 4K,L, Figure 5C, Figure 6C, F, G, I, J, Figure 7C-E, 7H, Figure 8D,E, Figure 9O, Figure 10M,N, and Figure 11-figure supplement 2D,E, with the untrained group as the comparison group. In addition, ANOVA with Bonferroni correction was used for Figure 5C, Figure 6C, F, G, I, J, Figure 7C-E,H, Figure 8D,E, and Figure 9O for multiple comparisons between the sham group and the ES onset group and between the sham group and the 10–20 Hz DES group.

We used a two-way ANOVA for Figure 3B-E, and the untrained group was used as the comparison group to conduct multiple comparisons with Tukey’s post hoc test.

A similar analysis was used for Figure 3F, with the 10-20 Hz DES group as the comparison group.”

e) Could the authors provide a photo of the implants from Supplementary figure 1, and reference the applied materials and production processes? Possibly, also say whether these implants are available to the scientific community. These questions could facilitate reproducibility of results.

Response: As requested by the reviewer, we have added detailed information concerning the electrode in Materials and Methods and Figure 1-figure supplement 1. The process of electrode production is referred to the reported literature (Mao GW et al., 2022), and the process was optimized and adjusted for the purpose of our study. The composition of materials in the electrode is now described in the revised manuscript (Page 37, lines 736-743).

“T-shaped FPC substrate was used for spinal epidural electrodes. The specific information was as follows: Polyimide material (PI) 9 μm , Glue 4 μm , Cuprum (Cu) 12 μm , PI 8 μm . Glue 4 The thickness of the gold coating on the electrode wire and contact surface is ignored, and the total thickness was 37 μm , regardless of plating thickness.

Information concerning the mouse hindlimb muscle stimulation electrode was revised as follows: Polyimide material (PI) 12.5 μm , Cuprum (Cu) 12 μm , Glue 25 μm , PI 12.5 μm . The thickness of the gold coating on the electrode wire and contact surface is ignored, and the total thickness was 62 μm , regardless of plating thickness.”

f) In the abstract, the first part of the following sentence sounds like a tautology and should be simplified: Different parameters of electrical stimulation encode diverse characteristics of electrical signals, which leads to various effects on the remodelling of neural circuits, indicating that the stimulation parameters are related to different neural activities.

Response: We are grateful for your careful review and the suggestion. we have simplified the statement as follows:

“The neural signals produced by different electrical stimulation parameters lead to characteristic neural circuit responses, indicating that the stimulation parameters are related to diverse neural activities.” (Page 2; lines 24-26)

Citations:

1. Poplawski GHD, Kawaguchi R, Van Niekerk E, Lu P, Mehta N, Canete P, Lie R, Dragatsis I, Meves JM, Zheng B, Coppola G, Tuszynski MH. Injured adult neurons regress to an embryonic transcriptional growth state. *Nature*. 2020 May;581(7806):77-82. doi: 10.1038/s41586-020-2200-5. Epub 2020 Apr 15. PMID: 32376949.
2. Capogrosso M, Wenger N, Raspopovic S, Musienko P, Beauparlant J, Bassi Luciani L, Courtine G, Micera S. A computational model for epidural electrical stimulation of spinal sensorimotor circuits. *J Neurosci*. 2013 Dec 4;33(49):19326-40. doi: 10.1523/JNEUROSCI.1688-13.2013. PMID: 24305828; PMCID: PMC6618777.
3. Tresch MC, Kiehn O. Motor coordination without action potentials in the mammalian spinal cord. *Nat Neurosci*. 2000 Jun;3(6):593-9. doi: 10.1038/75768. PMID: 10816316.
4. Arber S. Motor circuits in action: specification, connectivity, and function. *Neuron*. 2012 Jun 21;74(6):975-89. doi: 10.1016/j.neuron.2012.05.011. PMID: 22726829.
5. Arber S, Costa RM. Connecting neuronal circuits for movement. *Science*. 2018 Jun 29;360(6396):1403-1404. doi: 10.1126/science.aat5994. PMID: 29954969.
6. Pal A, Park H, Ramamurthy A, Asan AS, Bethea T, Johnkutty M, Carmel JB. Spinal cord associative plasticity improves forelimb sensorimotor function after cervical injury. *Brain*. 2022 Dec 19;145(12):4531-4544. doi: 10.1093/brain/awac235. PMID: 36063483; PMCID: PMC10200304.
7. Mishra AM, Pal A, Gupta D, Carmel JB. Paired motor cortex and cervical epidural electrical stimulation timed to converge in the spinal cord promotes lasting increases in motor responses. *J Physiol*. 2017 Nov 15;595(22):6953-6968. doi: 10.1113/JP274663. Epub 2017 Aug 20. PMID: 28752624; PMCID: PMC5685837.
8. Yang Q, Ramamurthy A, Lall S, Santos J, Ratnadurai-Giridharan S, Lopane M, Zareen N, Alexander H, Ryan D, Martin JH, Carmel JB. Independent replication of motor cortex and cervical spinal cord electrical stimulation to promote forelimb motor function after spinal cord injury in rats. *Exp Neurol*. 2019 Oct;320:112962. doi: 10.1016/j.expneurol.2019.112962. Epub 2019 May 21. PMID: 31125548; PMCID: PMC7035596.
9. Bertels H, Vicente-Ortiz G, El Kanbi K, Takeoka A. Neurotransmitter phenotype switching by spinal excitatory interneurons regulates locomotor recovery after spinal cord injury. *Nat Neurosci*. 2022 May;25(5):617-629. doi: 10.1038/s41593-022-01067-9. Epub 2022 May 6. PMID: 35524138; PMCID: PMC9076533
10. Angeli CA, Edgerton VR, Gerasimenko YP, Harkema SJ. Altering spinal cord excitability enables voluntary movements after chronic complete paralysis in

humans. *Brain*. 2014 May;137(Pt 5):1394-409. doi: 10.1093/brain/awu038. Epub 2014 Apr 8. Erratum in: *Brain*. 2015 Feb;138(Pt 2):e330. PMID: 24713270; PMCID: PMC3999714.

11. Chen B, Li Y, Yu B, Zhang Z, Brommer B, Williams PR, Liu Y, Hegarty SV, Zhou S, Zhu J, Guo H, Lu Y, Zhang Y, Gu X, He Z. Reactivation of Dormant Relay Pathways in Injured Spinal Cord by KCC2 Manipulations. *Cell*. 2018 Jul 26;174(3):521-535.e13. doi: 10.1016/j.cell.2018.06.005. Epub 2018 Jul 19. Erratum in: *Cell*. 2018 Sep 6;174(6):1599. PMID: 30033363; PMCID: PMC6063786.

12. Courtine G. Reducing neuronal inhibition restores locomotion in paralysed mice. *Nature*. 2018 Sep;561(7723):317-318. doi: 10.1038/d41586-018-06651-3. PMID: 30224731.

13. Bonizzato M, Pidpruzhnykova G, DiGiovanna J, Shkorbatova P, Pavlova N, Micera S, Courtine G. Brain-controlled modulation of spinal circuits improves recovery from spinal cord injury. *Nat Commun*. 2018 Aug 1;9(1):3015. doi: 10.1038/s41467-018-05282-6. PMID: 30068906; PMCID: PMC6070513.

14. van den Brand R, Heutschi J, Barraud Q, DiGiovanna J, Bartholdi K, Huerlimann M, Friedli L, Vollenweider I, Moraud EM, Duis S, Dominici N, Micera S, Musienko P, Courtine G. Restoring voluntary control of locomotion after paralyzing spinal cord injury. *Science*. 2012 Jun 1;336(6085):1182-5. doi: 10.1126/science.1217416. PMID: 22654062.

15. Asboth L, Friedli L, Beauparlant J, Martinez-Gonzalez C, Anil S, Rey E, Baud L, Pidpruzhnykova G, Anderson MA, Shkorbatova P, Batti L, Pagès S, Kreider J, Schneider BL, Barraud Q, Courtine G. Cortico-reticulo-spinal circuit reorganization enables functional recovery after severe spinal cord contusion. *Nat Neurosci*. 2018 Apr;21(4):576-588. doi: 10.1038/s41593-018-0093-5. Epub 2018 Mar 19. PMID: 29556028.

16. Formento E, Minassian K, Wagner F, Mignardot JB, Le Goff-Mignardot CG, Rowald A, Bloch J, Micera S, Capogrosso M, Courtine G. Electrical spinal cord stimulation must preserve proprioception to enable locomotion in humans with spinal cord injury. *Nat Neurosci*. 2018 Dec;21(12):1728-1741. doi: 10.1038/s41593-018-0262-6. Epub 2018 Oct 31. PMID: 30382196; PMCID: PMC6268129.

17. Buonocore M, Bonezzi C, Barolat G. Neurophysiological evidence of antidromic activation of large myelinated fibres in lower limbs during spinal cord stimulation. *Spine (Phila Pa 1976)*. 2008 Feb 15;33(4):E90-3. doi: 10.1097/BRS.0b013e3181642a97. PMID: 18277861.

18. Su CF, Haghghi SS, Oro JJ, Gaines RW. "Backfiring" in spinal cord monitoring. High thoracic spinal cord stimulation evokes sciatic response by antidromic sensory pathway conduction, not motor tract conduction. *Spine (Phila Pa 1976)*. 1992 May;17(5):504-8. PMID: 1621148.

19. Hunter JP, Ashby P. Segmental effects of epidural spinal cord stimulation in humans. *J Physiol.* 1994 Feb 1;474(3):407-19. doi: 10.1113/jphysiol.1994.sp020032. PMID: 8014902; PMCID: PMC1160332.
20. Mao GW, Zhang JJ, Su H, Zhou ZJ, Zhu LS, Lü XY, Wang ZG. A flexible electrode array for determining regions of motor function activated by epidural spinal cord stimulation in rats with spinal cord injury. *Neural Regen Res.* 2022 Mar;17(3):601-607. doi: 10.4103/1673-5374.320987. PMID: 34380900; PMCID: PMC8504402.

Reviewers' Comments:

Reviewer #1:

Remarks to the Author:

The authors have addressed my comments adequately. It would be helpful to state the method adjustment for multiple comparisons that was used in the Methods. The authors' response to my previous comments on this (on page 8 of their response) are very hard to understand. They mention a wide variety of different methods, all of which are appropriate. The manuscript should state clearly which method was used.

Eran Mukamel

Reviewer #2:

Remarks to the Author:

The Authors have put in a significant amount of work into the revisions.

The Authors have defined the early, mid and late responses with respect to the electrophysiological tests. The large amount of experimental results laid out in this manuscript now can be interpreted clearly.

These revisions included the use of activity-dependent c-fos labeling and the Ca²⁺ imaging which generated evidence for the activation of dorsal horn neurons after MS.

However, several serious issues remain that need to be addressed by the Authors for this manuscript to be of standard publication quality.

The attached file contains about 3+ pages of the errors I could identify.

The Authors have defined the early, mid and late responses. The large amount of experimental results laid out in this manuscript now can be interpreted clearly. These revisions included the use of activity-dependent c-fos labeling and the Ca²⁺ imaging which generated evidence for the activation of dorsal horn neurons after MS. However, several serious issues remain that need to be addressed by the Authors for this manuscript to be of standard publication quality.

- *What are the noteworthy results?*

The results showed that a unique range of stimulation frequency used for the electrical stimulation of muscle tissue in one muscle controlling the ankle and the toes has had superior effects in terms of movement improvements and also in terms of synaptic connectivity and muscle tissue innervation.

- *Will the work be of significance to the field and related fields? How does it compare to the established literature? If the work is not original, please provide relevant references.*

This body of work has a novel protocol incorporated here which will be an incremental albeit a significant advance in the field. Methodologically, if the technical feasibility of the dual stimulation methods will be as great as the Authors claim (about 90%) the field will also be advanced.

- *Does the work support the conclusions and claims, or is additional evidence needed?*
- *Are there any flaws in the data analysis, interpretation and conclusions? Do these prohibit publication or require revision?*

Yes, at this point in the manuscript draft- as noted in my detailed comments by Line numbers.

- *Is the methodology sound? Does the work meet the expected standards in your field?*

The methods are sufficient for the standards of the field for electrophysiology.

- *Is there enough detail provided in the methods for the work to be reproduced?*

After the first revisions, yes.

Critical errors and problems in the current format of this manuscript:

Recommend the use of the abbreviation - EEMS – instead of dual or DES – to be consistent with EES use for epidural electrical and MS for muscle stimulation as used in the Introduction.

Line 112- Magnetic resonance imaging is not described, yet there is a Figure in Fig. 1D where it is shown. Please, clarify if this was only done at time for one animal, or it was neglected from the manuscript methods. Unclear, if this was a one-time image or routinely done on the mice or what this imaging has added in the case of one animal if the data was not summarized. Was this purely used to adjust the electrode?

Line 116-This is incorrect information. Epidural stimulation cannot be used to stimulate the dorsal spinal cord. It is both sides of the cord that are stimulated. The conductive nature of the cerebrospinal fluid allows for the immediate excitation of multiple regions as shown by modeling studies (i.e. Capogrosso et al cited by the Authors).

Line 120: Here the Authors define and use SCEP. In Figure 1, the Authors use SCEE labeling beside the traces and the measurements. What is SCEE?

Line 120: References# 28, 29 does not describe the latency of spinal responses in mice when peripheral nerve stimulation or muscle stimulation is applied. Appropriate citation would be the referencing of any one of the following papers:

Meehan, C.F., Grondahl, L., Nielsen, J.B. and Hultborn, H. (2012), Fictive locomotion in the adult decerebrate and spinal mouse *in vivo*. *The Journal of Physiology*, 590: 289-300.

Tuan V Bui, Nicolas Stifani, Turgay Akay, Robert M Brownstone (2016) Spinal microcircuits comprising dI3 interneurons are necessary for motor functional recovery following spinal cord transection *eLife* 5:e21715

Stecina, K. (2017). "Midbrain stimulation-evoked lumbar spinal activity in the adult decerebrate mouse." *J Neurosci Methods* 288: 99-105.

All these papers indicate the latency of electrically evoked responses from the periphery to the spinal cord in a set-up that is nearly identical to the electrode positioning here.

Fig. 1-F,G and H- what is being stimulated here? Is the stimulation delivered to the SC or to the M here? It needs to be indicated on the Figure and also in the Figure legend.

Line 130- The blocking of sodium currents also would effect the dorsal roots in the portion that enter the spinal cord, but not the peripheral nerve. If these responses were evoked by spinal stimulation (which is not described in the text, in the figure or in the figure legends) then these responses after TTX reflect afferent fibres which were not blocked by the TTX.

Line 131 and 133: It would be more accurate to say that ER represents dorsal root activation before any synaptic activity occurs.

Line 136: Tizanidine is an alpha-2 receptor antagonist. There are alpha-2 receptors in multiple Rexed laminae (see. e.g. by the studies from Noga et al in the cat or by ... in the mouse spinal cord) and the transmission is effected in pathways via all those neurons that possess those receptors. One of the affected pathways is the group II afferent input relayed by some classes of the so called group II interneurons. However, pain transmission is also influenced by this drug; and it is not exclusively modulating the group II reflex pathway.

Line 173 – “This suggested that the epidural reaction could not have resulted from muscle spindle feedback evoked by the electrically induced contraction of muscle fibres.” This sentence contradicts what is written on Line 167-168: results suggested that MS activates sensory fibers, which in turn activate spinal dorsal horn neurons.” The phrase “epidural reaction” should be also replaced, as that is not a reaction, it is an evoked potential. Moreover, the Authors need to consider the possibility that muscle afferents will be activated both antidromically and orthodromically upon the MS. Electrical stimulation as delivered here would have recruited the largest diameter afferents and efferents and that signal would have been conducted to the muscle and to the cord. The illustration the Authors chose to show in Fig. 1-Supl 3 depicts a

simultaneous component in the cord and in the muscle-in fact, the recordings in Fig. 1-Sup. 2 showed that the conduction time from the cord to the muscle is the same – about 5ms. This is in fact, primarily the activation of dorsal roots. This is what the Authors recorded following MS as well – and there is no evidence here that muscle spindle afferent feedback is absent. In fact, most of the signals are the result of group I afferent fibre activation – as a large component remained after the epidural TTX application. Therefore this entire section, “This suggested that the epidural reaction could not have resulted from muscle spindle feedback, evoked by the electrically induced contraction of muscle fibers. Thus, in subsequent experiments, TA muscles received 400 μ A of stimulation via the DES system to transmit sensory feedback to the spinal cord.” – is convoluted and contradictory. Please, revise.

Line 193- There are multiple papers showing that high-frequency stimulation also promotes recovery of function after SCI – the Authors need to revise this sentence.

Line 259 – What type of movement were these recordings from? If locomotion, then Authors need to indicate where is the start of extension and flexion of the steps for Fig. 3J-O or for how many steps are represented in the time window shown for these panels to be possible comprehensible.

Fig. 3- panels J-O has a blue trace next to each panel. What is that trace? It is not 5 s long as stated in the figure legend. Explain what those traces are and why those are illustrated there. If those are averages, how many of those were averaged, what was the reference point for the averaging?

Fig.3 and Fig. 3-supl. 2- State whether the EMG recordings are raw data or filtered/rectified. Explain what the blue traces are because the expression “representative waveform” does not convey if these are single trials, or an average of several trial, or a typical burst selected from 10, or 100 or 1000 bursts (or from 10 s or 100 min of recording). Or if these are just simple one of the burst on the right enlarged and illustrated in a different time scale from those burst shown in black on the left panels? This should be added to each figure legend when traces are shown.

Line 287- Muscle contractility is a complex term. It is unclear in what context the Authors use this term here. However, as the Authors showed on Fig. 3 that all types of stimulation resulted in more EMG bursts in the TA than in the untrained group- this statement here is contradictory. Please, clarify what was measured by the length sensor. Also, clarify during what task where these measurements collected.

Line 296- Complete the sentence. Reads “... with an antibody that recognizes.” What?

Line 321- It needs to be indicated here that the mouse line used for these experiments was different from the line used for functional studies. The *Limbx1cre* line was used here, based on the description in figure 6A.

Line 322- When was this injection made and how long after the electrical stimulation training was the 2 weeks wait period applied? Was this procedure performed for all mice or just a subset of mice from which the ENG recordings and in vivo muscle functional testing was done?

Line 330-331- Incorrect statement. The expression “promoted the innervation of proprioceptive axons in the spinal cord after SCI” is an incorrect statement. There is nothing in this paper showing that “axons” become innervated”. Axons of the proprioceptive neurons innervate other neurons. Rephrase this.

Line 339-343- The observations shown in Fig. 6D regarding the intermediate and deeper layer’s innervation by DRG is purely illustrative. There has been no quantification done, no methods provided how would this be quantified therefore, these statements are presented without other quantitative supporting evidence other than the illustrations shown in Fig. 6. This is misleading and these conclusive statements should be removed or re-worded indicating the appearance and qualitative assessment being the reason for stating this.

Line 751- This paragraph is explaining the implantation of the spinal electrode. This electrode was sutured to the TA muscle? Then how would have been place to insert another electrode into the TA muscle? And the spinal electrode was that long that one end had to be sutured to the TA muscle? The illustrations in Fig. 1 were entirely different.

Line 772- Here it is stated that the SCEPs in the TA were after 15 ms, and it was used as the guidance for judging electrode placement. This contradicts what is stated earlier and what is depicted in Figures 1-3.

Lines 778-780-The threshold for TA activation was different based on Figure 1 than stated here (400 uA). In Fig. The threshold is between 300 and 400 uA. Threshold typically represent the stimulation intensity at which discernable responses of 50 uV potentials can be evoked by 2/5 trials. Based on the data the Authors showed; that intensity was below 400 uA; unless the Author’s definition of threshold has been different than the most typical one used in the field. Please, clarify.

Line 782- What is physiologically unstable? Please, define.

Line 856: In this context, it is incorrect to state that TTX blocks synaptic transmission. It effects all sodium channels. Especially those on axons in the white matter tracts before it diffuses into the gray matter given the intrathecal application of this chemical here in this study. Correct this statement and also the related statements which are misleading at multiple times throughout the paper.

Line 857-858: the chemical tizanidine, has multiple actions, one of which is that it “decreases the excitability of spinal interneurons supplied by Group II fibers” – and the Authors should avoid this statement in the methods and simply state what channels this chemical is targeting as done for the other chemicals written.

Discussion- How much were the mice supported during the spinal and muscle stimulation training? The amount of hindlimb loading for regenerating motor actions after spinal cord injury is a major factor; therefore it would be informative to know how much load was on the hindlimbs while the stimulation protocol was applied on the mice. Air-stepping (no load) or full body weight load alone has a major effect on re-wiring of spinal networks and differentiating between the load-related and the spinal and muscle stimulation-related effects would be

important. It is not addressed if the mice received the same amount of unloading and if that was normalized for all cohorts – thus addressing that in the Discussion would help the reader to accept that the reported main effects were not hindlimb load-related.

Reviewer #3:

Remarks to the Author:

After review, the article has much improved, and most concerns have been addressed. The authors provide many additional experiments, such as excellent additional readouts on Glutamate and Gaba neurotransmission in the spinal cord under different stimulation conditions.

I have few remaining remarks:

Response to Point 1a)

The electrophysiological results for time delayed response in EMG with respect to evoked potentials in the spinal cord support the idea of a direct activation of sensory fibers. C-fos expression and GCAMP6 experiments only provide proof for an activation of sensory circuits, not a proof whether spinal cord activation resulted from muscle fiber contraction or direct sensory fiber activation. Therefore, I encourage minimal adjustment in the wording, avoiding to say that c-fos expression demonstrates a 'direct' activation of sensory fibres.

Supplementary Video:

A) Could you please add a longer period for the DES 20 Hz stimulation at day 3 of stimulation and after recovery (e.g. 10 continuous steps). Currently there are so few steps, that selection bias is difficult to exclude, and it would be more convincing if the supreme performance of DES is supported by Video footage. Also, the DES 20 Hz stimulation at day three shows extensive contraction of the left leg, with some steps on the right leg. This, I would not call functional stepping, because the left leg is maximally flexed. Rather this shows, that there are additional side effects introduced through DES in terms of behavioral function. The flexed leg also raises the question, how you picked the leg that you chose for analysis? If you only picked always the better leg, then you should state this in the methods and point this out as a limitation of the study and DES at early timepoints.

B) For 40 Hz SCS and 15 Hz DES and particularly 20 Hz DES on day 3 of stimulation, it looks like somebody is repeatedly lifting the weight support arm where the animal is attached. In the DES 20 Hz this manipulation of the arm even has a lateral movement. The support arm is not fully visible, so it is unclear how this external perturbation came about. Lifting the support arm changes the contact of the animal to the ground, provides additional sensory cues to the leg, and therefore improves stepping. This is a critical aspect, as you might have not studied the effect of stimulation, but rather the effect of a stimulation plus dynamic weight adjustment, possibly with adjustments restricted to the DES conditions. Maybe that was the case? In the text, you also did not say whether the technicians were blinded to the stimulation conditions, only that they were blinded to trained mice. If that is the case, your DES effects might have potentially resulted from better physiotherapy in the DES condition, and not the stimulation itself. Can you provide any evidence that the technicians trained all animals the same way, without dynamic adjustments in body weight support, and whether they were blinded for the stimulation conditions and experimental design?

Minor remark: double check grammar in these sentence:

38 Overall, the

39 results provide insights into neural signal decoding during spinal sensorimotor circuit

40 reconstruction and suggest a novel approach for treating unique approach for treating

REVIEWER COMMENTS

Reviewer #1 (Remarks to the Author):

The authors have addressed my comments adequately. It would be helpful to state the method adjustment for multiple comparisons that was used in the Methods. The authors' response to my previous comments on this (on page 8 of their response) are very hard to understand. They mention a wide variety of different methods, all of which are appropriate. The manuscript should state clearly which method was used.

Eran Mukamel

Reviewer #2 (Remarks to the Author):

The Authors have put in a significant amount of work into the revisions.

The Authors have defined the early, mid and late responses with respect to the electrophysiological tests. The large amount of experimental results laid out in this manuscript now can be interpreted clearly.

These revisions included the use of activity-dependent c-fos labeling and the Ca²⁺ imaging which generated evidence for the activation of dorsal horn neurons after MS.

However, several serious issues remain that need to be addressed by the Authors for this manuscript to be of standard publication quality.

The attached file contains about 3+ pages of the errors I could identify.

- What are the noteworthy results?

The results showed that a unique range of stimulation frequency used for the electrical stimulation of muscle tissue in one muscle controlling the ankle and the toes has had superior effects in terms of movement improvements and also in terms of synaptic connectivity and muscle tissue innervation.

- Will the work be of significance to the field and related fields? How does it compare to the established literature? If the work is not original, please provide relevant references.

This body of work has a novel protocol incorporated here which will be an incremental albeit a significant advance in the field. Methodologically, if the technical feasibility of the dual stimulation methods will be as great as the Authors claim (about 90%) the field will also be advanced.

- Does the work support the conclusions and claims, or is additional evidence needed?
- Are there any flaws in the data analysis, interpretation and conclusions? Do these prohibit publication or require revision?

Yes, at this point in the manuscript draft- as noted in my detailed comments by Line numbers.

- Is the methodology sound? Does the work meet the expected standards in your field?

The methods are sufficient for the standards of the field for electrophysiology.

- Is there enough detail provided in the methods for the work to be reproduced?

After the first revisions, yes.

Critical errors and problems in the current format of this manuscript:

Recommend the use of the abbreviation - EEMS - instead of dual or DES - to be consistent with EES use for epidural electrical and MS for muscle stimulation as used in the Introduction.

Line 112- Magnetic resonance imaging is not described, yet there is a Figure in Fig. 1D where it is shown. Please, clarify if this was only done at time for one animal, or it was neglected from the manuscript methods. Unclear, if this was a one-time image or routinely done on the mice or what this imaging has added in the case of one animal if the data was not summarized. Was this purely used to adjust the electrode?

Line 116-This is incorrect information. Epidural stimulation cannot be used to stimulate the dorsal spinal cord. It is both sides of the cord that are stimulated. The conductive nature of the cerebrospinal fluid allows for the immediate excitation of multiple regions as shown by modeling studies (i.e. Capogrosso et al cited by the Authors).

Line 120: Here the Authors define and use SCEP. In Figure 1, the Authors use SCEE labeling beside the traces and the measurements. What is SCEE?

Line 120: References# 28, 29 does not describe the latency of spinal responses in mice when peripheral nerve stimulation or muscle stimulation is applied. Appropriate citation would be the referencing of any one of the following papers:

Meehan, C.F., Grondahl, L., Nielsen, J.B. and Hultborn, H. (2012), Fictive locomotion in the adult decerebrate and spinal mouse in vivo. *The Journal of Physiology*, 590: 289-300.

Tuan V Bui, Nicolas Stifani, Turgay Akay, Robert M Brownstone (2016) Spinal microcircuits comprising dI3 interneurons are necessary for motor functional recovery following spinal cord transection *eLife* 5:e21715

Stecina, K. (2017). "Midbrain stimulation-evoked lumbar spinal activity in the adult decerebrate mouse." *J Neurosci Methods* 288: 99-105.

All these papers indicate the latency of electrically evoked responses from the periphery to the spinal cord in a set-up that is nearly identical to the electrode positioning here.

Fig. 1-F,G and H- what is being stimulated here? Is the stimulation delivered to the SC or to the M here? It needs to be indicated on the Figure and also in the Figure legend.

Line 130- The blocking of sodium currents also would affect the dorsal roots in the portion that enter the spinal cord, but not the peripheral nerve. If these responses were evoked by spinal stimulation (which is not described in the text, in the figure or in the figure legends) then these responses after TTX reflect afferent fibres which were not blocked by the TTX.

Line 131 and 133: It would be more accurate to say that ER represents dorsal root activation before any synaptic activity occurs.

Line 136: Tizanidine is an alpha-2 receptor antagonist. There are alpha-2 receptors in multiple Rexed laminae (see, e.g. by the studies from Noga et al in the cat or by ... in the mouse spinal cord) and the transmission is effected in pathways via all those neurons that possess those receptors. One of the affected pathways is the group II afferent input relayed by some classes of the so called group II interneurons. However, pain transmission is also influenced by this drug; and it is not exclusively modulating the group II reflex pathway.

Line 173 – “This suggested that the epidural reaction could not have resulted from muscle spindle feedback evoked by the electrically induced contraction of muscle fibres.” This sentence contradicts what is written on Line 167-168: results suggested that MS activates sensory fibers, which in turn activate spinal dorsal horn neurons.” The phrase “epidural reaction” should be also replaced, as that is not a reaction, it is an evoked potential. Moreover, the Authors need to consider the possibility that muscle afferents will be activated both antidromically and orthodromically upon the MS. Electrical stimulation as delivered here would have recruited the largest diameter afferents and efferents and that signal would have been conducted to the muscle and to the cord. The illustration the Authors chose to show in Fig. 1-Supl 3 depicts a simultaneous component in the cord and in the muscle—in fact, the recordings in Fig. 1-Sup. 2 showed that the conduction time from the cord to the muscle is the same – about 5ms. This is in fact, primarily the activation of dorsal roots. This is what the Authors recorded following MS as well – and there is no evidence here that muscle spindle afferent feedback is absent. In fact, most of the signals are the result of group I afferent fibre activation – as a large component remained after the epidural TTX application. Therefore this entire section, “This suggested that the epidural reaction could not have resulted from muscle spindle feedback, evoked by the electrically induced contraction of muscle fibers. Thus, in subsequent experiments, TA muscles received 400 μ A of stimulation via the DES system to transmit sensory feedback to the spinal cord.” – is convoluted and contradictory. Please, revise.

Line 193- There are multiple papers showing that high-frequency stimulation also promotes recovery of function after SCI – the Authors need to revise this sentence.

Line 259 – What type of movement were these recordings from? If locomotion, then Authors need to indicate where is the start of extension and flexion of the steps for Fig. 3J-O or for how many steps are represented in the time window shown for these panels to be possible comprehensible.

Fig. 3- panels J-O has a blue trace next to each panel. What is that trace? It is not 5 s long as stated in the figure legend. Explain what those traces are and why those are illustrated there. If those are averages, how many of those were averaged, what was the reference point for the averaging?

Fig.3 and Fig. 3-supl. 2- State whether the EMG recordings are raw data or filtered/rectified. Explain what the blue traces are because the expression "representative waveform" does not convey if these are single trials, or an average of several trial, or a typical burst selected from 10, or 100 or 1000 bursts (or from 10 s or 100 min of recording). Or if these are just simple one of the burst on the right enlarged and illustrated in a different time scale from those burst shown in black on the left panels? This should be added to each figure legend when traces are shown.

Line 287- Muscle contractility is a complex term. It is unclear in what context the Authors use this term here. However, as the Authors showed on Fig. 3 that all types of stimulation resulted in more EMG bursts in the TA than in the untrained group- this statement here is contradictory. Please, clarify what was measured by the length sensor. Also, clarify during what task where these measurements collected.

Line 296- Complete the sentence. Reads "... with an antibody that recognizes." What?

Line 321- It needs to be indicated here that the mouse line used for these experiments was different from the line used for functional studies. The Limbx1cre line was used here, based on the description in figure 6A.

Line 322- When was this injection made and how long after the electrical stimulation training was the 2 weeks wait period applied? Was this procedure performed for all mice or just a subset of mice from which the ENG recordings and in vivo muscle functional testing was done?

Line 330-331- Incorrect statement. The expression "promoted the innervation of proprioceptive axons in the spinal cord after SCI" is an incorrect statement. There is nothing in this paper showing that "axons" become innervated". Axons of the proprioceptive neurons innervate other neurons. Rephrase this.

Line 339-343- The observations shown in Fig. 6D regarding the intermediate and deeper layer's innervation by DRG is purely illustrative. There has been no quantification done, no methods provided how would this be quantified therefore, these statements are presented without other quantitative supporting evidence other than the illustrations shown in Fig. 6. This is misleading and these conclusive statements should be removed or re-worded indicating the appearance and qualitative assessment being the reason for stating this.

Line 751- This paragraph is explaining the implantation of the spinal electrode. This electrode was sutured to the TA muscle? Then how would have been place to insert another electrode into the TA muscle? And the spinal electrode was that long that one end had to be sutured to the TA muscle? The illustrations in Fig. 1 were entirely different.

Line 772- Here it is stated that the SCEPs in the TA were after 15 ms, and it was used as the guidance for judging electrode placement. This contradicts what is stated earlier and what is depicted in Figures 1-3.

Lines 778-780-The threshold for TA activation was different based on Figure 1 than stated here (400 uA). In Fig. The threshold is between 300 and 400 uA. Threshold typically represent the stimulation intensity at which discernable responses of 50 uV potentials can be evoked by 2/5 trials. Based on the data the Authors showed; that intensity was below 400 uA; unless the Author's definition of threshold has been different than the most typical one used in the field. Please, clarify.

Line 782- What is physiologically unstable? Please, define.

Line 856: In this context, it is incorrect to state that TTX blocks synaptic transmission. It effects all sodium channels. Especially those on axons in the white matter tracts before it diffuses into the gray matter given the intrathecal application of this chemical here in this study. Correct this statement and also the related statements which are misleading at multiple times throughout the paper.

Line 857-858: the chemical tizanidine, has multiple actions, one of which is that it "decreases the excitability of spinal interneurons supplied by Group II fibers" – and the Authors should avoid this statement in the methods and simply state what channels this chemical is targeting as done for the other chemicals written.

Discussion- How much were the mice supported during the spinal and muscle stimulation training? The amount of hindlimb loading for regenerating motor actions after spinal cord injury is a major factor; therefore it would be informative to know how much load was on the hindlimbs while the stimulation protocol was applied on the mice. Air-stepping (no load) or full body weight load alone has a major effect on re-wiring of spinal networks and differentiating between the load-related and the spinal and muscle stimulation-related effects would be important. It is not addressed if the mice received the same amount of unloading and if that was normalized for all cohorts – thus addressing that in the Discussion would help the reader to accept that the reported main effects were not hindlimb load-related.

Reviewer #3 (Remarks to the Author):

After review, the article has much improved, and most concerns have been addressed. The authors provide many additional experiments, such as excellent additional readouts on Glutamate and Gaba neurotransmission in the spinal cord under different stimulation conditions.

I have few remaining remarks:

Response to Point 1a)

The electrophysiological results for time delayed response in EMG with respect to evoked potentials in the spinal cord support the idea of a direct activation of sensory fibers. C-fos expression and GCAMP6 experiments only provide proof for an activation of sensory circuits, not a proof whether spinal cord activation resulted from muscle fiber contraction or direct sensory fiber activation. Therefore, I encourage minimal adjustment in the wording, avoiding to say that c-fos expression demonstrates a 'direct' activation of sensory fibres.

Supplementary Video:

A) Could you please add a longer period for the DES 20 Hz stimulation at day 3 of stimulation and after recovery (e.g. 10 continuous steps). Currently there are so few steps, that selection bias is difficult to exclude, and it would be more convincing if the supreme performance of DES is supported by Video footage. Also, the DES 20 Hz stimulation at day three shows extensive contraction of the left leg, with some steps on the right leg. This, I would not call functional stepping, because the left leg is maximally flexed. Rather this shows, that there are additional side effects introduced through DES in terms of behavioral function. The flexed leg also raises the question, how you picked the leg that you chose for analysis? If you only picked always the better leg, then you should state this in the methods and point this out as a limitation of the study and DES at early timepoints.

B) For 40 Hz SCS and 15 Hz DES and particularly 20 Hz DES on day 3 of stimulation, it looks like somebody is repeatedly lifting the weight support arm where the animal is attached. In the DES 20 Hz this manipulation of the arm even has a lateral movement. The support arm is not fully visible, so it is unclear how this external perturbation came about. Lifting the support arm changes the contact of the animal to the ground, provides additional sensory cues to the leg, and therefore improves stepping. This is a critical aspect, as you might have not studied the effect of stimulation, but rather the effect of a stimulation plus dynamic weight adjustment, possibly with adjustments restricted to the DES conditions. Maybe that was the case? In the text, you also did not say whether the technicians were blinded to the stimulation conditions, only that they were blinded to trained mice. If that is the case, your DES effects might have potentially resulted from better physiotherapy in the DES condition, and not the stimulation itself. Can you provide any evidence that the technicians trained all animals the same way, without dynamic adjustments in body weight support, and whether they were blinded for the stimulation conditions and experimental design?

Minor remark: double check grammar in these sentence:

38 Overall, the

39 results provide insights into neural signal decoding during spinal sensorimotor circuit

40 reconstruction and suggest a novel approach for treating unique approach for treating

We sincerely appreciate the opportunity to further refine our manuscript and extend our gratitude to the reviewers for their valuable feedback. Their insightful comments and suggestions on additional data analysis and textual flaws have significantly enhanced our conclusions, contributing to substantial improvements in the manuscript. Below, we provide detailed responses to each of your questions. (*Our responses are in blue text*)

Reviewer #1 (Remarks to the Author):

The authors have addressed my comments adequately. It would be helpful to state the method adjustment for multiple comparisons that was used in the Methods. The authors' response to my previous comments on this (on page 8 of their response) are very hard to understand. They mention a wide variety of different methods, all of which are appropriate. The manuscript should state clearly which method was used.

Eran Mukamel

Response: Dr. Eran Mukamel, thank you very much for your insightful comments regarding our manuscript, and your helps make the shortcomings of this paper to be improved. We apologize for any confusion caused by the unclear description in the manuscript. We had modified the description of calibration method for single-cell transcriptome sequencing analysis as follows: “We adjusted p-values for multiple testing in R using Benjamini and Hochberg (BH) method to control the false discovery rate (FDR). Genes with an adjusted p-value ≤ 0.1 and a fold change ≥ 1.5 were considered significant”. (Pages 58-59, lines 1203-1206)

Reviewer #2 (Remarks to the Author):

The Authors have put in a significant amount of work into the revisions.

The Authors have defined the early, mid and late responses with respect to the electrophysiological tests. The large amount of experimental results laid out in this manuscript now can be interpreted clearly.

These revisions included the use of activity-dependent c-fos labeling and the Ca²⁺ imaging which generated evidence for the activation of dorsal horn neurons after MS.

However, several serious issues remain that need to be addressed by the Authors for this manuscript to be of standard publication quality.

The attached file contains about 3+ pages of the errors I could identify.

Thank you very much for your insightful comments regarding our manuscript. We have carefully revised it accordingly, and our responses are provided below on a point-by-point basis. Changes made to the manuscript have been highlighted in the revised version.

• What are the noteworthy results?

The results showed that a unique range of stimulation frequency used for the electrical stimulation of muscle tissue in one muscle controlling the ankle and the toes has had superior effects in terms of movement improvements and also in terms of synaptic connectivity and muscle tissue innervation.

• Will the work be of significance to the field and related fields? How does it compare to the established literature? If the work is not original, please provide relevant references.

This body of work has a novel protocol incorporated here which will be an incremental albeit a significant advance in the field. Methodologically, if the technical feasibility of the dual stimulation methods will be as great as the Authors claim (about 90%) the field will also be advanced.

• Does the work support the conclusions and claims, or is additional evidence needed?

• Are there any flaws in the data analysis, interpretation and conclusions? Do these prohibit publication or require revision?

Yes, at this point in the manuscript draft- as noted in my detailed comments by Line numbers.

• Is the methodology sound? Does the work meet the expected standards in your field?

The methods are sufficient for the standards of the field for electrophysiology.
• *Is there enough detail provided in the methods for the work to be reproduced?*
After the first revisions, yes.

Critical errors and problems in the current format of this manuscript:

Recommend the use of the abbreviation - EEMS – instead of dual or DES – to be consistent with EES use for epidural electrical and MS for muscle stimulation as used in the Introduction.

Response: We thank the reviewer for the suggestion. We have thoroughly revised the manuscript accordingly.

Line 112- Magnetic resonance imaging is not described, yet there is a Figure in Fig. 1D where it is shown. Please, clarify if this was only done at time for one animal, or it was neglected from the manuscript methods. Unclear, if this was a one-time image or routinely done on the mice or what this imaging has added in the case of one animal if the data was not summarized. Was this purely used to adjust the electrode?

Response: We sincerely apologize for the omission of the magnetic resonance imaging method in the manuscript. We have now included a comprehensive description of the MRI in the Materials and Methods. The purpose of MRI on the spinal cord was to ensure the stability of electrode placement during electrical stimulation training, thus only one mouse underwent examination subsequent to completion of the electrical stimulation training.

“After undergoing anesthetized mice were positioned supine and securely immobilized using medical tape. The spinal cord was imaged using a high-resolution multi-directional T2-weighted imaging (T2WI) technique with a 11.7T nuclear magnetic resonance imaging employing fast spin echo (FSE). The imaging parameters were as follows: repetition time (TR) =2000 ms, echo time (TE) =26 ms, field of view (FOV) =3 cm × 3 cm, matrix size =128 × 128, slice thickness = 0.5 mm, and cumulative scans performed four times. To minimize the influence of respiratory motion on the results, a respiratory gating technique was employed.” (Page 44, lines 888-896)

Line 116-This is incorrect information. Epidural stimulation cannot be used to stimulate the dorsal spinal cord. It is both sides of the cord that are stimulated. The conductive nature of the cerebrospinal fluid allows for the immediate excitation of multiple regions as shown by modeling studies (i.e. Capogrosso et al cited by the Authors).

Response: We totally accept the reviewer's point and sincerely apologize for our inaccuracies – the sentence now reads as follows: “To analyze the conduction of electrical signals in the TA sensorimotor reflex circuit under normal physiological conditions, the spinal cord (L2–L4) was stimulated along the midline in anesthetized, uninjured mice.” (Page 6, lines 117-119)

Line 120: Here the Authors define and use SCEP. In Figure 1, the Authors use SCEE labeling beside the traces and the measurements. What is SCEE?

Response: We sincerely apologize for the confusion resulting from our negligence. The EES was administered to evoke a response in the TA muscle. We define this evoked response recorded in the muscle as spinal cord–evoked potential (SCEP). Due to our negligence, we forgot to rectify SCEE to SCEP. We have rectified the image. (Figure 1 and Figure 2)

Line 120: References# 28, 29 does not describe the latency of spinal responses in mice when peripheral nerve stimulation or muscle stimulation is applied. Appropriate citation would be the referencing of any one of the following papers:

Meehan, C.F., Grondahl, L., Nielsen, J.B. and Hultborn, H. (2012), Fictive locomotion in the adult decerebrate and spinal mouse in vivo. The Journal of Physiology, 590: 289-300.

Tuan V Bui, Nicolas Stifani, Turgay Akay, Robert M Brownstone (2016) Spinal microcircuits comprising dI3 interneurons are necessary for motor functional recovery following spinal cord transection eLife 5:e21715

Stecina, K. (2017). "Midbrain stimulation-evoked lumbar spinal activity in the adult decerebrate mouse." J Neurosci Methods 288: 99-105.

All these papers indicate the latency of electrically evoked responses from the periphery to the spinal cord in a set-up that is nearly identical to the electrode positioning here.

Response: We thank the reviewer for this comment. The reviewer may have the

misunderstanding about our description here. Here, EES is applied to evoke responses in the TA muscle. We define the evoked response recorded in the TA as spinal cord-evoked potentials (SCEP). Furthermore, references #28 and #29 (line 120) describe the latency of SCEPs recorded in the muscle during EES. However, due to the significance of the references provided by the reviewer, we have cited these references which appropriate to describe the latency of spinal responses when muscle stimulation was applied. (Page 9, lines 176-178)

Fig. 1-F,G and H- what is being stimulated here? Is the stimulation delivered to the SC or to the M here? It needs to be indicated on the Figure and also in the Figure legend.

Response: We apologize for any confusion caused to the reviewer. The Figure has been supplemented with additional details, which are also provided in the Figure legend. The results depicted in Figure 1F, G, and H illustrate the SCEP recorded in the muscles subsequent to EES. The sentence now reads as follows: “The schematic illustrates which neurons, fibers, and/or circuits may be inhibited as stimulation signals delivered to the muscles from the spinal cord during each experimental operation in anesthetized intact mice”. (Figure 1F,G, and H, page 70, lines 1521-1526)

Line 130- The blocking of sodium currents also would effect the dorsal roots in the portion that enter the spinal cord, but not the peripheral nerve. If these responses were evoked by spinal stimulation (which is not described in the text, in the figure or in the figure legends) then these responses after TTX reflect afferent fibres which were not blocked by the TTX.

Response: Thank you very much for this important comment. We are greatly inspired by the reviewer’s insightful perspectives. Capogrosso et al. (2013) showed that EES-evoked ER after intrathecal injection of TTX was dependent on direct recruitment of motor axons. Combining the literature reports (Tresch, M.C. et al., 2000; Raastad, M. et al., 1998) and analysis of the results, we speculate that the ER recorded in the muscles after intrathecal delivery of TTX mainly result from efferent nerve recruitment. At the same time, we take into account the correct viewpoint of the reviewer: “The blocking of sodium currents also would effect the dorsal roots in the portion that enter the spinal cord, but not the peripheral nerve”. Therefore, we believe that ER may also be evoked by direct afferent nerve recruitment. In summary, our results shown that ER was not dependent on chemical neurotransmission of spinal cord neurons but rather relied on the direct recruitment of motor axons. Meanwhile,

afferent nerves may also contribute to ER. We have made the following revisions to enhance clarity in our manuscript:

“To evaluate the signal pathway corresponding to ERs, we blocked sodium currents through local intrathecal delivery of tetrodotoxin into spinal cord segments L5/L6 (Figure 1F). The intrathecal administration of TTX primarily affected the spinal cord and the dorsal roots in the portion that enter the spinal cord, with a lesser impact on efferent nerves and the peripheral nerve of DRG, which inhibited chemical neurotransmission evoked by action potentials²⁸⁻³². In this study, tetrodotoxin was found to abolish both MRs and LRs, but ERs were not affected (Figure 1F-H). These results suggested that ER was not dependent on chemical neurotransmission of spinal cord neurons but rather relied on the direct recruitment of motor axons. Meanwhile, the direct recruitment of afferent nerves may also contribute to ER”. (Page 7, lines 133-141)

Pharmacology and electrophysiological manipulations have been described in the Materials and Methods, and we have now provided explanations in the figure legend as well. (Page 45, lines 901-904; Page 70, lines 1521-1526)

Line 131 and 133: It would be more accurate to say that ER represents dorsal root activation before any synaptic activity occurs.

Response: Thank you very much for making us aware of this issue. As mentioned above, we accept the reviewer’s view. Capogrosso et al. (2013) showed that EES-evoked ER after intrathecal injection of TTX was dependent on direct recruitment of motor axons. At the same time, we take into account the correct viewpoint of the reviewer: “The blocking of sodium currents also would effect the dorsal roots in the portion that enter the spinal cord, but not the peripheral nerve”. And, since the latency of ER is the shortest, ER may be evoked by direct efferent or afferent nerve recruitment. Based on this, we believe that the reviewer’s opinion is also correct. Therefore, we have made the following modifications to the manuscript:

“To evaluate the signal pathway corresponding to ERs, we blocked sodium currents through local intrathecal delivery of tetrodotoxin into spinal cord segments L5/L6 (Figure 1F). The intrathecal administration of TTX primarily affected the spinal cord and the dorsal roots in the portion that enter the spinal cord, with a lesser impact on efferent nerves and the peripheral nerve of DRG, which inhibited chemical neurotransmission evoked by action potentials²⁸⁻³². In this study, tetrodotoxin was found to abolish both MRs and LRs, but ERs were not affected (Figure 1F-H). These results suggested that ER was not dependent on chemical neurotransmission of spinal cord neurons but rather relied on the direct recruitment of motor axons. Meanwhile,

the direct recruitment of afferent nerves may also contribute to ER". (Page 7, lines 133-141)

Line 136: Tizanidine is an alpha-2 receptor antagonist. There are alpha-2 receptors in multiple Rexed laminae (see, e.g. by the studies from Noga et al in the cat or by ... in the mouse spinal cord) and the transmission is effected in pathways via all those neurons that possess those receptors. One of the affected pathways is the group II afferent input relayed by some classes of the so called group II interneurons. However, pain transmission is also influenced by this drug; and it is not exclusively modulating the group II reflex pathway.

Response: Thank you very much for this important comment and we apologize for not having been more considerate in the first place. As pointed out by the reviewer, it should be noted that tizanidine does not exclusively regulate the group II reflex pathway, and pain transmission is also influenced by this drug. Therefore, we have made modifications to our description of the result: "Furthermore, intrathecal administration of the alpha2-adrenergic receptor agonist tizanidine not only suppressed group II reflex pathways, but also interfered with pain transmission" (Page 7, lines 141-143)

Line 173 – "This suggested that the epidural reaction could not have resulted from muscle spindle feedback evoked by the electrically induced contraction of muscle fibres." This sentence contradicts what is written on Line 167-168: results suggested that MS activates sensory fibers, which in turn activate spinal dorsal horn neurons." The phrase "epidural reaction" should be also replaced, as that is not a reaction, it is an evoked potential. Moreover, the Authors need to consider the possibility that muscle afferents will be activated both antidromically and orthodromically upon the MS. Electrical stimulation as delivered here would have recruited the largest diameter afferents and efferents and that signal would have been conducted to the muscle and to the cord. The illustration the Authors chose to show in Fig. 1-Supl 3 depicts a simultaneous component in the cord and in the muscle-in fact, the recordings in Fig. 1-Sup. 2 showed that the conduction time from the cord to the muscle is the same – about 5ms. This is in fact, primarily the activation of dorsal roots. This is what the Authors recorded following MS as well – and there is no evidence here that muscle spindle afferent feedback is absent. In fact, most of the signals are the result of group I afferent fibre activation – as a

large component remained after the epidural TTX application. Therefore this entire section, “This suggested that the epidural reaction could not have resulted from muscle spindle feedback, evoked by the electrically induced contraction of muscle fibers. Thus, in subsequent experiments, TA muscles received 400 μ A of stimulation via the DES system to transmit sensory feedback to the spinal cord.” – is convoluted and contradictory. Please, revise.

Response: Thank you very much for your insightful remarks. Your point of view provides a reminder that the original description lacks precision. As the reviewer pointed out, these findings do not demonstrate that the muscle spindle’s input is absent, and thus we modified the manuscript as you suggested, combining the conclusions of line 167 and line 173. The sentence now reads as follows: “These results indicate that MS activates the sensory circuit by directly activating sensory fibers, thereby eliciting epidural potentials. Thus, in subsequent experiments, TA muscles received 400 μ A of stimulation via the EEMS system to transmit sensory feedback to the spinal cord.” (Page 9, lines 179-182)

Following your suggestion, the term “epidural reaction” was revised to “evoked potential”. (Page 9, lines 181)

Line 193- There are multiple papers showing that high-frequency stimulation also promotes recovery of function after SCI – the Authors need to revise this sentence.

Response: We thank the reviewer for the suggestion – the sentence now reads as follows: “The current research suggests that electrical stimulation at a certain frequency range can contribute to the recovery of sensory and motor functions following spinal cord injury (SCI). To investigate the specific frequencies in the EEMS system that play a crucial role in electrical stimulation, we chose to identify the frequency characteristics of effective electrical stimulation in the frequency range 1–40 Hz.” (Page 10, lines 197-199)

Line 259 – What type of movement were these recordings from? If locomotion, then Authors need to indicate where is the start of extension and flexion of the steps for Fig. 3J-O or for how many steps are represented in the time window shown for these panels to be possible comprehensible.

Response: We thank the reviewer for this comment. To detect EMG bursts during rhythmic movement of mice, we designed a body weight-supporting device combined with a treadmill so that the hind limbs of the mice just touched the track

of the treadmill. The treadmill operated at a speed of 2 m·min⁻¹, and EMG signals from the TA muscles were observed when the mice were moving.

Surface EMG recordings were conducted during treadmill exercise, as depicted in Figure 3G. The TA muscle was specifically chosen as the target muscle in the EEMS system, and thus only surface EMG activity of the TA muscle was recorded. As depicted in Figure 3J-O, mice from the sham group exhibited five steps within 5 s, while both untrained and trained groups displayed three to four steps. (Page 12, lines 253-257; Page 85, lines 1590-1595; Page 90, lines 1618-1620)

Fig. 3- panels J-O has a blue trace next to each panel. What is that trace? It is not 5 s long as stated in the figure legend. Explain what those traces are and why those are illustrated there. If those are averages, how many of those were averaged, what was the reference point for the averaging?

Response: We thank the reviewer for this comment. To enhance clarity, we have delineated the bursts employed for magnification using dashed boxes within the black traces, and updated Figure 3 and Figure 3-figure supplement 2. The blue trace is a magnified view of the EMGs burst in the dashed box on the left, which is to facilitate the observation of the EMGs burst characteristics during TA contraction. It should be noted that the blue trace does not depict an average waveform, but merely magnifies the black trace on the left. (Page 85, lines 1590-1593; Page 90, lines 1608-1620)

Fig.3 and Fig. 3-supl. 2- State whether the EMG recordings are raw data or filtered/rectified. Explain what the blue traces are because the expression “representative waveform” does not convey if these are single trials, or an average of several trial, or a typical burst selected from 10, or 100 or 1000 bursts (or from 10 s or 100 min of recording). Or if these are just simple one of the burst on the right enlarged and illustrated in a different time scale from those burst shown in black on the left panels? This should be added to each figure legend when traces are shown.

Response: Thank you very much for this comment. The EMG recordings depicted in the figure have been suitably filtered (Band Stop channel 1 parameter, 50 Hz; Band Stop channel 2 parameter, 100 Hz; and Low Pass channel parameter, 2000 Hz). (Page 55, lines 1136-1138)

The black track is taken from a 30-second consecutive EMGs recording. The

blue track represents a typical burst within the black track. To enhance clarity, we have delineated the bursts employed for magnification using dashed boxes within the black traces. We have added this information to the figure legend. (Page 85, lines 1590-1593; Page 90, lines 1608-1620)

Line 287- Muscle contractility is a complex term. It is unclear in what context the Authors use this term here. However, as the Authors showed on Fig. 3 that all types of stimulation resulted in more EMG bursts in the TA than in the untrained group- this statement here is contradictory. Please, clarify what was measured by the length sensor. Also, clarify during what task where these measurements collected.

Response: We thank the reviewer for this comment. The TA muscle is a flexor muscle that controls the ankle joint. When the TA muscle contracts, the ankle joint flexes. Here, muscle contractility refers to the stretching and contracting of muscles, and also represents the flexion process of the ankle joint. Pressure-sensitive sensors are used to measure the pressure exerted on the sensor by muscle contractions. A pressure-sensitive sensor was attached to the TA muscle epidermis, and then 1-V constant voltage was applied to the sensor. Mice moved freely in the open field, and when the TA muscle relaxed, the squeezing force of the muscle on the sensor was small, resulting in a large sensor resistance, and the current passing through was denoted I_0 ; when the TA muscle contracted, the squeezing force of the muscle on the sensor was large, resulting in a reduction of sensor resistance, and the current passing through was denoted I . The relative current through the sensor was as follows: $\Delta I / I_0 = (I - I_0) / I_0$. $\Delta I / I_0$ was applied to quantify the degree of contraction of the TA muscle. (Figure 4A; Page 14, lines 285-286; Pages 56-57, lines 1157-1174)

We would like to express our gratitude once again to the reviewer for your valuable comments, which have prompted us to recognize that it is more appropriate to modify “The duration of a single EMG burst” as “The duration of a single TA burst”. (Page 13, lines 272, 274, and 280)

In Figure 3, the EMG data was measured on a treadmill under conditions of body weight support (90%~). As pointed out by the reviewer, the results in Figure 3 showed that all types of stimulation resulted in more EMG bursts in the TA than in the untrained group, as all types of electrical stimulation can activate the sensorimotor neural circuit to varying degrees. However, each TA burst consisted of a series of EMG bursts, which were recorded in the TA muscle. Thus, the number of EMG bursts is not equivalent to the number of TA bursts. At the same time, the results showed that compared to the untrained group, the duration of each TA burst in the 10-20 Hz EEMS group was significantly shortened, and the amplitude was

significantly increased. The shortening of the duration of the TA burst may contribute to the production of more TA contractions within a unit time, and the increase in amplitude helps produce stronger muscle contraction force. This is consistent with the results described in Figure 4, which showed that compared with the untrained group, both frequency and intensity of TA muscle contraction significantly increased within a unit of time in mice from 10-20 Hz EEMS group.

Line 296- Complete the sentence. Reads "... with an antibody that recognizes." What?

Response: We appreciate the reviewer's comment—the sentence now reads as follows: "NMJs were labeled by immunofluorescence staining with α -bungarotoxin (α -BTX) to label the acetylcholine receptor (AChR) and with antibodies to neurofilament/synapsin (NF/Syn) to label axon terminals." (Page 15, lines 304-306)

Line 321- It needs to be indicated here that the mouse line used for these experiments was different from the line used for functional studies. The *Limb1cre* line was used here, based on the description in figure 6A.

Response: Thank you very much for making us aware of this issue. The virus tracer experiment depicted in Figure 6 was conducted using *Lbx1^{Cre}* mice. In Figure 8, the virus tracer experiment was performed utilizing *ChAT^{Cre}* mice. The remaining virus tracer experiments and functional studies were carried out employing C57BL/6J mice. We have now included the genetic information of mice in our manuscript. (Page 5, lines 101-105; Pages 14-15, lines 302-304; Page 16, lines 327-330, lines 341-345; Page 18, lines 372-374; Page 19, lines 390-393; Page 20, lines 412-415; Page 22, lines 454-456; Page 23, lines 486-488; Page 24, lines 512-515; Page 27, lines 574-577)

Line 322- When was this injection made and how long after the electrical stimulation training was the 2 weeks wait period applied? Was this procedure performed for all mice or just a subset of mice from which the ENG recordings and in vivo muscle functional testing was done?

Response: Thank you very much for this important comment. In Figure 6, the spinal cord injection virus is performed before implanting the spinal cord electrode. One week after electrical stimulation, virus injection was performed in the DRG and

muscle, followed by two weeks of electrical stimulation training before perfusion. This procedure was only performed on a subset of mice from which the EMG recordings and in vivo muscle functional testing. The remaining functionally tested mice were used to label interneurons, detect calcium fluorescence signals, and probe neurotransmitters. We have supplemented information regarding the timing of virus injection in the Materials and Methods. (Page 16, lines 327-330, 341-345; Pages 46-47, lines 936-962)

Line 330-331- Incorrect statement. The expression “promoted the innervation of proprioceptive axons in the spinal cord after SCI” is an incorrect statement. There is nothing in this paper showing that “axons” become innervated”. Axons of the proprioceptive neurons innervate other neurons. Rephrase this.

Response: We are sorry for the inaccuracy—the sentence now reads as follows: “These results suggested that 10- to 20- Hz EEMS promoted the axons of the proprioceptive neurons innervate the spinal neurons.” (Page 16, lines 337-339)

Line 339-343- The observations shown in Fig. 6D regarding the intermediate and deeper layer’s innervation by DRG is purely illustrative. There has been no quantification done, no methods provided how would this be quantified therefore, these statements are presented without other quantitative supporting evidence other than the illustrations shown in Fig. 6. This is misleading and these conclusive statements should be removed or re-worded indicating the appearance and qualitative assessment being the reason for stating this.

Response: Thank you very much for this important comment and we apologize for not having been more considerate in the first place. As pointed out by the reviewer, statistical analysis has not been conducted on the results presented in Figure 6D. We sincerely apologize for any inappropriate conclusive statements made. These conclusive statements have been removed to improve the quality of the manuscript, and we have revised this part of the manuscript: “The appearance of DRG fibers extending into the ventral spinal cord, where motoneurons reside, was observed in the 10-20 Hz EEMS groups (Figure 6D). In order to further observe the connection between DRG fibers and interneurons, AAV-DIO-mCherry was injected into the intermediate area of gray matter at the L2 – L4 segment of *Lbx1^{cre}* mice to label the interneurons”. (Pages 16-17, lines 345-348)

Line 751- This paragraph is explaining the implantation of the spinal electrode. This electrode was sutured to the TA muscle? Then how would have been place to insert another electrode into the TA muscle? And the spinal electrode was that long that one end had to be sutured to the TA muscle? The illustrations in Fig. 1 were entirely different.

Response: We apologize for any confusion caused by the unclear description in the manuscript. Instead of being sutured to the TA muscle, the spinal electrode's ears are stitched to the muscles adjacent to the vertebrae. On the other hand, the epidermis of the hind limb is incised and the muscle electrodes are securely affixed to the surface of the TA muscle, while suturing is performed to secure the position of the electrodes. We have made revisions to the inaccurate descriptions of materials and methods. (Page 39, lines 776-782)

Line 772- Here it is stated that the SCEPs in the TA were after 15 ms, and it was used as the guidance for judging electrode placement. This contradicts what is stated earlier and what is depicted in Figures 1-3.

Response: We apologize for the unclear organization in our previous manuscript. Our intended message is that following EES, the evoked potentials recorded in TA muscle terminate within a duration of 15 ms; whereas after MS, the evoked potentials recorded in the spinal cord cease within 10 ms. Based on your suggestion, we have revised the manuscript for this section: "Then, we calculated that the time from the start of spinal pulse to the end of muscle response was 13.34 ± 0.66 ms, and the time from the start of muscle pulse to the end of spinal response was 8.35 ± 0.49 ms. The stimulus frequency range we selected was 1-40 Hz. To prevent overlap of the dual stimulation effects, the sum of the time from the beginning of the spinal pulse to the end of the muscle response and the time from the beginning of the muscle pulse to the end of the spinal response was no more than 25 ms (1/40 s). Therefore, the interval between spinal stimulation and muscle stimulation was set at 15 ms to ensure that the SCEP was fully released." (Page 40, lines 800-808)

Lines 778-780-The threshold for TA activation was different based on Figure 1 than stated here (400 uA). In Fig. The threshold is between 300 and 400 uA. Threshold typically represent the stimulation intensity at which discernable

responses of 50 uV potentials can be evoked by 2/5 trials. Based on the data the Authors showed; that intensity was below 400 uA; unless the Author's definition of threshold has been different than the most typical one used in the field. Please, clarify.

Response: We thank the reviewer for their thorough examination of our manuscript and figures. As pointed out by the reviewer, the results in Figure 1J indicate that the threshold for MS is between 300 and 400 μ A. On this basis, in order to ensure that each MS during the electrical stimulation training process can effectively activate the sensory circuit, the intensity of MS is set to 400 μ A. The sentence now reads as follows: "The threshold for MS was between 300 and 400 μ A. In order to ensure that each MS could effectively activate the sensory circuit during the electrical stimulation training process, a 400 μ A current was applied to the TA muscle, and the effective response (unimodal peak) was recorded in the spinal epidural within 10 ms. Thus, TA muscles received electrical stimulation of 400 μ A in the EEMS and MS systems, transmitting sensory feedback to the spinal cord." (Page 41, lines 811-815)

Line 782- What is physiologically unstable? Please, define.

Response: We thank the reviewer for this comment. It has been reported that EES with a current range of 210 to 400 μ A can facilitate hind limb movement in rats with spinal cord injury (Asboth, L., et al. 2018; Bonizzato, M., et al. 2018, 2021; Moraud, E.M., et al. 2018). However, due to their smaller size, mice have a lower tolerance for stimulation intensity compared to rats. We found that when the intensity of the EES exceeded 300 μ A, the mice suffered violent twitching of the whole body and stiffness of the hind limbs, indicating that the mice were in an unstable state. (Page 41, lines 819-821)

Line 856: In this context, it is incorrect to state that TTX blocks synaptic transmission. It effects all sodium channels. Especially those on axons in the white matter tracts before it diffuses into the gray matter given the intrathecal application of this chemical here in this study. Correct this statement and also the related statements which are misleading at multiple times throughout the paper.

Response: We thank the reviewer for this comment. We totally accept the reviewer's comment and sincerely apologize for our inaccuracies – the sentence now reads as follows: "pharmacological modulation of EES-evoked motor responses was tested 30 min after using the sodium channel blocker tetrodotoxin (1 μ M,

MedChemExpress, USA);” (Page 45, lines 901-904)

The accurate expression should be that TTX blocks the sodium currents, thereby inhibiting the chemical neurotransmission induced by the action potential. And the manuscript has been fully updated: “To evaluate the signal pathway corresponding to ERs, we blocked sodium currents through local intrathecal delivery of tetrodotoxin into spinal cord segments L5/L6 (Figure 1F). The intrathecal administration of TTX primarily affected the spinal cord and the dorsal roots in the portion that enter the spinal cord, with a lesser impact on efferent nerves and the peripheral nerve of DRG, which inhibited chemical neurotransmission evoked by action potentials²⁸⁻³². In this study, tetrodotoxin was found to abolish both MRs and LRs, but ERs were not affected (Figure 1F-H). These results suggested that ER was not dependent on chemical neurotransmission of spinal cord neurons but rather relied on the direct recruitment of motor axons. Meanwhile, the direct recruitment of afferent nerves may also contribute to ER” (Page 7, lines 131-141)

Line 857-858: the chemical tizanidine, has multiple actions, one of which is that it “decreases the excitability of spinal interneurons supplied by Group II fibers” – and the Authors should avoid this statement in the methods and simply state what channels this chemical is targeting as done for the other chemicals written.

Response: We thank the reviewer for the suggestion. Following your advice, we have removed the statement that “decreases the excitability of spinal interneurons supplied by Group II fibers”. The sentence now reads as follows: “pharmacological modulation of EES-evoked motor responses was tested 30 min after using the sodium channel blocker tetrodotoxin (1 μ M, MedChemExpress, USA); the 2-adrenergic receptor agonist tizanidine (1 mM, MedChemExpress);” (Page 45, lines 901-904)

Discussion- How much were the mice supported during the spinal and muscle stimulation training? The amount of hindlimb loading for regenerating motor actions after spinal cord injury is a major factor; therefore it would be informative to know how much load was on the hindlimbs while the stimulation protocol was applied on the mice. Air-stepping (no load) or full body weight load alone has a major effect on re-wiring of spinal networks and differentiating between the load-related and the spinal and muscle stimulation-related effects would be important. It is not addressed if the mice received the same amount of unloading and if that was normalized for all cohorts – thus addressing that in the Discussion would help the reader to accept that the reported main effects were not hindlimb load-related.

Response: We thank the reviewer for this comment. In all types of electrical stimulation training (EES, MS, EEMS, and EEShc), the trunk of the mouse is fixed to a simple fixator, while the limbs are in a suspended state (no load).

Based on your suggestion, we included a description of the load on the hind limbs of mice in the Discussion: “Awake mice trained with electrical stimulation were fixed to a simple fixator in the study. The trunk of the mouse was fixed to the fixator, while its limbs were in a suspended state (no load). As is well known, there are neuronal circuits in the lumbosacral segment of the spinal cord that control motor output, called central pattern generator (CPG)^{56,57}. Body weight supported treadmill training can provide combining sensory cues of tactile, proprioceptive, and kinesthetic for CPG in SCI mice^{26,58}. This multimodal sensory input is crucial for activating the basic neural circuits in the spinal cord, adjusting motor patterns, and improving motor performance^{58,59}. However, in order to analyze the characteristics of electrical signals that promote the remodeling of spinal cord local neural circuits and the recovery of motor function, spinal T9 complete transection model was used in this study. Complete T9 spinal cord transection leads to the complete interruption of supraspinal input to the lumbosacral neural circuit. Combined with the condition of complete spinal cord transection and hind limb suspension, the supraspinal excitation source in the lumbar segment was interrupted, and the touch, proprioception and kinesthetic of the hind limb were excluded. EEMS is used to simulate the motor output of the spinal cord and peripheral sensory feedback under normal physiological conditions. The results showed that under the condition of no load on the hindlimb, only 10-20 Hz EEMS could effectively promote the rearrangement of local neural circuits and the recovery of hindlimb motor function after spinal cord injury compared with other electrical stimulation groups of mice.” (Pages 30-31, lines 634-653)

Reviewer #3 (Remarks to the Author):

After review, the article has much improved, and most concerns have been addressed. The authors provide many additional experiments, such as excellent additional readouts on Glutamate and Gaba neurotransmission in the spinal cord under different stimulation conditions.

Once again, we appreciate the comments of the reviewer and hope our responses and revisions address your points with satisfaction.

I have few remaining remarks:

Response to Point 1a)

The electrophysiological results for time delayed response in EMG with respect to evoked potentials in the spinal cord support the idea of a direct activation of sensory fibers. C-fos expression and GCAMP6 experiments only provide proof for an activation of sensory circuits, not a proof whether spinal cord activation resulted from muscle fiber contraction or direct sensory fiber activation. Therefore, I encourage minimal adjustment in the wording, avoiding to say that c-fos expression demonstrates a ‘direct’ activation of sensory fibres.

Response: Thank you very much for this comment. We totally accept the reviewer’s comment and sincerely apologize for our inaccuracies. Indeed, the expression of c-Fos and GCAMP6 only indicates that the sensory circuit of the spinal cord is activated during muscle stimulation. Further, the simultaneous recording of evoked signals in the spinal cord and muscles after muscle stimulation proved that sensory fibers were directly activated by muscle stimulation. In summary, the results show that muscle stimulation activates sensory circuits by activating sensory fibers. Hence, we have removed the term “direct” in our description of c-Fos. The sentence now reads as follows: “To determine whether MS could activate sensory fibers, we first examined the expression of c-Fos, a neuronal activity marker, in the lumbar spinal cord 1.5 h after MS”. (Page 8, lines 159-161)

In addition, the conclusions of c-Fos expression and GCAMP6 experiment were integrated with the conclusions of electrophysiological experiment: “These results indicate that MS activates the sensory circuit by directly activating sensory fibers, thereby eliciting epidural potentials. Thus, in subsequent experiments, TA muscles received 400 μ A of stimulation via the EEMS system to transmit sensory feedback to the spinal cord.” (Page 9, lines 179-182)

Supplementary Video:

A) Could you please add a longer period for the DES 20 Hz stimulation at day 3 of stimulation and after recovery (e.g. 10 continuous steps). Currently there are so few steps, that selection bias is difficult to exclude, and it would be more convincing if the supreme performance of DES is supported by Video footage. Also, the DES 20 Hz shows extensive contraction of the left leg, with some steps on the right leg. This, I would not call functional stepping, because the left leg is maximally flexed. Rather this shows, that there are additional side effects introduced through DES in terms of behavioral function. The flexed leg also raises the question, how you picked the leg that you chose for analysis? If you only picked always the better leg, then you should state this in the methods and point this out as a limitation of the study and DES at early timepoints.

Response: We thank the reviewer for the suggestion. According to the suggestion provided by the reviewer, we conducted additional experiments and video recordings in order to eliminate the impact of subjective factors on the mice. In the supplementary video, we present a sequential recording of 20 s by each mouse.

The new video results revealed that after three days of electrical stimulation, no noticeable functional gait was observed in the hind limbs of all mice within the electrical stimulation group. Subsequently, following 18 days of stimulation, a superior recovery in motor function was observed in the right hind limb compared to the left hind limb within the 10-20 Hz EEMS group. The spinal cord electrode was implanted in the midline of the epidural space, as shown in Figure 1A. The muscle electrode was implanted in the TA of the right hind limb, not in the TA of the left hind limb, so that, EEMS was applied to the right side of the body and only EES was applied to the left side in the same mouse. Additionally, video analysis revealed that only a few mice in the 10-20 Hz EEMS group exhibited extensive contractions of their left hind limbs, which may be attributed to individual differences.

Furthermore, since electrodes were only implanted in the right hind limb in the EEMS system, analysis was limited to the right hind limb for all functional tests. (Pages 43-44, lines 874-876; Page 55, lines 1124-1125, 1132-1134; Pages 56-57, lines 1157-1162)

B) For 40 Hz SCS and 15 Hz DES and particularly 20 Hz DES on day 3 of stimulation, it looks like somebody is repeatedly lifting the weight support arm where the animal is attached. In the DES 20 Hz this manipulation of the arm even has a lateral movement. The support arm is not fully visible, so it is unclear how this external perturbation came about. Lifting the support arm changes the contact

of the animal to the ground, provides additional sensory cues to the leg, and therefore improves stepping. This is a critical aspect, as you might have not studied the effect of stimulation, but rather the effect of a stimulation plus dynamic weight adjustment, possibly with adjustments restricted to the DES conditions. Maybe that was the case? In the text, you also did not say whether the technicians were blinded to the stimulation conditions, only that they were blinded to trained mice. If that is the case, your DES effects might have potentially resulted from better physiotherapy in the DES condition, and not the stimulation itself. Can you provide any evidence that the technicians trained all animals the same way, without dynamic adjustments in body weight support, and whether they were blinded for the stimulation conditions and experimental design?

Response: Thank you very much for this important comment. We apologize for the confusion that we have created. Due to the absence of a body weight-supported automatic adjustment system in the treadmill used in our study, a research assistant was required to manually adjust the support arm to ensure that the hind limbs of mice made proper contact with the treadmill. At the same time, the research assistant needed to adjust the support arm in real time to maintain the balance of the mice's body when the mice were moving on the treadmill, which resulted in the movement of the support arm. Nonetheless, the research assistant did not subjectively adjust the support arm to help the mice move.

The reviewer's feedback prompted us to conduct a new round of experiments. In the updated video, the research assistant made minimal adjustments to the support arm while maintaining the balance of the mouse body to ensure minimal impact on each mouse. In this study, the assistant solely manipulated the support arm of the treadmill, excluding involvement in the experimental design and grouping, and was ignorant of the stimulus conditions. (Page 56, lines 1154-1155)

Minor remark: double check grammar in these sentence:

38 Overall, the

39 results provide insights into neural signal decoding during spinal sensorimotor circuit

40 reconstruction and suggest a novel approach for treating unique approach for treating

Response: Thank you very much for the valuable feedback provided by the reviewer. We have revised the grammar of this sentence: "Overall, the results provide insights into neural signal decoding during spinal sensorimotor circuit reconstruction, suggesting that EEMS is a promising new method for the treatment of spinal cord

injury.” (Page 2, lines 38-41)

Citations:

Capogrosso, M., et al. A computational model for epidural electrical stimulation of spinal sensorimotor circuits. *The Journal of neuroscience : the official journal of the Society for Neuroscience* 33, 19326-19340 (2013).

Tresch, M.C. & Kiehn, O. Motor coordination without action potentials in the mammalian spinal cord. *Nature neuroscience* 3, 593-599 (2000).

Raastad, M., Enríquez-Denton, M. & Kiehn, O. Synaptic signaling in an active central network only moderately changes passive membrane properties. *Proceedings of the National Academy of Sciences of the United States of America* 95, 10251-10256 (1998).

Asboth, L., et al. Cortico-reticulo-spinal circuit reorganization enables functional recovery after severe spinal cord contusion. *Nature neuroscience* 21, 576-588 (2018).

Bonizzato, M., et al. Brain-controlled modulation of spinal circuits improves recovery from spinal cord injury. *Nature communications* 9, 3015 (2018).

Moraud, E.M., et al. Closed-loop control of trunk posture improves locomotion through the regulation of leg proprioceptive feedback after spinal cord injury. *Scientific reports* 8, 76 (2018).

Bonizzato, M., et al. Multi-pronged neuromodulation intervention engages the residual motor circuitry to facilitate walking in a rat model of spinal cord injury. *Nature communications* 12, 1925 (2021).

Reviewers' Comments:

Reviewer #2:

Remarks to the Author:

The Authors have addressed all my concerns. Now the manuscript describes all experiments in a way that this research may be repeatable. There are no additional changes or questions.

Reviewer #3:

Remarks to the Author:

Thank you for your responses. I have few remaining remarks relating to quality and reproducibility of the rehabilitation training:

1) You still did not explicitly state, whether the rehabilitation training in your original experiments (that fuel all the biological result figures) were blinded. Please comment if this was the case or not, and also state this in your methods. This is an important potential limitation of the study results, that needs transparent communication to the audience.

2) Also, in your response you confirm that in the original experiments there had been a lot of manual body weight adjustments by the rehabilitation trainers, that are poorly controlled. You should add this information to your methods, together with the information that there was no automated body weight support system to determine the actual weight load for each mouse during training.

3) One last question is, whether in your original dataset you made sure that the different stimulation conditions were spread equally among different animal trainers. Again, if you let the trainer interact manually with the body-weight support system, you might have measured the performance of the different trainers, rather than the stimulation conditions. Can you please state, how you ensured that all animals were randomly exposed to the different trainers (per group, and timepoints)? If no random allocation occurred to the trainers please state this as another limitation in the manuscript.

The new dataset appears to reproduce your study results. Overall, I have no further major concerns, and embrace publication with minor remarks above.

REVIEWERS' COMMENTS

Reviewer #2 (Remarks to the Author):

The Authors have addressed all my concerns. Now the manuscript describes all experiments in a way that this research may be repeatable. There are no additional changes or questions.

Reviewer #3 (Remarks to the Author):

Thank you for your responses. I have few remaining remarks relating to quality and reproducibility of the rehabilitation training:

1) You still did not explicitly state, whether the rehabilitation training in your original experiments (that fuel all the biological result figures) were blinded. Please comment if this was the case or not, and also state this in your methods. This is an important potential limitation of the study results, that needs transparent communication to the audience.

2) Also, in your response you confirm that in the original experiments there had been a lot of manual body weight adjustments by the rehabilitation trainers, that are poorly controlled. You should add this information to your methods, together with the information that there was no automated body weight support system to determine the actual weight load for each mouse during training.

3) One last question is, whether in your original dataset you made sure that the different stimulation conditions were spread equally among different animal trainers. Again, if you let the trainer interact manually with the body-weight support system, you might have measured the performance of the different trainers, rather than the stimulation conditions. Can you please state, how you ensured that all animals were randomly exposed to the different trainers (per group, and timepoints)? If no random allocation occurred to the trainers please state this as another limitation in the manuscript.

The new dataset appears to reproduce your study results. Overall, I have no further major concerns, and embrace publication with minor remarks above.

We sincerely appreciate the opportunity to further refine our manuscript and extend our gratitude to the reviewers for their valuable feedback. Below, we provide detailed responses to each of your questions. (*Our responses are in blue text*)

Reviewer #2 (Remarks to the Author):

The Authors have addressed all my concerns. Now the manuscript describes all experiments in a way that this research may be repeatable. There are no additional changes or questions.

Response: Once again, we appreciate the comments of the reviewer 2 and are delighted that our previous responses and revisions satisfactorily addressed her/his points.

Reviewer #3 (Remarks to the Author):

Thank you for your responses. I have few remaining remarks relating to quality and reproducibility of the rehabilitation training:

1) You still did not explicitly state, whether the rehabilitation training in your original experiments (that fuel all the biological result figures) were blinded. Please comment if this was the case or not, and also state this in your methods. This is an important potential limitation of the study results, that needs transparent communication to the audience.

Response: We thank the reviewer for this comment. The reviewer may have misinterpreted the results in Fig. 3 and Fig. S6. The treadmill was only used for trajectory testing of the hind limbs and EMG testing of the TA at the end of the electrical stimulation training, and was not used for rehabilitation training of the mice. Also, all biological results tested in the manuscript were performed after the end of electrical stimulation training and did not involve treadmill training. The treadmill training in the experiment was blinded - this sentence now reads as follows:

“There were three identical treadmills in the laboratory for the experiment, and six research assistants were involved. Each group contained six mice, which were randomly assigned to six research assistants, ensuring that the mice of different stimulus groups were spread equally among different research assistants. Research assistants were blinded to the stimulation condition and grouping of the assigned mice.” (Page 53, lines 1138-1143)

2) Also, in your response you confirm that in the original experiments there had been a lot of manual body weight adjustments by the rehabilitation trainers, that are poorly controlled. You should add this information to your methods, together with the information that there was no automated body weight support system to determine the actual weight load for each mouse during training.

Response: We thank the reviewer for this comment. Based on your suggestion, we have added relevant information to the methods.

“Due to the absence of a body weight-supported automatic adjustment system in the treadmill used in our study, a research assistant was required to manually adjust the support arm to ensure that the hind limbs of mice made proper contact with the treadmill. This resulted in undetermined information about the weight load of each mouse during training.” (Page 52, line 1121-1125)

“At the same time, the research assistant needed to adjust the support arm in real time to maintain the balance of the mice’s body when the mice were moving on the treadmill, which resulted in the movement of the support arm. Nonetheless, the research assistant did not subjectively adjust the support arm to help the mice move. To minimize the impact on each mouse, research assistants were asked to keep the mouse’s body balanced while minimizing adjustments to the support arm.” (Page 52, line 1128-1134)

3) One last question is, whether in your original dataset you made sure that the different stimulation conditions were spread equally among different animal trainers. Again, if you let the trainer interact manually with the body-weight support system, you might have measured the performance of the different trainers, rather than the stimulation conditions. Can you please state, how you ensured that all animals were randomly exposed to the different trainers (per group, and timepoints)? If no random allocation occurred to the trainers please state this as another limitation in the manuscript.

Response: Thank you very much for this important comment. Based on the reviewer’s comment, we speculate that reviewer still have misunderstandings about the use of treadmills and grouping of treadmill experiment. In this study, the treadmill was only a testing tool and was not used for rehabilitation training. As we answered in the first question: the treadmill was only used for trajectory testing of the hind limbs and EMG testing of the TA at the end of the electrical stimulation training, and was not used for rehabilitation training of the mice. Also, all biological results tested in the manuscript were performed only after the end of electrical stimulation training and did not involve treadmill training.

In the original dataset, we ensured that the different stimulus conditions were equally distributed among the different research assistants. In addition, different timepoints are not involved in the treadmill testing, only different groups are included. Therefore, we add relevant information in the manuscript: “Each group contained six mice, which were randomly assigned to six research assistants, ensuring that the mice of different stimulus groups were spread equally among different research assistants. Research assistants were blinded to the stimulation condition and grouping of the assigned mice” (Page 53, lines 1140-1143)

As pointed out by the reviewer, we could not ensure that each research assistant trained the mice to a consistent degree. Therefore, in order to obtain higher quality data, we asked the research assistants to make minimal adjustments to the support arms while maintaining the balance of the mice’s bodies to ensure minimal impact on each mouse. We added the following sentence to the manuscript: “To minimize the

impact on each mouse, research assistants were asked to keep the mouse's body balanced while minimizing adjustments to the support arm." (Page 52, lines 1132-1134)

The new dataset appears to reproduce your study results. Overall, I have no further major concerns, and embrace publication with minor remarks above.

Response: Once again, we appreciate the comments of the reviewer and hope our responses and revisions address your points with satisfaction.